# Evolution of neural activity in circuits bridging sensory and abstract knowledge

**Francesca Mastrogiuseppe[1]\*[†], Naoki Hiratani[2], Peter Latham[1]**

[1]Gatsby Computational Neuroscience Unit, University College London, London, United Kingdom; [2]Center for Brain Science, Harvard University, Harvard, United States

**Abstract** The ability to associate sensory stimuli with abstract classes is critical for survival. How are these associations implemented in brain circuits? And what governs how neural activity evolves during abstract knowledge acquisition? To investigate these questions, we consider a circuit model that learns to map sensory input to abstract classes via gradient-descent synaptic plasticity. We focus on typical neuroscience tasks (simple, and context-dependent, categorization), and study how both synaptic connectivity and neural activity evolve during learning. To make contact with the current generation of experiments, we analyze activity via standard measures such as selectivity, correlations, and tuning symmetry. We find that the model is able to recapitulate experimental observations, including seemingly disparate ones. We determine how, in the model, the behaviour of these measures depends on details of the circuit and the task. These dependencies make experimentally testable predictions about the circuitry supporting abstract knowledge acquisition in the brain.

## Editor's evaluation

The findings of the paper are very valuable for neuroscientists studying the learning of abstract representations. It provides compelling evidence that neural networks trained on two-way classification tasks will develop responses whose category and context selectivity profiles depend on key network details, such as neural activation functions and initial connectivity. These results can explain apparently contradictory results in the experimental literature, and make new experimental predictions for testing in the future.

**\*For correspondence:**
fran.mastrogiuseppe@gmail.com

**Present address:**
[†]Champalimaud Research, Lisbon, Portugal

## Introduction

Everyday decisions do not depend on the state of the world alone; they also depend on internal, non-sensory variables that are acquired with experience. For instance, over time we learn that in most situations salads are good for us while burgers are not, while in other contexts (e.g., before a long hike in the mountains) the opposite is true. The ability to associate sensory stimuli with abstract variables is critical for survival; how these associations are learned is, however, poorly understood.

Although we do not know how associations are learned, we do have access to a large number of experimental studies addressing how neural activity evolves while animals learn to classify stimuli into abstract categories (*Asaad et al., 1998*; *Messinger et al., 2001*; *Freedman et al., 2001*; *Freedman and Assad, 2006*; *Reinert et al., 2021*). Such experiments have probed two kinds of associations between stimuli and categories: fixed associations (*Freedman and Assad, 2006*; *Fitzgerald et al., 2011*; *Cromer et al., 2010*) (in which, e.g., stimuli are either in category A or in category B), and flexible ones (*Wallis et al., 2001*; *Stoet and Snyder, 2004*; *Roy et al., 2010*; *Reinert et al., 2021*) (in which, e.g., stimuli are in category A in one context and category B in another).

A consistent finding in these experiments is that activity of single neurons in associative cortex develops selectivity to task-relevant abstract variables, such as category (*Freedman et al., 2001*; *Fitzgerald et al., 2011*; *Reinert et al., 2021*) and context (*White and Wise, 1999*; *Wallis et al., 2001*; *Stoet and Snyder, 2004*). Neurons, however, typically display selectivity to multiple abstract variables (*Rigotti et al., 2013*), and those patterns of mixed selectivity are often hard to intepret (*Cromer et al., 2010*; *Roy et al., 2010*; *Hirokawa et al., 2019*).

Instead of focussing on one neuron at the time, one can alternatively consider large populations of neurons and quantify how those, as a whole, encode abstract variables. This approach has led, so far, to apparently disparate observations. Classical work indicates that neurons in visual cortex encode simple sensory variables (e.g., two opposite orientations) via negatively correlated responses (*Hubel and Wiesel, 1962*; *Olshausen and Field, 2004*): neurons that respond strongly to a given variable respond weakly to the other one, and vice versa. Those responses, furthermore, are symmetric (*DeAngelis and Uka, 2003*): about the same number of neurons respond strongly to one variable, or the other. In analogy with sensory cortex, one can thus hypothesize that neurons in associative cortex encode different abstract variables (e.g., categories A and B) via negatively correlated, and symmetric responses. Evidence in favour of this type of responses has been reported in monkeys (*White and Wise, 1999*; *Cromer et al., 2010*; *Roy et al., 2010*; *Freedman and Miller, 2008*) and mice (*Reinert et al., 2021*) prefrontal cortex (PFC). However, evidence in favour of a different type of responses has been reported in a different set of experiments from monkeys lateral intraparietal (LIP) cortex (*Fitzgerald et al., 2013*). In that case, responses to categories A and B were found to be positively correlated: neurons that learn to respond strongly to category A also respond strongly to category B, and neurons that learn to respond weakly to category A also respond weakly to category B. Furthermore, responses were strongly asymmetric: almost all neurons displayed the strongest response to the same category (despite monkeys did not display behavioural biases towards one category or the other).

In this work, we use neural circuit models to shed light on these experimental results. To this end, we hypothesize that synaptic connectivity in neural circuits evolves by implementing gradient descent on an error function (*Richards et al., 2019*). A large body of work has demonstrated that, under gradient-descent plasticity, neural networks can achieve high performance on both simple and complex tasks (*LeCun et al., 2015*). Recent studies have furthermore shown that gradient-descent learning can be implemented, at least approximately, in a biologically plausible way (*Lillicrap et al., 2016*; *Whittington and Bogacz, 2017*; *Sacramento et al., 2018*; *Akrout et al., 2019*; *Payeur et al., 2021*; *Pogodin and Latham, 2020*; *Boopathy and Fiete, 2022*). Concomitantly, gradient-based learning has been used to construct network models for a variety of brain regions and functions (*Yamins and DiCarlo, 2016*; *Kell et al., 2018*; *Mante et al., 2013*; *Chaisangmongkon et al., 2017*). A precise understanding of how gradient-descent learning shapes representations in neural circuits is however still lacking.

Motivated by this hypothesis, we study a minimal circuit model that learns through gradient descent to associate sensory stimuli with abstract categories, with a focus on tasks inspired by those used in experimental studies. Via mathematical analysis and simulations, we show that the model can capture the experimental findings discussed above. In particular, after learning, neurons in the model become selective to category and, if present, context; this result is robust, and independent of the details of the circuit and the task. On the other hand, whether correlations after learning are positive or negative, and whether population tuning to different categories is asymmetric or not, is not uniquely determined, but depends on details. We determined how, in the model, activity measures are modulated by circuit details (activation function of single neurons, learning rates, initial connectivity) and task features (number of stimuli, and whether or not the associations are context dependent). These dependencies make experimentally testable predictions about the underlying circuitry. Overall, the model provides a framework for interpreting seemingly disparate experimental findings, and for making novel experimental predictions.

## Results

We consider classification into mutually exclusive abstract classes which, as above, we refer to as categories A and B. We consider two tasks: a simple, linearly separable one (*Freedman and Assad, 2006*; *Fitzgerald et al., 2011*; *Cromer et al., 2010*) and a context-dependent, nonlinearly separable

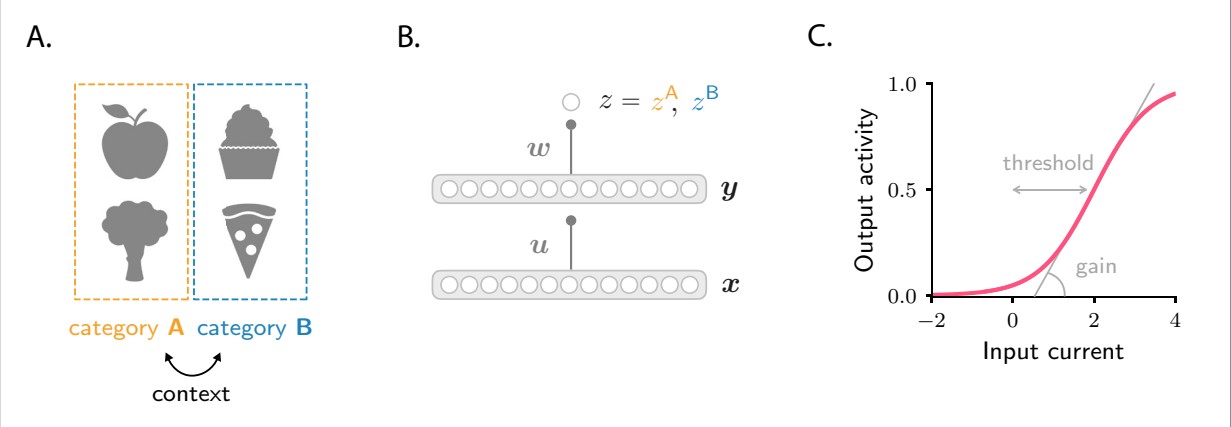

**Figure 1.** Schematics of tasks and circuit model used in the study. (**A**) Illustration of the two categorization tasks. In the simple categorization task, half the stimuli are associated with category A and the other half with category B. In the context-dependent task, associations are reversed across contexts: stimuli associated with category A in context 1 are associated with category B in context 2, and vice versa. (**B**) The circuit consists of a sensory input layer ($x$), an intermediate layer ($y$), and a readout neuron ($z$). The intermediate ($u$) and readout ($w$) weights evolve under gradient descent plasticity (***Equation 1***). (**C**) The activation functions, $\Psi$ and $\Phi$, are taken to be sigmoids characterized by a threshold and a gain. The gain, which controls the sensitivity of activity to input, is the slope of the function at its steepest point; the threshold, which controls activity sparsity, is the distance from the steepest point to zero.

one (***Wallis et al., 2001***; ***Roy et al., 2010***; ***Reinert et al., 2021***; ***Figure 1A***). We assume that for both, categorization is implemented with a two-layer circuit, as shown in ***Figure 1B***, and that the synaptic weights evolve via gradient descent. Our goal is to determine how the activity in the intermediate layer evolves with learning, and how this evolution depends on the task and the biophysical details of the circuit. We start by describing the model. We then consider circuits that learn the simple, linearly separable, categorization task, and analyze how learning drives changes in activity. Finally, we extend the analysis to the context-dependent, nonlinearly separable, task.

## Circuit model

We consider a simple feedforward circuit as in ***Figure 1B***. A vector $x$, which models the input from sensory areas, is fed into an intermediate layer of neurons which represents a higher-level, associative area. The intermediate layer activity is given by $y = \Psi(u \cdot x)$, where $u$ is a all-to-all connectivity matrix. That activity projects to a readout neuron, which learns, over time, to predict the category associated with each sensory input. The activity of the readout neuron, $z$, is taken to be $z = \Phi(w \cdot y)$, where $w$ is a readout vector. The activation functions $\Phi$ and $\Psi$ are sigmoidals that encapsulate the response properties of single neurons; they are parametrized by a threshold and a gain (***Figure 1C***; Materials and methods Circuit).

The goal of the circuit is to adjust the synaptic weights, $u$ and $w$, so that the readout neuron fires at rate $z = z^A$ when the sensory input is associated with category A, and at rate $z = z^B$ when the sensory input is associated with B (***Figure 1B***). In the simple categorization task, half the stimuli are associated with category A and the other half with B. In the context-dependent task, associations are reversed across contexts: stimuli associated with category A in context 1 are associated with category B in context 2, and vice versa (***Figure 1A***). We use $\mathcal{E}(u, w)$ to denote the average error between $z$ and its target value, and assume that the synaptic weights evolve, via gradient descent, to minimize the error. If the learning rates are small, the weights evolve according to

$$\frac{\mathrm{d}u}{\mathrm{d}t} = -\eta_u \frac{\partial \mathcal{E}(u, w)}{\partial u} \tag{1a}$$

$$\frac{\mathrm{d}w}{\mathrm{d}t} = -\eta_w \frac{\partial \mathcal{E}(u, w)}{\partial w}, \tag{1b}$$

where $t$ represents learning time and $\eta_u$ and $\eta_w$ are learning rates which, for generality, we allow to be different.

Before learning, the synaptic weights are random. Consequently, activity in the intermediate layer, $\boldsymbol{y}$, is unrelated to category, and depends only on sensory input. As the circuit learns to associate sensory inputs with abstract categories, task-relevant structure emerges in the connectivity matrix $\boldsymbol{u}$, and thus in the intermediate layer as well. Analyzing how activity in the intermediate layer evolves over learning is the focus of this work.

## Evolution of activity during the simple categorization task

We first analyze the simple task, for which we can derive results in a transparent and intuitive form. We then go on to show that similar (although richer) results hold for the context-dependent one.

In the simple categorization task, each sensory input vector $\boldsymbol{x}^s$ represents a stimulus (for example, an odor, or an image), which is associated with one of the two mutually exclusive categories A and B. In the example shown below, we used 20 stimuli, of which half are associated with category A, and the other half are associated with category B. Sensory input vectors corresponding to different stimuli are generated at random and assumed to be orthogonal to each other; orthogonality is motivated by the decorrelation performed by sensory areas (but this assumption can be dropped without qualitatively changing the main results, see Materials and methods Simple categorization task with structured inputs and heterogeneity and *Figure 2—figure supplement 4*).

We start our analysis by simulating the circuit numerically, and investigating the properties of neural activity, $\boldsymbol{y}$, in the intermediate layer. A common way to characterize the effects of learning on single-neuron activity is through the category selectivity index, a quantity that is positive when activity elicited by within-category stimuli is more similar than activity elicited by across-category stimuli, and negative otherwise. It is defined as (*Freedman et al., 2001*; *Freedman and Assad, 2006*; *Reinert et al., 2021*) (Materials and methods Simple task: category selectivity)

$$S_i = \frac{\langle (y_i^s - y_i^{s'})^2 \rangle_{s,s' \text{ diff cat}} - \langle (y_i^s - y_i^{s'})^2 \rangle_{s,s' \text{ same cat}}}{\langle (y_i^s - y_i^{s'})^2 \rangle_{s,s' \text{ diff cat}} + \langle (y_i^s - y_i^{s'})^2 \rangle_{s,s' \text{ same cat}}} \tag{2}$$

where $y_i^s$ represents the activity of neuron $i$ in response to sensory input $s$, and angle brackets, $\langle \cdot \rangle_{s,s'}$, denote an average over sensory input pairs. The subscript 'same cat' refers to averages over the same category (A–A or B–B) and 'diff cat' to averages over different categories (A–B).

Before learning, the responses of single neurons to different stimuli are random and unstructured. Thus, responses to stimuli paired with category A are statistically indistinguishable from responses to stimuli paired with category B (*Figure 2A*). This makes the category selectivity index zero on average (*Figure 2B*). After learning, the responses of single neurons depend on category: within-category responses become more similar than across-category responses, resulting in two separate distributions (*Figure 2E*). As a consequence, the category selectivity index for most neuron increases; correspondingly, average selectivity increases from zero to positive values (*Figure 2F*), thus reproducing the behaviour observed in experimental studies (*Freedman et al., 2001*; *Freedman and Assad, 2006*; *Reinert et al., 2021*). To determine whether this effect is robust, we varied the parameters that describe the task (number of stimuli) and the biophysical properties of the circuit (the threshold and gain of neurons, *Figure 1C*, and the learning rates of the two sets of synaptic weights, $\eta_u$ and $\eta_w$). We found that the selectivity increase is a universal property – it is observed in all circuit models that successfully learned the task, independent of the parameters. Activity from a second example circuit is shown in *Figure 2I, J*; additional simulations are shown in *Figure 2—figure supplement 1A*.

Category selectivity tells us about the behaviour of single neurons. But how does the population as a whole change its activity over learning? To quantify that, we compute signal correlations, defined to be the Pearson correlation coefficient between the activity elicited by two different stimuli (*Cromer et al., 2010*). Results are summarized in the correlation matrices displayed in *Figure 2C, G, K*. As the task involves 20 stimuli, the correlation matrix is 20 × 20; stimuli are sorted according to category.

As discussed above, before learning the responses of neurons in the intermediate layer are random and unstructured. Thus, activity in response to different stimuli is uncorrelated; this is illustrated in *Figure 2C*, where all non-diagonal entries of the correlation matrix are close to zero. Of particular interest are the upper-right and lower-left blocks of the matrix, which correspond to pairs of activity vectors elicited by stimuli in different categories. The average of those correlations, which we refer to as category correlation, is shown to the right of each correlation matrix. Before learning, the category correlation is close to zero (*Figure 2C*). Over learning, the correlation matrices develop structure.

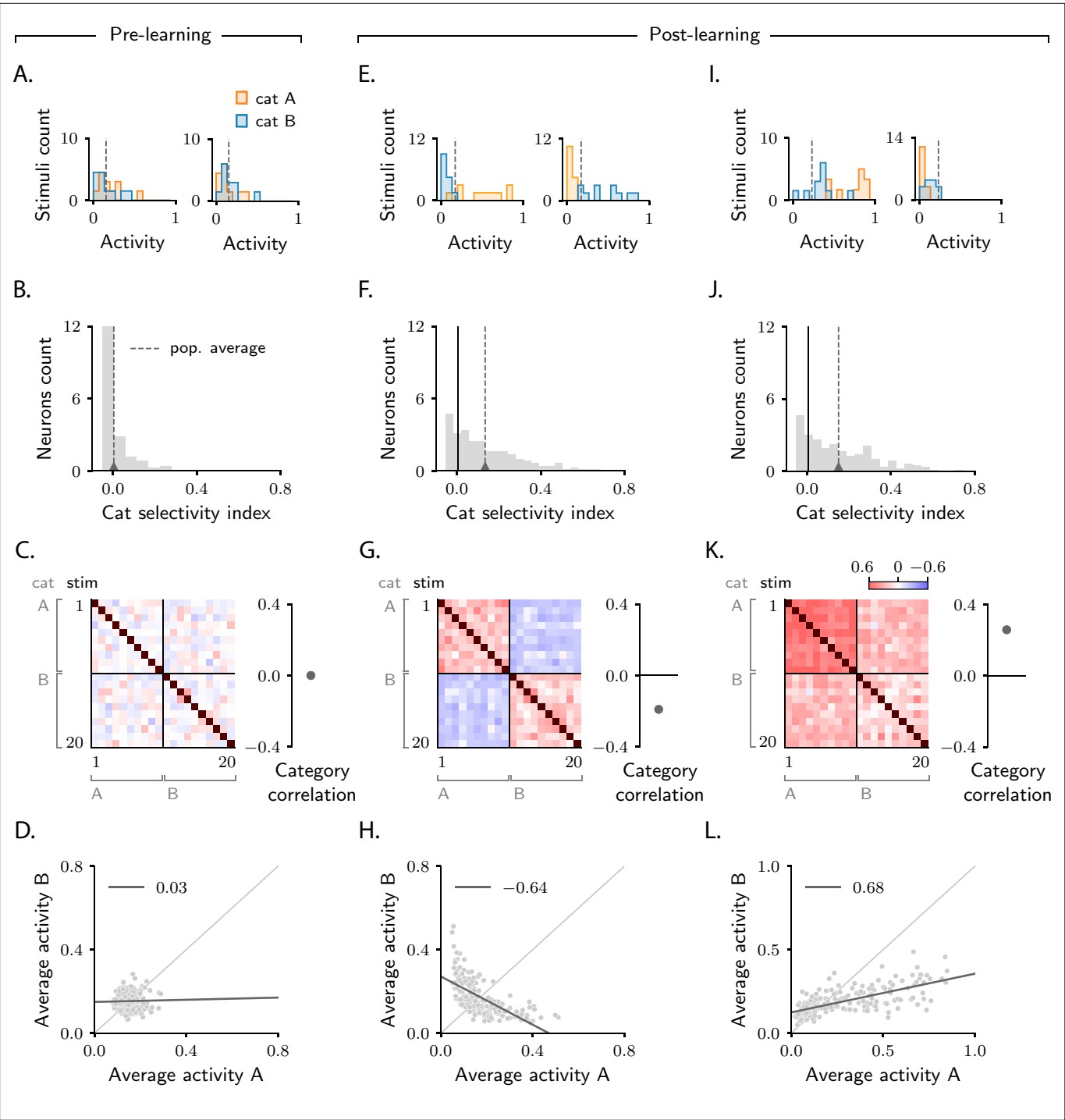

**Figure 2.** Characterization of activity evolution during the simple categorization task. Results from simulations. The first column (**A–D**) shows a naive circuit (pre-learning); the second (**E–H**) and third (**I–L**) columns show two trained circuits (post-learning), characterized by different sets of parameters (see below). (**A, E, I**) Histograms of single-neuron activity in response to stimuli associated with category A (orange) and category B (blue). Left and right show two sample neurons from the intermediate layer. Grey dashed lines indicate the average activity across the population. (**B, F, J**) Histograms of category selectivity (*Equation 2*) across the population of neurons in the intermediate layer. Grey dashed lines indicate the average selectivity across the population. In panels F and J, the black vertical lines indicate the initial value of average selectivity. (**C, G, K**) Signal correlation matrices. Each entry shows the Pearson correlation coefficient, averaged over neurons (*Equation 72*), between activity elicited by different stimuli. In these examples, we used 20 stimuli. Diagonal entries (brown) are all equal to 1. Category correlation (namely, the average of the correlations within the off-diagonal blocks, which contain stimuli in different categories) is shown on the right of the matrices. In panels G and K, the black horizontal lines near zero indicate the initial values of category correlation. (**D, H, L**) Population responses to categories A and B. Each dot represents a neuron in the intermediate layer, with

*Figure 2 continued on next page*

*Figure 2 continued*

horizontal and vertical axes showing the responses to stimuli associated with categories A and B, respectively, averaged over stimuli. Grey line: linear fit, with Pearson correlation coefficient shown in the figure legend. Parameters are summarized in *Table 1* (Materials and methods Tables of parameters).

The online version of this article includes the following figure supplement(s) for figure 2:

**Figure supplement 1.** Characterization of activity evolution during the simple categorization task; additional results, part I.

**Figure supplement 2.** Characterization of activity evolution during the simple categorization task; additional results, part II.

**Figure supplement 3.** Learning curves.

**Figure supplement 4.** Simple categorization task with structured inputs and heterogeneity.

Correlations become different within the two diagonal, and the two off-diagonal blocks, indicating that learning induces category-dependent structure. In *Figure 2G*, the average correlation within the off-diagonal blocks is negative; the category correlation is thus negative (*Cromer et al., 2010*; *Roy et al., 2010*; *Freedman and Miller, 2008*). The model does not, however, always produce negative correlation: varying model details – either the parameters of the circuit or the number of stimuli – can switch the category correlation from negative to positive (*Fitzgerald et al., 2013*; one example is shown in *Figure 2K*).

To illustrate the difference in population response when category correlation is negative versus positive, for each neuron in the intermediate layer we plot the average response to stimuli associated with category B (vertical axis) versus A (horizontal axis). Before learning, activity is unstructured, and the dots form a random, uncorrelated cloud (*Figure 2D*). After learning, the shape of this cloud depends on category correlation. In *Figure 2H*, where the category correlation is negative, the cloud has a negative slope. This is because changes in single-neuron responses to categories A and B have opposite sign: a neuron that increases its activity in response to category A decreases its activity in response to category B (*Figure 2E* left), and vice versa (*Figure 2E* right). In *Figure 2L*, where the category correlation is positive, the cloud has, instead, a positive slope. Here, changes in single-neuron responses to categories A and B have the same sign: a neuron that increases its activity in response to category A also increases its activity in response to category B (*Figure 2I*, left), and similarly for a decrease (*Figure 2I*, right).

Negative versus positive slope is not the only difference between *Figure 2H and L*: they also differ in symmetry with respect to the two categories. In *Figure 2H*, about the same number of neurons respond more strongly to category A than to category B (*Reinert et al., 2021*). In *Figure 2L*, however, the number of neurons that respond more strongly to category A is significantly larger than the number of neurons that respond more strongly to category B (*Fitzgerald et al., 2013*). Furthermore, as observed in experiments reporting positive correlations (*Fitzgerald et al., 2013*), the mean population activity in response to category A is larger than to category B, and the range of activity in response to A is larger than to B. The fact that the population response to A is larger than to B is not a trivial consequence of having set a larger target for the readout neuron in response to A than to B ($z^A > z^B$): as shown in *Figure 2—figure supplement 2B, D*, example circuits displaying larger responses to B can also be observed. Response asymmetry is discussed in detail in Materials and methods Asymmetry in category response.

In sum, we simulated activity in circuit models that learn to associate sensory stimuli to abstract categories via gradient-descent synaptic plasticity. We observed that single neurons consistently develop selectivity to abstract categories – a behaviour that is robust with respect to model details. How the population of neurons responds to category depended, however, on model details: we observed both negatively correlated, symmetric responses and positively correlated, asymmetric ones. These observations are in agreement with experimental findings (*Freedman and Assad, 2006*; *Fitzgerald et al., 2013*; *Cromer et al., 2010*; *Reinert et al., 2021*).

## Analysis of the simple categorization task

What are the mechanisms that drive activity changes over learning? And how do the circuit and task details determine how the population responds? To address these questions, we performed mathematical analysis of the model. Our analysis is based on the assumption that the number of neurons in each layer of the circuit is much larger than the number of sensory inputs to classify – a regime that is relevant to the systems and tasks we study here. In that regime, the number of synaptic weights that the circuit can tune is very large, and so a small change in each weight is sufficient to learn the task.

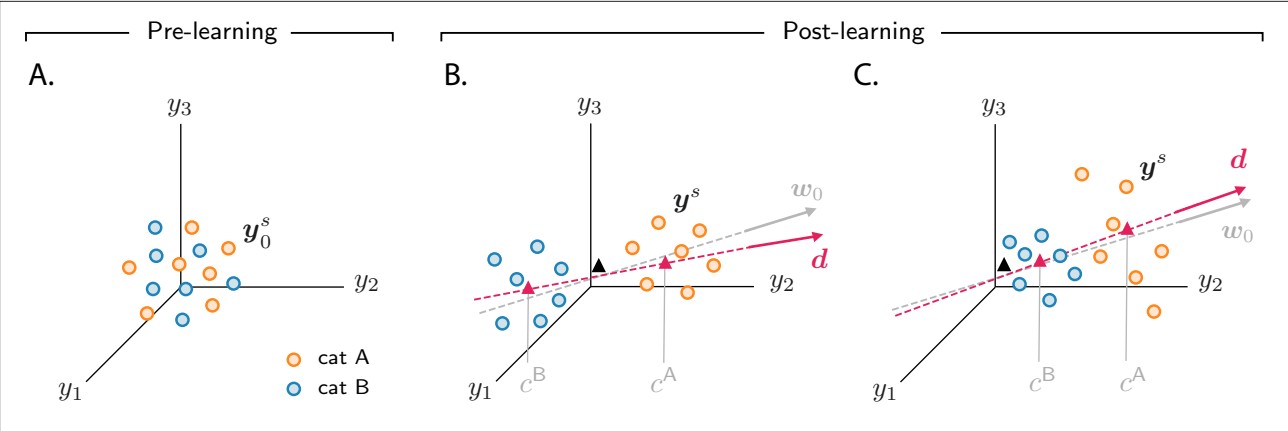

**Figure 3.** Analysis of activity evolution during the simple categorization task. Results from mathematical analysis. (**A–C**) Cartoons illustrating how activity evolves over learning. The three columns are as in **Figure 2**: pre-learning (first column) and post-learning for two different circuits (second and third columns). Circles show activity in the intermediate layer in response to different stimuli, displayed in a three-dimensional space where axes correspond to the activity of three sample neurons. Orange and blue circles are associated, respectively, with categories A and B. Before learning, activity is unstructured (panel A). After learning (panels B and C), the activity vectors develop a component along the common direction $d$ (**Equation 3**), shown as a magenta line, and form two clouds, one for each category. The centers of those clouds are indicated by magenta triangles; their positions along $d$ are given, approximately, by $c^A$ and $c^B$. The black triangle indicates the center of initial activity. In panel B, $c^A$ and $c^B$ have opposite sign, so the clouds move in opposite directions with respect to initial activity; in panel C, $c^A$ and $c^B$ have the same sign, so the clouds move in the same direction. For illustration purposes, we show a smaller number of stimuli (14, instead of 20) than in **Figure 2**. Simulated data from the circuits displayed in **Figure 2** are shown in **Figure 2—figure supplement 1B**.

The online version of this article includes the following figure supplement(s) for figure 3:

**Figure supplement 1.** Comparison between finite-size networks and approximate mathematical description for the simple categorization task, part I.

**Figure supplement 2.** Comparison between finite-size networks and approximate mathematical description for the simple categorization task, part II.

This makes the circuit amenable to mathematical analysis (**Jacot et al., 2018**; **Lee et al., 2019**; **Liu et al., 2020**; **Hu et al., 2020**); full details are reported in Materials and methods Evolution of connectivity and activity in large circuits, here we illustrate the main results.

We start with the simple categorization task illustrated in the previous section, and use the mathematical framework to shed light on the simulations described above (**Figure 2**). **Figure 3A** shows, schematically, activity in the intermediate layer before learning (see **Figure 2—figure supplement 1B** for simulated data). Axes on each plot correspond to activity of three sample neurons. Each dot represents activity in response to a different sensory input; orange and blue dots indicate activity in response to stimuli associated with categories A and B, respectively. Before learning, activity is determined solely by sensory inputs, which consist of random, orthogonal vectors. Consequently, the initial activity vectors form an unstructured cloud in activity space, with orange and blue circles intermingled (**Figure 3A**).

Over learning, activity vectors in **Figure 3A** move. Specifically, over learning all activity vectors acquire a component that is aligned with a common, stimulus-independent direction. Activity after learning can thus be approximated by

$$\boldsymbol{y}^s \simeq \boldsymbol{y}_0^s + c^s \boldsymbol{d} \tag{3}$$

where $\boldsymbol{y}_0^s$ indicates initial activity in response to sensory input $s$, and $\boldsymbol{d}$ indicates the common direction along which activity acquires structure. The coefficients $c^s$, which measure the strength of the components along the common direction $\boldsymbol{d}$, are determined by category: they are approximately equal to $c^A$ if the sensory input is associated with category A, and $c^B$ otherwise. Consequently, over learning, activity vectors associated with different categories are pushed apart along $\boldsymbol{d}$; this is illustrated in **Figure 3B, C**, which show activity for the two circuits analyzed in the second and third column of **Figure 2**, respectively. Activity thus forms two distinct clouds, one for each category; the centers of the two clouds along $\boldsymbol{d}$ are given, approximately, by $c^A$ and $c^B$. The mathematical framework detailed in Materials and methods Simple categorization task allows us to derive closed-form expressions for the clustering direction $\boldsymbol{d}$ and the coefficients $c^A$ and $c^B$. In the next two sections, we take advantage

of those expressions to determine how the different activity patterns shown in *Figure 2* depend on task and circuit parameters.

The fact that activity clusters by category tells us immediately that the category selectivity index of single neurons increases over learning, as observed in simulations (*Figure 2F, J*). To see this quantitatively, note that from the point of view of a single neuron, $i$, *Equation 3* reads

$$y_i^s \simeq y_{0,i}^s + c^s d_i. \tag{4}$$

Since $c^s$ is category dependent, while $d_i$ is fixed, the second term in the right-hand side of *Equation 4* separates activity associated with different categories (*Figure 2E, I*), and implies an increase in the category selectivity index (*Equation 2*; *Figure 2F, J*). The generality of *Equation 4* indicates that the increase in selectivity is a robust byproduct of gradient-descent learning, and so can be observed in any circuit that learns the categorization task, regardless of model details. This explains the increase in selectivity consistently observed in simulations (*Figure 2F, J* and *Figure 2—figure supplement 1A*).

## Correlations reflect circuit and task properties

While the behaviour of category selectivity is consistent across all circuit models, the behaviour of population responses is not: as shown in *Figure 2*, over learning responses can become negatively correlated and symmetric (*Figure 2G, H*), or positively correlated and asymmetric (*Figure 2K, L*). The reason is illustrated in *Figure 3B, C*. In *Figure 3B*, the centers of the category clouds along $d$, $c^A$ and $c^B$, have, respectively, a positive and a negative sign relative to the center of initial activity (denoted by a black triangle). As a consequence, the two clouds move in opposite directions. The population thus develops, over learning, negative category correlation (*Figure 2G, H*): if the activity of a given

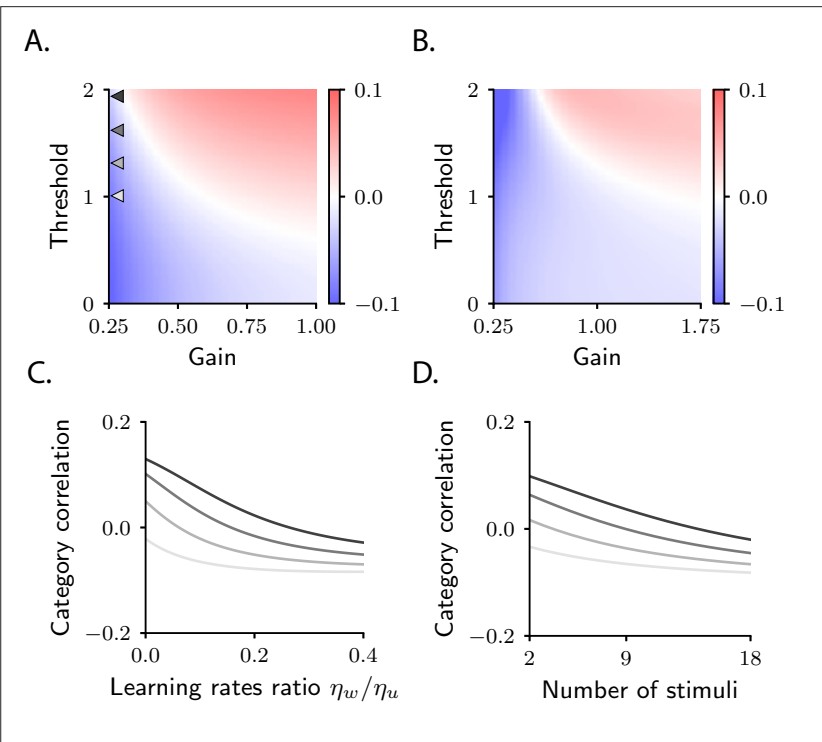

**Figure 4.** Category correlation depends on circuit and task properties. (**A**) Category correlation as a function of the threshold and gain of the readout neuron. Grey arrows indicate the threshold and gain that are used in panels C and D. The learning rate ratio, $\eta_w/\eta_u$, is set to 0.4 here and in panels B and D. (**B**) Category correlation as a function of the threshold and gain of neurons in the intermediate layer; details as in panel A. (**C**) Category correlation as a function of the learning rate ratio. The threshold and gain of the readout neuron are given by the triangles indicated in panel A, matched by colour. (**D**) Category correlation as a function of the number of stimuli; same colour code as in panel C. In all panels, correlations were computed from the approximate theoretical expression given in Materials and methods Simple task: category correlation (*Equation 74*). Parameters are summarized in *Table 1* (Materials and methods Tables of parameters).

neuron increases for one category, it decreases for the other, and vice versa. Furthermore, if $c^A$ and $c^B$ have similar magnitude (which is the case for **Figure 2G, H**), activity changes for the two categories have similar amplitude, making the response to categories A and B approximately symmetric. In **Figure 3C**, on the other hand, $c^A$ and $c^B$ are both positive; clouds associated with the two categories move in the same direction relative to the initial cloud of activity. This causes the population to develop positive category correlation (**Figure 2K, L**): if the activity increases for one category, it also increases for the other, and similarly for a decrease. Because the magnitude of $c^A$ is larger than $c^B$, activity changes for category A are larger than for B, making the response to categories A and B asymmetric.

This analysis tells us that whether negative or positive category correlation emerges depends on the relative signs of $c^A$ and $c^B$. We can use mathematical analysis to compute the value and sign of $c^A$ and $c^B$, and thus predict how category correlation changes over learning (Materials and methods Simple task: category correlation). We find that the biophysical details of the circuit play a fundamental role in determining category correlation. In **Figure 4A**, we show category correlation as a function of the threshold and gain of the readout neuron (**Figure 1C**). We find that varying those can change the magnitude and sign of correlations, with positive correlations favoured by large values of the threshold and gain and negative correlations favoured by small values. Category correlation is also affected by the threshold and gain of neurons in the intermediate layer. This can be seen in **Figure 4B**, which shows that larger values of the threshold and gain tend to favour positive correlation. An equally important role is played by the relative learning rates of the the readout, $w$, and the intermediate weights, $u$. As illustrated in **Figure 4C**, increasing the ratio of the learning rates, $\eta_w/\eta_u$, causes the correlation to decrease. Overall, these results indicate that category correlation depends on multiple biophysical aspects of the circuit, which in turn are likely to depend on brain areas. This suggests that correlation can vary across brain areas, which is in agreement with the observation that positive correlations reported in monkeys area LIP are robust across experiments (**Fitzgerald et al., 2013**), but inconsistent with the correlations observed in monkeys PFC (**Cromer et al., 2010**).

Category correlation also depends on the total number of stimuli, a property of the task rather than the circuit (Materials and methods Simple task: category correlation, **Equation 77**). This is illustrated in **Figure 4D**, which shows that increasing the number of stimuli causes a systematic decrease in correlation. The model thus makes the experimentally testable prediction that increasing the number of stimuli should push category correlation from positive to negative values. This finding is in agreement with the fact that negative correlations are typically observed in sensory cortex, as well as machine-learning models trained on benchmark datasets (**Papyan et al., 2020**) – that is, in cases where the number of stimuli is much larger than in the current task.

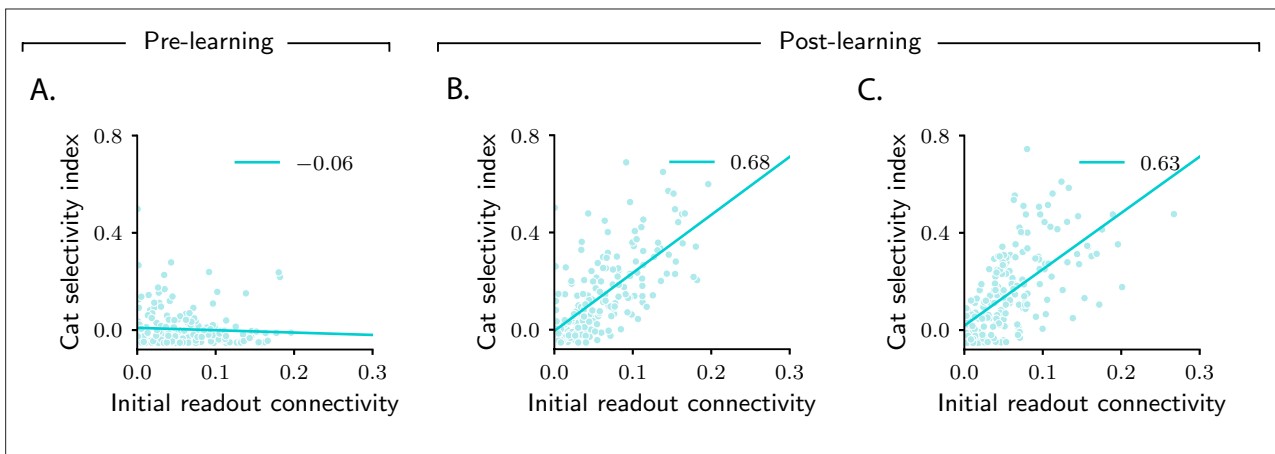

**Figure 5.** Magnitude of category selectivity depends on connectivity with the readout neuron. (**A–C**) Category selectivity as a function of the initial readout connectivity $w_{0,i}$ (in absolute value). The three columns are as in **Figure 2**: pre-learning (first column) and post-learning for two different circuits (second and third columns). Each dot represents a neuron in the intermediate layer. Cyan line: linear fit, with Pearson correlation coefficient shown in the figure legend.

## Patterns of selectivity are shaped by initial connectivity

We conclude our analysis of the simple categorization task by taking a closer look at category selectivity. We have already observed, in *Figure 2F, J*, that the category selectivity of neurons in the intermediate layer increase over learning. However, as shown in those figures, the amount it increases can vary markedly across the population – a finding that reproduces the variability commonly seen in experiments (*Freedman and Assad, 2006*; *Fitzgerald et al., 2011*; *Reinert et al., 2021*). The model naturally explains this variability: as can be seen in *Equation 4*, the magnitude of category-related activity changes (and, consequently, the magnitude of category selectivity) depends, for a given neuron $i$, on the magnitude of $d_i$. Mathematical analysis (see Materials and methods Simple task: computing activity, especially *Equation 55*) indicates that, for the current task, the category direction $\boldsymbol{d}$ is approximately aligned with the vector that specifies connectivity between the intermediate and the readout neurons, $\boldsymbol{w}$, before learning starts; we denote this vector $\boldsymbol{w}_0$ (*Figure 3B, C*). As a consequence, only neurons that are initially strongly connected to the readout neuron – that is, neurons for which $w_{0,i}$ is large – exhibit a large selectivity index (*Figure 5B, C*).

Why does activity cluster along the initial readout $\boldsymbol{w}_0$? As described above, the output of the circuit, $z$, depends on the dot product $\boldsymbol{w} \cdot \boldsymbol{y}$, where $\boldsymbol{w}$ are the readout weights after learning. Consequently, the final activity in the intermediate layer, $\boldsymbol{y}$, must include a category-dependent component along $\boldsymbol{w}$. Such a component can be aligned either with the initial readout weights, $\boldsymbol{w}_0$, or with the readout weights changes. The fact that activity changes are mostly aligned with $\boldsymbol{w}_0$ indicates that the learning algorithm is characterized by a sort of inertia, which makes it rely on initial connectivity structure much more heavily than on the learned one. As showed in Materials and methods Evolution of connectivity and activity in large circuits, this is a property of networks with a large number of neurons relative to the number of stimuli, which are characterized by small weights changes (*Jacot et al., 2018*).

In terms of biological circuits, *Figure 5* predicts that changes in selectivity are determined by the strength of synaptic connections a neuron makes, before learning, to downstream readout areas. Experiments consistent with this prediction have been recently reported: studies in rodents PFC (*Ye et al., 2016*; *Hirokawa et al., 2019*) found that all neurons which were highly selective to a given abstract variable were characterized by similar downstream projections (i.e., they projected to the same area). These experiments would provide evidence for our model if two conditions were met. First, neurons in the downstream area should behave as readout neurons: over learning, their activity should increasingly depend on the abstract variable. Second, the strength of the synaptic connections that neurons make to downstream neurons should correlate with selectivity (*Figure 5B, C*). Both predictions could be tested with current experimental technology.

In sum, we analyzed activity in the intermediate layer of circuits that learned the simple categorization task. We found that activity gets reshaped along a common, stimulus-independent direction (*Equation 3*), which is approximately aligned with the initial readout vector $\boldsymbol{w}_0$. Activity vectors associated with different categories develop two distinct clouds along this direction – a fact that explains the increase in category selectivity observed in *Figure 2F, J*. We also found that the sign of the category correlation depends on the circuit (threshold and gain of neurons in the intermediate and readout layers, and relative learning rates) and on the task (number of stimuli). Modifying any of these can change the direction the clouds of activity move along $\boldsymbol{w}_0$, which in turn changes the sign of category correlation, thus explaining the different behaviours observed in *Figure 2G, H and K, L*.

## Evolution of activity during the context-dependent categorization task

We now consider a more complex categorization task. Here, stimuli–category associations are not fixed, but context dependent: stimuli that are associated with category A in context 1 are associated with category B in context 2, and vice versa. Context-dependent associations are core to a number of experimental tasks (*Wallis et al., 2001*; *Stoet and Snyder, 2004*; *Roy et al., 2010*; *McKenzie et al., 2014*; *Reinert et al., 2021*), and are ubiquitous in real-world experience.

In the model, the two contexts are signaled by distinct sets of context cues (e.g., two different sets of visual stimuli) (*Wallis et al., 2001*; *Stoet and Snyder, 2004*). As for the stimuli, context cues are represented by random and orthogonal sensory input vectors. On every trial, one stimulus and one context cue are presented; the corresponding sensory inputs are combined linearly to yield the total sensory input vector $\boldsymbol{x}^s$ (Materials and methods Context-dependent task: task definition). This task is computationally much more involved than the previous one, primarily because context induces

nontrivial correlational structure: in the simple task, all sensory input vectors were uncorrelated; in the context-dependent task, that is no longer true. For instance, two sensory inputs with the same stimulus and different context cues are highly correlated. In spite of this high correlation, though, they can belong to different categories – for instance, when context cues are associated with different contexts. In contrast, two sensory inputs with different stimuli and different context cues are uncorrelated, but they can belong to the same category. From a mathematical point of view, this correlational structure makes sensory input vectors nonlinearly separable. This is in stark contrast to the simple task, for which sensory input vectors were linearly separable (*Barak et al., 2013*). In fact, this task is a generalization of the classical XOR task where, rather than just two stimuli and two context cues, there are more than two of each (*McKenzie et al., 2014*). In the example shown below, we used 8 stimuli and 8 context cues.

We are again interested in understanding how activity in the intermediate layer evolves over learning. We start by investigating this via simulations (*Figure 6*). As in *Figure 2B, F, J*, we first measure category selectivity (*Equation 2*). Before learning, activity is characterized by small selectivity, which is weakly negative on average (*Figure 6A*; the fact that average category selectivity is initially weakly negative is due to the composite nature of inputs for this task, see Materials and methods Detailed analysis of category selectivity). Over learning, the average category selectivity increases (*Figure 6D*). We tested the robustness of this behaviour by varying the parameters that control both the circuit (threshold and gain of neurons, learning rates) and task (number of stimuli and context cues). As in the simple task, we found that the average category selectivity increases in all circuit models, regardless of the parameters (*Figure 6G* and *Figure 6—figure supplement 1A*).

While in the simple task we could only investigate the effect of category on activity, in this task we can also investigate the effect of context. For this we measure context selectivity which, analogously to category selectivity, quantifies the extent to which single-neuron activity is more similar within than across contexts (Materials and methods Context-dependent task: category and context selectivity, *Equation 122*). Context selectivity is shown in *Figure 6B, E*. We find, as we did for category selectivity, that average context selectivity increases over learning – a behaviour that is in agreement with experimental findings (*Wallis et al., 2001*; *Stoet and Snyder, 2004*). The increase in context selectivity is, as for category, highly robust, and does not depend on model details (*Figure 6H* and *Figure 6—figure supplement 1A*).

Finally, we analyze signal correlations; these are summarized in the correlation matrices displayed in *Figure 6C, F, I*. As we used 8 stimuli and 8 context cues, and all stimuli–context cues combinations are permitted, each correlation matrix is 64 × 64. Trials are sorted according to context cue first and stimulus second; with this ordering, the first half of trials corresponds to context 1 and the second half to context 2, and the off-diagonal blocks are given by pairs of trials from different contexts.

*Figure 6C* shows the correlation matrix before learning. Here, the entries in the correlation matrix are fully specified by sensory input, and can take only three values: large (brown), when both the stimuli and the context cues are identical across the two trials; intermediate (red), when the stimuli are identical but the context cues are not, or vice versa; and small (white), when both stimulus and context cues are different. *Figure 6F, I* show correlation matrices after learning for two circuits characterized by different parameters. As in the simple task, the matrices acquire additional structure during learning, and that structure can vary significantly across circuits (*Figure 6F, I*). To quantify this, we focus on the off-diagonal blocks (pairs of trials from different contexts) and measure the average of those correlations, which we refer to as context correlation. Context correlation behaves differently in the two circuits displayed in *Figure 6F and I*: it decreases over learning in *Figure 6F*, whereas it increases in *Figure 6I*. Thus, as in the simple task, the behaviour of correlations is variable across circuits. This variability is not restricted to context correlation: as in the simple task, category correlation is also variable (*Figure 6—figure supplement 1A*), and the population response to categories A and B can be symmetric or asymmetric depending on model details (*Figure 6—figure supplement 2A, B*).

## Analysis of the context-dependent categorization task

To uncover the mechanisms that drive learning-induced activity changes, we again analyse the circuit mathematically. The addition of context makes the analysis considerably more complicated than for the simple task; most of the details are thus relegated to Materials and methods Context-dependent categorization task; here we discuss the main results.

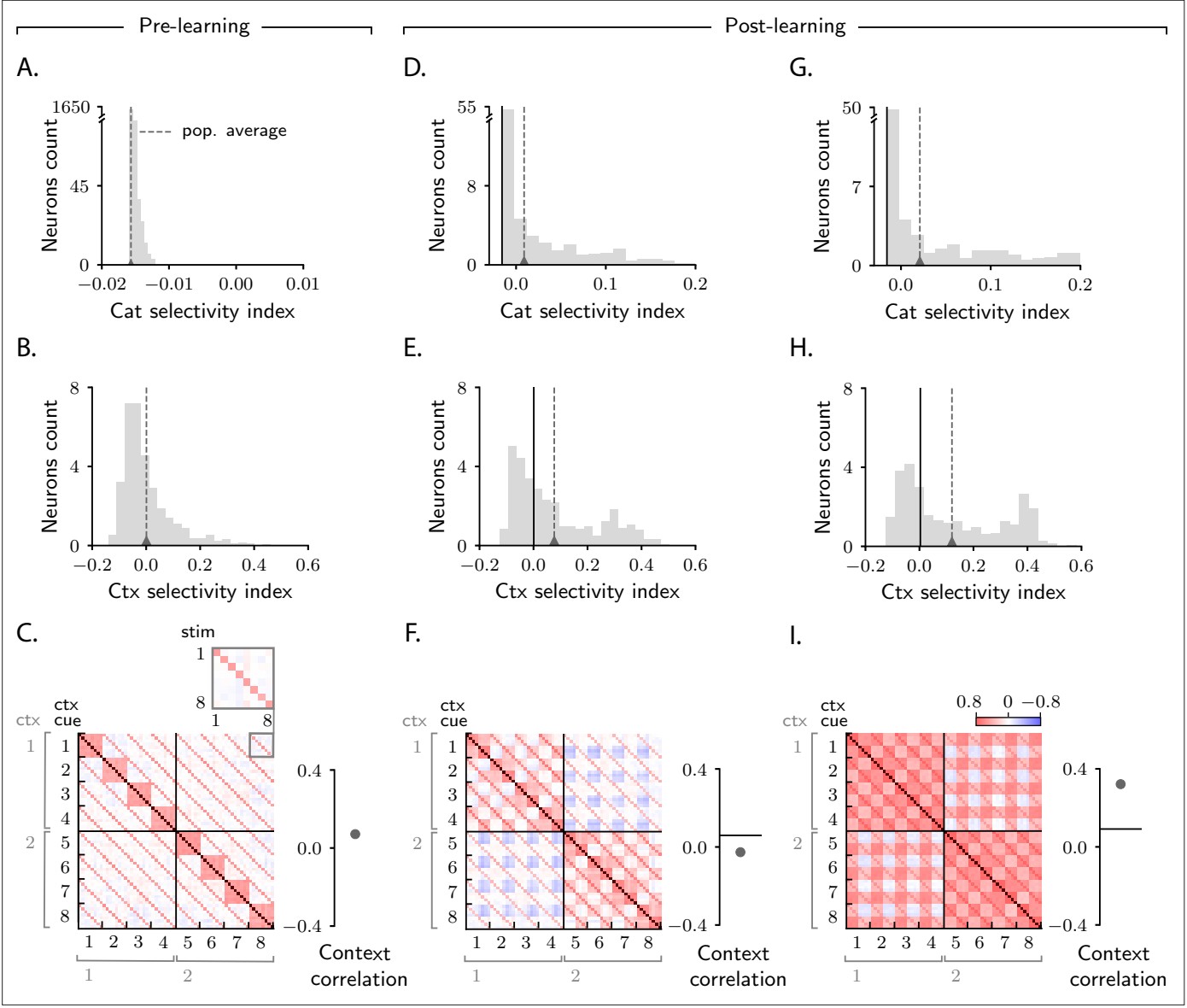

**Figure 6.** Characterization of activity evolution during the context-dependent categorization task. Results from simulations. The first column (**A–C**) shows a naive circuit (pre-learning); the second (**D–F**) and third (**G–I**) columns show two trained circuits (post-learning), characterized by different sets of parameters. (**A, D, G**) Histogram of category selectivity (*Equation 2*) across the population of neurons in the intermediate layer (note that the vertical axis has been expanded for visualization purposes). Grey dashed lines indicate the average selectivity across the population. In panels D and G, the black vertical lines indicate the initial value of the average selectivity. Note that the distribution of category selectivity is different from the distribution observed in the simple task (*Figure 2F, J*); the distribution is now heavy tailed, with only a fraction of the neurons acquiring strong category selectivity (see also *Figure 8B*). (**B, E, H**) Histogram of context selectivity (Materials and methods Context-dependent task: category and context selectivity, *Equation 122*), details as in A, D, and G. (**C, F, I**) Correlation matrices. Each entry shows the Pearson correlation coefficient between activity from different trials. There are 8 stimuli and 8 context cues, for a total of 64 trials (i.e., 64 stimulus/context cue combinations). Diagonal entries (brown) are all equal to 1. The inset on the top of panel C shows, as an example, a magnified view of correlations among trials with context cues 1 and 8, across all stimuli (1–8). To the right of the matrices we show the context correlation, defined to be the average of the correlations within the off-diagonal blocks (trials in different contexts). In panels F and I, the black horizontal lines indicate the initial value of context correlation. Parameters are summarized in *Table 1* (Materials and methods Tables of parameters).

The online version of this article includes the following figure supplement(s) for figure 6:

**Figure supplement 1.** Characterization of activity evolution during the context-dependent categorization task; additional results, part I.

**Figure supplement 2.** Characterization of activity evolution during the context-dependent categorization task; additional results, part II.

**Figure supplement 3.** Characterization of activity evolution during the context-dependent categorization task; additional results, part III.

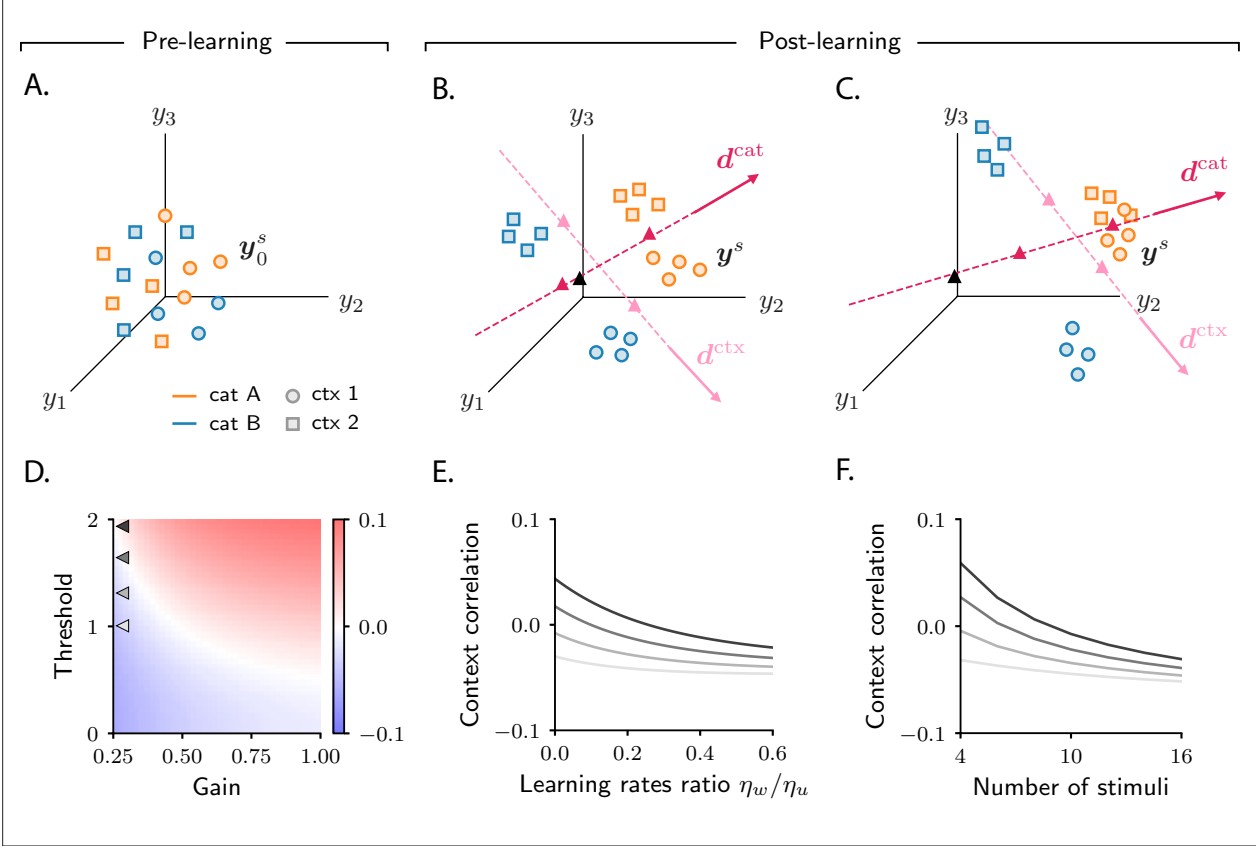

**Figure 7.** Analysis of activity evolution during the context-dependent categorization task. Results from mathematical analysis. (**A–C**) Cartoons illustrating how activity evolves over learning. Orange and blue symbols are associated with categories A and B, respectively; circles and squares are associated with contexts 1 and 2. Before learning, activity is mostly unstructured (panel A). After learning, activity forms four clouds, one for each combination of category and context. The center of the activity vectors associated with categories A and B and contexts 1 and 2 are indicated, respectively, by magenta and pink triangles. The black triangle indicates the center of initial activity. The cartoons in panels A–B–C refer to the three circuits illustrated in the three columns of **Figure 6**; for illustration purposes, we show a reduced number of stimuli and context cues (4 instead of 8). Simulated data from the circuits displayed in **Figure 6** are shown in **Figure 6—figure supplement 1D**. (**D**) Change in context correlation over learning as a function of the threshold and gain of the readout neuron. Grey arrows indicate the threshold and gain that are used in panels E and F. (**E**) Change in context correlation over learning as a function of the ratio of learning rates in the two layers. (**F**) Change in context correlation over learning as a function of the number of stimuli. Correlations in panels D–F were computed from the approximate theoretical expression given in Materials and methods Context-dependent task: category and context correlation. Parameters are given in **Table 1** (Materials and methods Tables of parameters).

The online version of this article includes the following figure supplement(s) for figure 7:

**Figure supplement 1.** Comparison between finite-size networks and approximate mathematical description for the context-dependent categorization task.

*Figure 7A* shows, schematically, activity before learning (see *Figure 6—figure supplement 1D* for simulated data). Each point represents activity on a given trial, and is associated with a category (A, orange; B, blue) and a context (1, circles; 2, squares). Before learning, activity is mostly unstructured (*Figure 7A*, Materials and methods Detailed analysis of category selectivity); over learning, though, it acquires structure (*Figure 7B, C*). As in the simple task (*Figure 3B, C*), activity vectors get re-arranged into statistically distinguishable clouds. While in the simple task clouds were determined by category, here each cloud is associated with a combination of category and context. As a result, four clouds are formed: the cloud of orange circles corresponds to category A and context 1; orange squares to category A and context 2; blue circles to category B and context 1; and blue squares to category B and context 2.

The transition from unstructured activity (*Figure 7A*) to four clouds of activity (*Figure 7B, C*) occurs by learning-induced movement along two directions: $d^{cat}$, which corresponds to category, and $d^{ctx}$, which corresponds to context. Activity vectors in different categories move by different amounts along

$d^{\mathrm{cat}}$; this causes the orange and blue symbols in *Figure 7B, C* to move apart, so that activity vectors associated with the same category become closer than vectors associated with opposite categories. As in the simple task, this in turn causes the category selectivity to increase, as shown in *Figure 6D, G* (Materials and methods Detailed analysis of category selectivity). Similar learning dynamics occurs for context: activity vectors from different contexts move by different amounts along $d^{\mathrm{ctx}}$. This causes the squares and circles in *Figure 7B, C* to move apart, so that activity vectors from the same context become closer than vectors from different contexts. Again, this in turn causes the context selectivity to increase, as shown in *Figure 6E, H* (Materials and methods Detailed analysis of context selectivity). Mathematical analysis indicates that the increase in clustering by category and context is independent of model parameters (*Figure 6—figure supplement 1B*), which explains the robustness of the increase in selectivity observed in simulations.

The category- and the context-related structures that emerge in *Figure 7B, C* have different origins and different significance. The emergence of category-related structure is, perhaps, not surprising: over learning, the activity of the readout neuron becomes category dependent, as required by the task; such dependence is then directly inherited by the intermediate layer, where activity clusters by category. This structure was already observed in the simple categorization task (*Figure 3B, C*). The emergence of context-related structure is, on the other hand, more surprising, since the activity of the readout neuron does not depend on context. Nevertheless, context-dependent structure, in the form of clustering, emerges in the intermediate layer activity. Such novel structure is a signature of the gradient-descent learning rule used by the circuit (*Canatar et al., 2021*). The mechanism through which context clustering emerges is described in detail in Materials and methods Detailed analysis of context selectivity. But, roughly speaking, context clustering emerges because, for a pair of sensory inputs, how similarly their intermediate-layer representations evolve during learning is determined both by their target category and their correlations (*Equation 27*, Materials and methods Evolution of connectivity and activity in large circuits). In the simple task, initial correlations were virtually nonexistent (*Figure 2C*), and thus activity changes were specified only by category; in the context-dependent task, initial correlations have structure (*Figure 6C*), and that structure critically affects neural representations. In particular, inputs with the same context tend to be relatively correlated, and those are also likely to be associated with the same category; their representations are thus clustered by the learning algorithm, resulting in context clustering.

While the clustering by category and context described in *Figure 7B, C* is robust across circuits, the position of clouds in the activity space is not. As in the simple task, the variability in cloud position explains the variability in context correlation (although the relationship between clouds position and correlations is more complex in this case, see Materials and methods Context-dependent categorization task). In *Figure 7D–F*, we show how context correlation depends on model parameters. This dependence is qualitatively similar to that of category correlation in the simple task: context correlation depends on the threshold and gain of neurons (compare *Figure 7D* and *Figure 4A*), on the relative learning rate $\eta_w/\eta_u$ (compare *Figure 7E* and *Figure 4C*), and on the number of stimuli (compare *Figure 7F* and *Figure 4D*). However, we find that the region of parameter space leading to an increase in correlation shrinks substantially compared to the simple task (*Figure 6—figure supplement 2C*, see also Materials and methods Context-dependent task: computing activity); this is in line with the observation that correlations decrease to negative values when the complexity of the task increases, as shown in *Figure 4D*.

## Patterns of pure and mixed selectivity are shaped by initial activity

As a final step, we take a closer look at single-neuron selectivity. Analysis from the previous sections indicates that the average selectivity to both category and context increases over learning. And, as in the simple task, the increase is highly variable across neurons (*Figure 6D, E and G, H*). To determine which neurons become the most selective to category and context, we analyze the directions along which clustering to category and context occurs, $d^{\mathrm{cat}}$ and $d^{\mathrm{ctx}}$ (*Figure 7B, C*). In analogy with the simple task, neurons that strongly increase selectivity to category are characterized by a large component along the category direction $d^{\mathrm{cat}}$; similarly, neurons that strongly increase selectivity to context are characterized by a large component along the context direction $d^{\mathrm{ctx}}$ (*Figure 6—figure supplement 3A, B*).

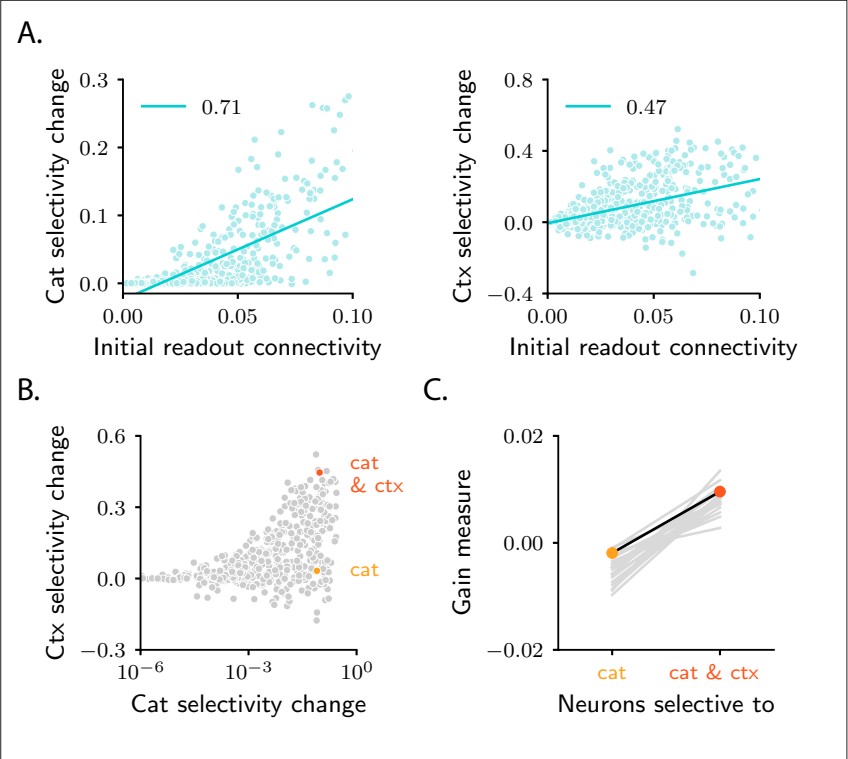

**Figure 8.** Patterns of pure and mixed selectivity to category and context. (**A**) Changes in category selectivity (left) and context selectivity (right) as a function of the initial readout connectivity, $w_{0,i}$ (in absolute value). Details as in **Figure 5B, C**. (**B**) Changes in context selectivity as a function of changes in category selectivity. Note the logarithmic scale on the $x$-axis; this is required by the heavy-tailed behaviour of category selectivity (**Figure 6D, G**). We highlighted two sample neurons: one with strong, pure selectivity to category (yellow) and one with strong, mixed selectivity to category and context (orange). (**C**) Neurons that develop pure and mixed selectivity are characterized by different patterns of initial activity. Here, we plot the gain-based measure of activity defined in **Equation 183** for neurons that belong to the former (left), and the latter (right) group. The former group includes neurons for which the change in category selectivity, but not the change in context selectivity, is within the top 15% across the population. The latter group includes neurons for which the change in both category and context selectivity is within the top 15%. Dots show results for the circuit analyzed in panels A and B. Grey lines show results for 20 different circuit realizations; note that the slope is positive for all circuits. All panels in the figure show results for the circuit displayed in the second column of **Figure 6**; the circuit displayed in the third column yields qualitatively similar results (**Figure 6—figure supplement 3C, D**).

Analysis in Materials and methods Analysis of patterns of context and category selectivity shows that both the category and context directions, $\boldsymbol{d}^{\text{cat}}$ and $\boldsymbol{d}^{\text{ctx}}$, are strongly correlated with the initial readout vector $\boldsymbol{w}_0$. As in the simple task, this leads to the prediction that neurons that strongly increase selectivity to either category or context are, before learning, strongly connected to the downstream readout neuron (**Figure 8A**).

Although $\boldsymbol{d}^{\text{cat}}$ and $\boldsymbol{d}^{\text{ctx}}$ are both correlated with $\boldsymbol{w}_0$, they are not perfectly aligned (Materials and methods Analysis of patterns of context and category selectivity). In principle, then, for a given neuron (here, neuron $i$), both $d_i^{\text{cat}}$ and $d_i^{\text{ctx}}$ could be large (implying mixed selectivity to both abstract variables, category and context), or only one could be large (implying pure selectivity to only one abstract variable, category or context), or both could be small (implying no selectivity at all). While all combinations are possible in principle, in the model they do not all occur. In **Figure 8B**, we plot changes in context selectivity as a function of changes in category selectivity. We observe that, among all the neurons that strongly increase their selectivity, some increase selectivity to both category and context (orange sample neuron), and others increase selectivity to category, but not context (yellow sample neuron). In contrast, none increases selectivity to context but not category. This makes the following experimental prediction: among all the neurons that are strongly connected to the readout, neurons with pure selectivity to category and neurons with mixed selectivity to category and context should

be observed, but neurons with pure selectivity to context should not. The asymmetry between category and context arises because, in the model, the readout neuron learns to read out category, but not context. We show in *Figure 6—figure supplement 3E, F* that if a second readout neuron, which learns to read out context, is included in the circuit, neurons with strong pure selectivity to context are also observed.

What determines whether a given neuron develops pure selectivity to category, or mixed selectivity to category and context? Analysis reported in Materials and methods Analysis of patterns of context and category selectivity indicates that these two populations are characterized by different properties of the initial activity. In particular, the two populations are characterized by different initial patterns of the response gain (defined as the slope of the activation function, *Figure 1C*, at the response), which measures the local sensitivity of neurons to their inputs. The exact patterns that the response gain takes across the two populations is described in detail in Materials and methods Analysis of patterns of context and category selectivity (*Equation 183*); the fact that pure- and mixed-selective neurons can be distinguished based on these patterns is illustrated in *Figure 8C*. Overall, these results indicate that initial activity, which is mostly unstructured and task-agnostic, plays an important role in learning: it breaks the symmetry among neurons in the intermediate layer, and determines which functional specialization neurons will display over learning.

## Discussion

How does the brain learn to link sensory stimuli to abstract variables? Despite decades of experimental (*Asaad et al., 1998*; *Messinger et al., 2001*; *Freedman and Assad, 2006*; *Reinert et al., 2021*) and theoretical (*Rosenblatt, 1958*; *Barak et al., 2013*; *Engel et al., 2015*) work, the answer to this question remains elusive. Here, we hypothesized that learning occurs via gradient-descent synaptic plasticity. To explore the implications of this hypothesis, we considered a minimal model: a feedforward circuit with one intermediate layer, assumed to contain a large number of neurons compared to the number of stimuli. This assumption allowed us to thoroughly analyze the model, and thus gain insight into how activity evolves during learning, and how that evolution supports task acquisition.

We focused on two categorization tasks: a simple one (*Figure 2*), in which category was determined solely by the stimulus, and a complex one (*Figure 6*), in which category was determined by both the stimulus and the context. We showed that, over learning, single neurons become selective to abstract variables: category (which is explicitly reinforced) and context (which is not; instead, it embodies the task structure, and is only implicitly cued). From a geometrical perspective, the emergence of selectivity during learning is driven by clustering: activity associated with stimuli in different categories is pushed apart, forming distinct clusters (*Figure 3*). In the context-dependent task, additional clustering occurs along a second, context-related axis; this results in activity forming four different clouds, one for each combination of category and context (*Figure 7*). While the behaviour of selectivity is highly stereotyped, the behaviour of signal correlations and tuning symmetry is not, but depends on details (*Figure 4*). From a geometrical perspective, the variability in correlations and symmetry is due to the variability in the position of category and context clusters with respect to initial activity.

Our work was motivated partly by the observation that responses to different categories in monkeys area LIP were positively correlated and asymmetric (*Fitzgerald et al., 2013*) – a finding that seems at odds with experimental observations in sensory, and other associative, brain areas (*Cromer et al., 2010*; *Reinert et al., 2021*). It has been suggested that those responses arise as a result of learning that drives activity in area LIP onto an approximately one-dimensional manifold (*Ganguli et al., 2008*; *Fitzgerald et al., 2013*; *Summerfield et al., 2020*). Our results are broadly in line with this hypothesis: for the simple categorization task, which is similar to *Fitzgerald et al., 2013*, we showed that activity stretches along a single direction (*Equation 3*, *Figure 3C*). Analysis in Materials and methods Evolution of activity further shows that not only at the end of learning, but at every learning epoch, activity is aligned along a single direction; the whole learning dynamics is thus approximately one-dimensional. However, in the context-dependent categorization task, activity stretches along two dimensions (*Figure 7B, C*), indicating that one dimension does not always capture activity.

Our analysis makes several experimental predictions. First, it makes specific predictions about how category and context correlations should vary with properties of the circuit (threshold and gain of neurons, relative learning rates) and with the task (number of stimuli, context dependence) (*Figure 4*). These could be tested with current technology; in particular, testing the dependence on task variables

only requires recording neural activity. Second, it predicts that selectivity is shaped by connectivity with downstream areas, a result that is in line with recent experimental observations (*Glickfeld et al., 2013*; *Ye et al., 2016*; *Hirokawa et al., 2019*; *Gschwend et al., 2021*). More specifically, it predicts that, for a given neuron, selectivity correlates with the strength of the synaptic connection that the neuron makes to the downstream neurons that read out category (*Figure 5B, C* and *Figure 8A*). Across all neurons that are strongly connected to downstream readout neurons, selectivity to category and context is distributed in a highly stereotyped way: during learning, some neurons develop mixed selective to category and context, others develop pure selectivity to category, but none develop pure selectivity to context (*Figure 8B*). Moreover, whether a neuron develops mixed or pure selectivity depends on initial activity (*Figure 8C*).

## Previous models for categorization

Previous theoretical studies have investigated how categorization can be implemented in multi-layer neural circuits (*Barak et al., 2013*; *Babadi and Sompolinsky, 2014*; *Litwin-Kumar et al., 2017*; *Pannunzi et al., 2012*; *Engel et al., 2015*; *Villagrasa et al., 2018*; *Min et al., 2020*). Several of those studies considered a circuit model in which the intermediate connectivity matrix, $u$, is fixed, and only the readout vector, $w$, evolves (via Hebbian plasticity) over learning (*Barak et al., 2013*; *Babadi and Sompolinsky, 2014*; *Litwin-Kumar et al., 2017*). This model can learn both the simple (linearly separable) and complex (nonlinearly separable) tasks (*Barak et al., 2013*). Because there is no learning in the intermediate connectivity, activity in the intermediate layer remains unstructured, and high dimensional, throughout learning. This stands in sharp contrast to our model, where learning leads to structure in the form of clustering – and, thus, a reduction in activity dimensionality.

One study did consider a model in which both the intermediate and the readout connectivity evolve over learning, according to reward-modulated Hebbian plasticity (*Engel et al., 2015*). This circuit could learn a simple categorization task but, in contrast to our study, learning did not lead to significant changes in the activity of the intermediate layer. When feedback connectivity was introduced, learning did lead to activity changes in the intermediate layer, and those activity changes led to an increase in category selectivity – a finding that is in line with ours. There were, however, several notable differences relative to our model. First, learning of the intermediate and readout weights occurred on separate timescales: the intermediate connectivity only started to significantly change when the readout connectivity was completely rewired; in our model, in contrast, the two set of weights evolve on similar timescales. Second, population responses were negatively correlated and symmetric; whether positively correlated and asymmetric responses (as seen in experiments, *Fitzgerald et al., 2013*, and in our model) can also be achieved remains to be established. Third, context-dependent associations, that are core to a variety of experimental tasks (*Wallis et al., 2001*; *Roy et al., 2010*; *McKenzie et al., 2014*; *Brincat et al., 2018*; *Reinert et al., 2021*; *Mante et al., 2013*), were not considered. Whether reward-modulated Hebbian plasticity can be used to learn context-dependent tasks is unclear, and represents an important avenue for future work.

## Gradient-descent learning in the brain

A common feature of the studies described above is that learning is implemented via Hebbian synaptic plasticity – a form of plasticity that is known to occur in the brain. Our model, on the other hand, uses gradient-descent learning in a multi-layer network, which requires back-propagation of an error signal; whether and how such learning is implemented in the brain is an open question (*Whittington and Bogacz, 2019*). A number of recent theoretical studies have proposed biologically plausible architectures and plasticity rules that can approximate back-propagation on simple and complex tasks (*Lillicrap et al., 2016*; *Sacramento et al., 2018*; *Akrout et al., 2019*; *Whittington and Bogacz, 2017*; *Payeur et al., 2021*; *Pogodin and Latham, 2020*; *Boopathy and Fiete, 2022*). Understanding whether these different implementations lead to differences in activity represents a very important direction for future research. Interestingly, recent work has showed that it is possible to design circuit models where the learning dynamics is identical to the one studied in this work, but the architecture is biologically plausible (*Boopathy and Fiete, 2022*). We expect our results to directly translate to those models. Other biologically plausible setups might be characterized, instead, by different activity evolution. Recent work (*Song et al., 2021*; *Bordelon and Pehlevan, 2022*) made use of a formalism similar to ours to describe learning dynamics induced

by a number of different biologically plausible algorithms and uncovered non-trivial, qualitatively different dynamics. Whether any of these dynamics leads to different neural representations in neuroscience-inspired categorization tasks like the ones we studied here is an open, and compelling, question.

In this work, we used mathematical analysis to characterize the activity changes that emerge during gradient-descent learning. Our analysis relied on two assumptions. First, the number of neurons in the circuit is large compared to the number of stimuli to classify. Second, the synaptic weights are chosen so that the initial activity in all layers of the network lies within an intermediate range (i.e., it neither vanishes nor saturates) before learning starts (*Jacot et al., 2018*; *Chizat et al., 2019*; *Lee et al., 2019*; *Liu et al., 2020*). These two assumptions are reasonable for brain circuits, across time scales ranging from development to animals' lifetimes; a discussion on the limitations of our approach is given in Materials and methods Evolution of activity in finite-size networks.

A prominent feature of learning under these assumptions is that changes in both the synaptic weights and activity are small in amplitude (Materials and methods Evolution of connectivity and activity in large circuits). This has an important implication: the final configuration of the circuit depends strongly on the initial one. We have showed, for example, that the selectivity properties that single neurons display at the end of learning are determined by their initial activity and connectivity (*Figure 5B, C* and *Figure 8A, C*). Moreover, the final distribution of selectivity indices, and the final patterns of correlations, bear some resemblance to the initial ones (see, e.g., *Figure 6*); for this reason, we characterized activity evolution via changes in activity measures, rather than their asymptotic, post-learning values. Overall, these findings stress the importance of recording activity throughout the learning process to correctly interpret neural data (*Steinmetz et al., 2021*; *Latimer and Freedman, 2021*).

Circuits that violate either of the two assumptions discussed above may exhibit different gradient-descent learning dynamics than we saw in our model (*Chizat et al., 2019*), and could result in different activity patterns over learning. Previous studies have analyzed circuits with linear activation functions and weak connectivity (weak enough that activity is greatly attenuated as it passes through the network). However, linear activation functions can only implement a restricted set of tasks (*Saxe et al., 2019*; *Li and Sompolinsky, 2021*; *Moroshko et al., 2020*; in particular, they cannot implement the context-dependent task we considered). Developing tools to analyze arbitrary circuits will prove critical to achieving a general understanding of how learning unfolds in the brain (*Mei et al., 2018*; *Yang and Hu, 2021*; *Flesch et al., 2022*).

## Beyond simplified models

Throughout this work, we focussed on two simplified categorization tasks, aimed at capturing the fundamental features of the categorization tasks commonly used in systems neuroscience (*Freedman and Assad, 2006*; *Fitzgerald et al., 2011*; *Wallis et al., 2001*). The mathematical framework we developed to analyze those tasks could, however, easily be extended in several directions, including tasks with more than two categories (*Fitzgerald et al., 2011*; *Reinert et al., 2021*; *Mante et al., 2013*) and tasks involving generalization to unseen stimuli (*Barak et al., 2013*; *Canatar et al., 2021*). An important feature missing in our tasks, though, is memory: neuroscience tasks usually involve a delay period during which the representation of the output category must be sustained in the absence of sensory inputs (*Freedman and Assad, 2006*; *Fitzgerald et al., 2011*; *Wallis et al., 2001*). Experiments indicate that category representations are different in the stimulus presentation and the delay periods (*Freedman and Assad, 2006*). Investigating these effects in our tasks would require the addition of recurrent connectivity to the model. Mathematical tools for analyzing learning dynamics in recurrent networks is starting to become available (*Mastrogiuseppe and Ostojic, 2019*; *Schuessler et al., 2020*; *Dubreuil et al., 2022*; *Susman et al., 2021*), which could allow our analysis to be extended in that direction.

To model categorization, we assumed a quadratic function for the error $\mathcal{E}$ (Materials and methods Circuit) – an assumption that effectively casts our categorization tasks into a regression problem. This made the model amenable to mathematical analysis, and allowed us to derive transparent equations to characterize activity evolution. Recent machine-learning work has showed that, at least in some categorization setups (*Hui and Belkin, 2021*), a cross-entropy function might result in better learning performance. The mathematical framework used here is, however, not well suited to studying networks with such an error function (*Lee et al., 2019*). Investigating whether and how our findings

extend to networks trained with a cross-entropy error function represents an interesting direction for future work.

Finally, in this study we focussed on a circuit model with a single intermediate layer. In the brain, in contrast, sensory inputs are processed across a number of stages within the cortical hierarchy. Our analysis could easily be extended to include multiple intermediate layers. That would allow our predictions to be extended to experiments involving multi-area recordings, which are increasingly common in the field (*Goltstein et al., 2021*). Current recording techniques, furthermore, allow monitoring neural activity throughout the learning process (*Reinert et al., 2021*; *Goltstein et al., 2021*); those data could be used in future studies to further test the applicability of our model.

### Bridging connectivity and selectivity

In this work, we considered a circuit with a single readout neuron, trained to discriminate between two categories. One readout neuron is sufficient because, in the tasks we considered, categories are mutually exclusive (*Fitzgerald et al., 2013*). We have found that the initial readout weights play a key role in determining the directions of activity evolution, suggesting that circuits with different or additional readout neurons might lead to different activity configurations. For example, one might consider a circuit with two readout neurons, each highly active in response to a different category. And indeed, recent work in mouse PFC suggests that two readout circuits are used for valence – one strongly active for positive valence, and one strongly active for negative one (*Ye et al., 2016*). Also, in context-dependent tasks, one might consider a circuit with an additional readout for context. We have showed in *Figure 6—figure supplement 3E, F* that this model leads to different experimental predictions for selectivity than the model with only one readout for category (*Figure 8B*). Altogether, these observations indicate that functional properties of neurons are tightly linked to their long-range projections – a pattern that strongly resonates with recent experimental findings (*Hirokawa et al., 2019*; *Yang et al., 2022*). Constraining model architectures with connectomics, and then using models to interpret neural recordings, represents a promising line of future research.

## Materials and methods

### Overview

In the main text, we made qualitative arguments about the evolution of activity over learning. Here, we make those arguments quantitative. We start with a detailed description of the circuit model (Section Model). We then derive approximate analytical expressions that describe how activity in the circuit evolves over learning (Section Evolution of connectivity and activity in large circuits). To this end, we use an approach that is valid for large circuits. We apply this approach first to the simple task (Section Simple categorization task), then to the context-dependent one (Section Context-dependent categorization task). Finally, we provide details on the numerical implementation of circuit models and analytical expressions (Section Software).

### Model

#### Circuit

We consider a feedforward circuit with a single intermediate layer (*Figure 1B*). For simplicity, we assume that the input and the intermediate layer have identical size $N$, and we consider $N$ to be large. The sensory input vector is indicated with $\boldsymbol{x}$. Activity in the intermediate layer reads

$$\boldsymbol{y} = \Psi(\boldsymbol{k}) \tag{5a}$$

$$\boldsymbol{k} \equiv \boldsymbol{u} \cdot \boldsymbol{x}. \tag{5b}$$

Here, $\boldsymbol{k}$ represents the synaptic drive and $\boldsymbol{u}$ is an $N \times N$ connectivity matrix. Activity in the readout layer is given by

$$z = \Phi(h) \tag{6a}$$

$$h \equiv \boldsymbol{w} \cdot \boldsymbol{y} \tag{6b}$$

where $h$ is the synaptic drive and $\boldsymbol{w}$ is an $N$-dimensional readout vector.

The activation functions $\Psi$ and $\Phi$ are non-negative, monotonically increasing functions that model the input-to-output properties of units in the intermediate and readout layer, respectively. In simulations, we use sigmoidal functions,

$$\Psi(x) = \frac{1}{1+\exp(-\Theta_1(x-\Theta_2))}, \tag{7}$$

and similarly for $\Phi(x)$ (*Figure 1C*). The parameters of the activation functions, $\Theta_1$ and $\Theta_2$, determine the gain and threshold, respectively, with the gain (defined to be the slope at $x = \Theta_2$) given by $\Theta_1/4$. Their values, which vary across simulations, are given in Section Tables of parameters.

The synaptic weights, $u$ and $w$, are initialized at random from a zero-mean Gaussian distribution with variance $1/N$. The sensory input vectors $x$ are also drawn from a zero-mean Gaussian distribution (see Sections Simple task: task definition and Context-dependent task: task definition), but with variance equal to 1,

$$w_{0,i}, u_{0,ij} \sim \mathcal{N}(0, N^{-1}) \tag{8a}$$

$$x_i \sim \mathcal{N}(0, 1) \tag{8b}$$

where the subscript '0' on the weights indicates that those are evaluated before learning starts. This choice of initialization ensures that, before learning, the amplitude of both the synaptic drive ($h$, and the components of $k$) and the activity ($z$, and the components of $y$) are independent of the circuit size (i.e., $O(1)$ in $N$).

## Gradient-descent plasticity

The circuit learns to categorize $P$ sensory input vectors $x^s$ ($s = 1, \ldots, P$), with $P \ll N$. For each input vector, the target activity of the readout neuron, $\tilde{z}^s$, is equal to either $z^A$ or $z^B$ (Sections Simple task: task definition and Context-dependent task: task definition), which correspond to high and low activity, respectively. The weights are adjusted to minimize the loss, $\mathcal{E}(u, w)$, which is defined to be

$$\mathcal{E}(u, w) \equiv \frac{1}{2P} \sum_{s=1}^{P} (\tilde{z}^s - z^s)^2 \tag{9}$$

where $z^s$ is the activity of the readout neuron (*Equation 6a*) in response to the sensory input $x^s$. The weights are updated according to full-batch vanilla gradient descent. If the learning rates, $\eta_u$ and $\eta_w$, are sufficiently small, the evolution of the connectivity weights can be described by the continuous-time equations (*Equation 1a*, *Equation 1b*)

$$\frac{du}{dt} = -\eta_u \frac{\partial \mathcal{E}(u, w)}{\partial u} \tag{10a}$$

$$\frac{dw}{dt} = -\eta_w \frac{\partial \mathcal{E}(u, w)}{\partial w} \tag{10b}$$

where $t$ indicates learning time.

## Evolution of connectivity and activity in large circuits

Our goal is to understand how learning affects activity in the intermediate layer, $y$. We do that in two steps. In the first step, we analyze the evolution of the synaptic weights. In particular, we determine the weights after learning is complete – meaning after the loss (*Equation 9*) has been minimized (Section Evolution of connectivity). In the second step, we use the learned weights to determine activity (Section Evolution of activity). We work in the large-$N$ regime, which allows us to make analytic headway (*Jacot et al., 2018*; *Lee et al., 2019*; *Liu et al., 2020*). We then validate our large-$N$ analysis with finite-$N$ simulations (Section Evolution of activity in finite-size networks, *Figure 3—figure supplement 1*, *Figure 3—figure supplement 2*, *Figure 6—figure supplement 1*, *Figure 7—figure supplement 1*).

### Evolution of connectivity

It is convenient to make the definitions

$$\boldsymbol{u} \equiv \boldsymbol{u}_0 + \Delta \boldsymbol{u} \tag{11a}$$

$$\boldsymbol{w} \equiv \boldsymbol{w}_0 + \Delta \boldsymbol{w} \tag{11b}$$

where $\boldsymbol{u}_0$ and $\boldsymbol{w}_0$ are the initial weights (*Equation 8a*), and $\Delta \boldsymbol{u}$ and $\Delta \boldsymbol{w}$ are changes in the weights induced by learning (*Equation 10*). Using *Equation 10*, with the loss given by *Equation 9*, we see that $\Delta \boldsymbol{u}$ and $\Delta \boldsymbol{w}$ evolve according to

$$\frac{\mathrm{d}\Delta \boldsymbol{u}(t)}{\mathrm{d}t} = -\eta_u \frac{\partial \mathcal{E}}{\partial \Delta \boldsymbol{u}} = \frac{\eta_u}{P} \sum_{s=1}^{P} \epsilon^s(t) \frac{\partial h^s}{\partial \Delta \boldsymbol{u}} \tag{12a}$$

$$\frac{\mathrm{d}\Delta \boldsymbol{w}(t)}{\mathrm{d}t} = -\eta_w \frac{\partial \mathcal{E}}{\partial \Delta \boldsymbol{w}} = \frac{\eta_w}{P} \sum_{s=1}^{P} \epsilon^s(t) \frac{\partial h^s}{\partial \Delta \boldsymbol{w}} \tag{12b}$$

where $\epsilon^s$ is proportional to the error associated with sensory input $\boldsymbol{x}^s$,

$$\epsilon^s \equiv \left( \tilde{z}^s - \Phi(h^s) \right) \Phi'(h^s). \tag{13}$$

To evaluate the partial derivatives on the right-hand side of *Equation 12*, we need to express $h^s$ in terms of $\Delta \boldsymbol{u}$ and $\Delta \boldsymbol{w}$. Combining *Equation 6b* with *Equation 5* and *Equation 11*, we have

$$h^s = (\boldsymbol{w}_0 + \Delta \boldsymbol{w}) \cdot \Psi(\boldsymbol{u}_0 \cdot \boldsymbol{x}^s + \Delta \boldsymbol{u} \cdot \boldsymbol{x}^s). \tag{14}$$

To proceed, we assume that changes in the connectivity, $\Delta \boldsymbol{u}$ and $\Delta \boldsymbol{w}$, are small. That holds in the large-$N$ limit (the limit we consider here) because when each neuron receives a large number of inputs, none of them has to change very much to cause a large change in the output (we make this reasoning more quantitative in Section A low-order Taylor expansion is self-consistent in large circuits). Then, Taylor-expanding the nonlinear activation function $\Psi$ in *Equation 14*, and keeping only terms that are zeroth and first order in the weight changes $\Delta \boldsymbol{u}$ and $\Delta \boldsymbol{w}$, we have

$$h^s \simeq h_0^s + \boldsymbol{w}_0 \cdot \left[ \Psi'(\boldsymbol{k}_0^s) \odot (\Delta \boldsymbol{u} \cdot \boldsymbol{x}^s) \right] + \Delta \boldsymbol{w} \cdot \Psi(\boldsymbol{k}_0^s), \tag{15}$$

where $\odot$ indicates element-wise multiplication, and we have defined

$$\boldsymbol{k}_0^s \equiv \boldsymbol{u}_0 \cdot \boldsymbol{x}^s \tag{16a}$$

$$h_0^s \equiv \boldsymbol{w}_0 \cdot \Psi(\boldsymbol{k}_0^s). \tag{16b}$$

For now, we assume that the three terms in the right-hand side of *Equation 15* are of similar magnitude, and that higher-order terms in $\Delta \boldsymbol{u}$ and $\Delta \boldsymbol{w}$ are smaller, and so can be neglected. We will verify these assumptions post hoc (Section A low-order Taylor expansion is self-consistent in large circuits). Inserting *Equation 15* into *Equation 12*, we arrive at

$$\frac{\mathrm{d}\Delta \boldsymbol{u}(t)}{\mathrm{d}t} = \frac{\eta_u}{P} \sum_{s=1}^{P} \epsilon^s(t) \left[ \boldsymbol{w}_0 \odot \Psi'(\boldsymbol{k}_0^s) \right] \boldsymbol{x}^s \tag{17a}$$

$$\frac{\mathrm{d}\Delta \boldsymbol{w}(t)}{\mathrm{d}t} = \frac{\eta_w}{P} \sum_{s=1}^{P} \epsilon^s(t) \Psi(\boldsymbol{k}_0^s) \tag{17b}$$

(we used the notation where two adjacent vectors correspond to an outer product; i.e., $(\boldsymbol{ab})_{ij} = a_i b_j$).

The only quantity on the right-hand side of *Equation 17* that depends on time is $\epsilon^s$. Consequently, we can immediately write down the solution,

$$\Delta \boldsymbol{u}(t) = \frac{1}{N} \sum_{s=1}^{P} c^s(t) \left[ \boldsymbol{w}_0 \odot \Psi'(\boldsymbol{k}_0^s) \right] \boldsymbol{x}^s \tag{18a}$$

$$\Delta \boldsymbol{w}(t) = \frac{1}{N} \sum_{s=1}^{P} c^s(t) \frac{\eta_w}{\eta_u} \Psi(\boldsymbol{k}_0^s) \tag{18b}$$

where the coefficients $c^s$ are found by solving the differential equation

$$\frac{P}{\eta_u N} \frac{\mathrm{d}c^s(t)}{\mathrm{d}t} = \epsilon^s(t) \tag{19}$$

with initial conditions $c^s(t = 0) = 0$. The right-hand side of *Equation 19* depends on time through the synaptic drive, $h^s$ (*Equation 13*), which in turn depends on $\Delta u$ and $\Delta w$ through *Equation 15*, and thus, via *Equation 18*, on the coefficients $c^s(t)$. Consequently, *Equation 19* is a closed differential equation for the coefficients $c^s(t)$.

In the general case, *Equation 19* must be solved numerically. If, however, we are not interested in the full learning dynamics, but care only about connectivity and activity once learning has converged ($t \to \infty$), we can use the fact that dynamics in *Equation 17* are guaranteed to converge to a global minimum of the error function $\mathcal{E}$ (*Liu et al., 2020*). For our loss function and tasks, the minimum occurs at $\mathcal{E} = 0$. At that point, $z^s(t \to \infty) = \tilde{z}^s$; equivalently,

$$h^s(t \to \infty) = \Phi^{-1}(\tilde{z}^s), \tag{20}$$

where $\Phi^{-1}$ is the inverse of the activation function of the readout neurons (which exists because $\Phi$ is a monotonically increasing function).

To find $c^s(t \to \infty)$, we simply express $h^s(t \to \infty)$ in terms of $c^s(t \to \infty)$, and insert that into *Equation 20*. To reduce clutter, we define (in a slight abuse of notation) $c^s$ without an argument to be its asymptotic value,

$$c^s \equiv c^s(t \to \infty). \tag{21}$$

Combining *Equation 15* for $h^s$ with *Equation 18* for $\Delta w$ and $\Delta u$, we have

$$h^s(t \to \infty) = h_0^s + \boldsymbol{w}_0 \cdot \left( \frac{1}{N} \sum_{q=1}^{P} c^q (\boldsymbol{x}^q \cdot \boldsymbol{x}^s) \left[ \boldsymbol{w}_0 \odot \Psi'(\boldsymbol{k}_0^q) \odot \Psi'(\boldsymbol{k}_0^s) \right] \right) + \frac{\eta_w}{\eta_u} \frac{1}{N} \sum_{q=1}^{P} c^q \left( \Psi(\boldsymbol{k}_0^q) \cdot \Psi(\boldsymbol{k}_0^s) \right). \tag{22}$$

We can simplify the second term in the right-hand side by explicitly evaluating the dot product,

$$
\begin{aligned}
\boldsymbol{w}_0 \cdot \left[ \boldsymbol{w}_0 \odot \Psi'(\boldsymbol{k}_0^q) \odot \Psi'(\boldsymbol{k}_0^s) \right] &= \sum_{i=1}^{N} w_{0,i}^2 \Psi'(k_{0,i}^q) \Psi'(k_{0,i}^s) \\
&\equiv N \langle w_{0,i}^2 \Psi'(k_{0,i}^q) \Psi'(k_{0,i}^s) \rangle_i
\end{aligned}
\tag{23}
$$

where the notation $\langle . \rangle_i$ indicates an average over the index $i$.

Since $N$ is large, we can interpret population averages such as *Equation 23* as expectations over the probability distribution of $w_{0,i}$ and $k_{0,i}^s$. An immediate implication is that *Equation 23* simplifies,

$$
\begin{aligned}
N \langle w_{0,i}^2 \Psi'(k_{0,i}^q) \Psi'(k_{0,i}^s) \rangle_i &= N \langle w_{0,i}^2 \rangle_i \langle \Psi'(k_{0,i}^q) \Psi'(k_{0,i}^s) \rangle_i \\
&= \langle \Psi'(k_{0,i}^q) \Psi'(k_{0,i}^s) \rangle_i.
\end{aligned}
\tag{24}
$$

For the first equality we used the independence of $w_{0,i}$ and $k_{0,i}$; for the second we used the fact that the elements of $\boldsymbol{w}_0$ are drawn from a zero-mean Gaussian with variance $N^{-1}$ (*Equation 8a*). We can thus rewrite *Equation 22* as

$$h^s(t \to \infty) = h_0^s + \sum_{q=1}^{P} c^q \langle x_i^q x_i^s \rangle_i \langle \Psi'(k_{0,i}^q) \Psi'(k_{0,i}^s) \rangle_i + \frac{\eta_w}{\eta_u} \sum_{q=1}^{P} c^q \langle \Psi(k_{0,i}^q) \Psi(k_{0,i}^s) \rangle_i. \tag{25}$$

Combining this with *Equation 20*, we conclude that

$$\Phi^{-1}(\tilde{z}^s) - h_0^s = \sum_{q=1}^{P} \left[ \langle x_i^q x_i^s \rangle_i \langle \Psi'(k_{0,i}^q) \Psi'(k_{0,i}^s) \rangle_i + \frac{\eta_w}{\eta_u} \langle \Psi(k_{0,i}^q) \Psi(k_{0,i}^s) \rangle_i \right] c^q. \tag{26}$$

*Equation 26* is a $P$-dimensional linear system of equations for the coefficients $c^s$, $s = 1, \dots, P$ (the term in brackets is a $P \times P$ matrix with indices $s$ and $q$). For the tasks we consider (Sections Simple categorization task and Context-dependent categorization task), this system can be solved analytically, yielding a closed-form expression for the coefficients $c^s$.

## Evolution of activity

It is now straightforward to determine how activity in the intermediate layer, $y^s(t)$, evolves. Inserting *Equation 18* into *Equation 5*, and Taylor expanding the nonlinear activation function $\Psi$ to first order in $\Delta u$, we arrive at

$$y^s(t) \equiv y_0^s + \Delta y^s(t) \simeq y_0^s + \sum_{q=1}^{P} c^q(t) v^{qs} \tag{27}$$

where

$$v^{qs} \equiv \langle x_i^q x_i^s \rangle_i w_0 \odot \Psi'(k_0^q) \odot \Psi'(k_0^s). \tag{28}$$

To reduce clutter, we define (following the notation in the previous section) $y^s$ without an argument to be its asymptotic value: $y^s \equiv y^s(t \to \infty)$. Thus, *Equation 27* becomes

$$y^s \simeq y_0^s + \sum_{q=1}^{P} c^q v^{qs}. \tag{29}$$

Because of the term $w_0$ on the right-hand side of *Equation 28*, the elements of $v^{qs}$ scale as $N^{-1/2}$. Thus, changes in activity are small compared to the initial activity, which is $O(1)$.

In what follows, we refer to $\{v^{qs}\}_{qs}$ as spanning vectors, and to the coefficients $c^q$ as the activity coordinates. We observe that all spanning vectors have a non-zero overlap with the initial readout vector $w_0$, as

$$v^{qs} \cdot w_0 = \langle x_i^q x_i^s \rangle_i \langle \Psi'(k_{0,i}^q) \Psi'(k_{0,i}^s) \rangle_i \equiv \rho^{qs} \ . \tag{30}$$

This implies that, for every spanning vector, we can write

$$v^{qs} = \rho^{qs} w_0 + \delta v^{qs} \tag{31}$$

where $\rho^{qs}$ is given by *Equation 30* (since $w_0 \cdot w_0 = 1$) and $\delta v^{qs}$ is a residual component due to the nonlinearity of the activation function $\Psi$:

$$\delta v^{qs} = \langle x_i^q x_i^s \rangle_i w_0 \odot \left( \Psi'(k_0^q) \odot \Psi'(k_0^s) - \langle \Psi'(k_{0,i}^q) \Psi'(k_{0,i}^s) \rangle_i \mathbf{1} \right). \tag{32}$$

The notation $\mathbf{1}$ indicates a vector whose components are all equal to 1: $\mathbf{1} \equiv (1, 1, ..., 1)$.

## A low-order Taylor expansion is self-consistent in large circuits

To conclude our theoretical derivation, we verify that the approximations we made in Section Evolution of connectivity are valid in large circuits. Specifically, we show that the approximate expression for $h^s$, *Equation 15* (which was derived by Taylor expanding the nonlinear activation function $\Psi$), is self-consistent when $N$ is large. As a first step, we compute the size of $\Delta u$ and $\Delta w$, and show that in the large-$N$ limit they are small compared to $u_0$ and $w_0$, respectively. We then Taylor-expand $\Psi$ in *Equation 14* to all orders, and show that the terms that were included in *Equation 15* (zeroth- and first-order terms in connectivity changes) are indeed the dominant ones.

Assuming that the term in brackets in *Equation 26*, when viewed as a $P \times P$ matrix, is invertible (which is generically the case when $P \ll N$), it follows that, with respect to $N$

$$c^s \sim O(1). \tag{33}$$

This result applies to the asymptotic ($t \to \infty$) value of $c^s$ (*Equation 21*). We assume, though, that the learning process is smooth enough that $c^s(t)$ remains at most $O(1)$ for all $t$. Under this assumption, the results we derive in this section are valid at any point during learning.

Using *Equation 33*, along with the fact that $w_{0,i} \sim O(N^{-1/2})$ while all other variables are $O(1)$, we see from *Equation 18* that

$$\Delta u_{ij} \sim O(N^{-3/2}) \tag{34a}$$

$$\Delta w_i \sim O(N^{-1}). \tag{34b}$$

When $N$ is large, both are small compared to the initial weights $\boldsymbol{u}_0$ and $\boldsymbol{w}_0$, whose elements are $O(N^{-1/2})$ (**Equation 8**).

**Equation 34** suggests that a low-order Taylor expansion is self-consistent, but it is not proof. We thus turn directly to **Equation 14**. The three terms in the right-hand side of **Equation 15** are re-written in **Equation 25**, and it is clear from that expression that they are all $O(1)$. To determine the size of the higher-order terms, we need the complete Taylor expansion of **Equation 14**. That is given by

$$h^s = \sum_{n=0}^{\infty} \frac{1}{n!} (\boldsymbol{w}_0 + \Delta \boldsymbol{w}) \cdot \left( \Psi^{(n)}(\boldsymbol{k}_0^s) \odot (\Delta \boldsymbol{u} \cdot \boldsymbol{x}^s)^n \right) \tag{35}$$

where $\Psi^{(n)}$ is the $n^{th}$ derivative of $\Psi$, and the exponentiation in $(\Delta \boldsymbol{u} \cdot \boldsymbol{x}^s)^n$ is taken element-wise. The higher-order terms (i.e., the terms not included in **Equation 15**) are

$$h^s_{\text{higher order}} = \sum_{n=2}^{\infty} \frac{N}{n!} \langle w_{0,i} \Psi^{(n)}(k_{0,i}^s)(\Delta \boldsymbol{u} \cdot \boldsymbol{x}^s)_i^n \rangle_i + \sum_{n=1}^{\infty} \frac{N}{n!} \langle \Delta w_i \Psi^{(n)}(k_{0,i}^s)(\Delta \boldsymbol{u} \cdot \boldsymbol{x}^s)_i^n \rangle_i \tag{36}$$

where we have replaced dot products with averages over indices. Using **Equation 18a**, and taking into account the fact that $w_{0,i}$ and $k_{0,i}$ are independent, we observe that

$$\frac{N}{n!} \langle w_{0,i} \Psi^{(n)}(k_{0,i}^s)(\Delta \boldsymbol{u} \cdot \boldsymbol{x}^s)_i^n \rangle_i \sim N \langle w_{0,i}^{n+1} \rangle_i \times O(1). \tag{37}$$

Similarly, this time using both **Equation 18a** and **Equation 18b**, we have

$$\frac{N}{n!} \langle \Delta w_i \Psi^{(n)}(k_{0,i}^s)(\Delta \boldsymbol{u} \cdot \boldsymbol{x}^s)_i^n \rangle_i \sim \langle w_{0,i}^n \rangle_i \times O(1). \tag{38}$$

Inserting these into **Equation 36** then gives us

$$h^s_{\text{higher order}} \sim \sum_{n=2}^{\infty} \frac{1}{n!} N \langle w_{0,i}^{n+1} \rangle_i \times O(1) + \sum_{n=1}^{\infty} \frac{1}{n!} \langle w_{0,i}^n \rangle_i \times O(1). \tag{39}$$

Finally, using the fact that the $w_{0,i}$ are drawn independently from a zero-mean Gaussian with variance $N^{-1}$ (**Equation 8a**), we see that $\langle w_{0,i}^n \rangle_i$ is proportional to $N^{-n/2}$ when $n$ is even and $N^{-(n+1)/2}$ when $n$ is odd. Consequently, the largest term in the expression for $h^s_{\text{higher order}}$ is proportional to $N^{-1}$. The higher-order terms can, therefore, be neglected in the large $N$ limit.

## Evolution of activity in finite-size networks

The equations that describe the evolution of connectivity and activity that were derived in Sections Evolution of connectivity and Evolution of activity are accurate if two assumptions are satisfied: (1) the circuit is very large ($N \gg 1$), and (2) the synaptic weights are initialized to be $O(N^{-1/2})$ (**Equation 8a**), which guarantees that synaptic drives and activity neither vanish nor explode at initialization. Both assumptions are reasonable for brain circuits, and correspond to rather standard modelling choices in theoretical neuroscience.

In this work, we use the analytical expressions derived for large $N$ to describe activity evolution in finite-size networks. This is a crude approximation, as dealing with finite $N$ would require, in principle, integrating corrective terms into our equations (**Huang and Yau, 2020**). How accurate is this approximation? Several machine-learning studies have investigated this question across tasks, architectures, and loss functions (**Jacot et al., 2018**; **Chizat et al., 2019**; **Hu et al., 2020**; **Geiger et al., 2020**; **Yang and Hu, 2021**). Because of the Taylor expansions used in Sections Evolution of connectivity and Evolution of activity, for fixed $N$, good accuracy is expected when the amplitude of activity changes is small. Via **Equation 29**, we see that the latter increases with the number of sensory input vectors $P$, implying that good accuracy is expected when $P$ is small. For fixed $P$, furthermore, the amplitude of activity changes increases with correlations among sensory inputs (**Equation 28**), implying that good accuracy is expected when sensory input correlations are small. As detailed in Sections Simple task: task definition and Context-dependent task: task definition, sensory input correlations are smaller in the simple than in the context-dependent task, which implies that accuracy in the former task is expected to be higher than in the latter. The amplitude of activity changes also depends on the amplitude of activity coordinates $c^s$ (**Equation 29**). We show in Sections Simple task: computing activity and Context-dependent task: computing activity that activity coordinates are usually smaller in the simple

than in the context-dependent task, which again implies that accuracy in the former task is expected to be higher than in the latter. Overall, those arguments suggest that good accuracy is expected when the task is easy, and thus the training loss converges to zero very quickly (*Hu et al., 2020*). Finally, we expect accuracy to depend on properties of the activation function $\Psi$, with accuracy increasing as $\Psi$ becomes more linear in its effective activation range.

In *Figure 3—figure supplement 1*, *Figure 3—figure supplement 2*, *Figure 6—figure supplement 1*, and *Figure 7—figure supplement 1*, we evaluate accuracy by performing a systematic comparison between approximate analytical expressions (large $N$) and circuit simulations (finite $N$). We find good agreement for the full range of parameters considered in the study. Specifically, the theory correctly predicts qualitatively, and in some cases also quantitatively, the behaviour of all activity measures discussed in the main text. As expected, the agreement is stronger in the simple (*Figure 3—figure supplement 1* and *Figure 3—figure supplement 2*) than in the context-dependent task (*Figure 6—figure supplement 1* and *Figure 7—figure supplement 1*).

## Simple categorization task

### Simple task: task definition

We first consider a simple categorization task. Each stimulus is represented by an input pattern $\boldsymbol{\mu}^S$, with $S = 1, \ldots, Q$, where $Q$ is the total number of stimuli. The $\boldsymbol{\mu}^S$ are random vectors whose entries are drawn independently from a zero-mean, unit-variance Gaussian distribution. Every sensory input vector $\boldsymbol{x}^s$ corresponds to a stimulus,

$$\boldsymbol{x}^s = \boldsymbol{\mu}^S; \tag{40}$$

consequently, the number of sensory input vectors, $P$, is equal to $Q$ (the upper-case notation $S$ is used for consistency with the context-dependent task; see Section Context-dependent task: task definition). To leading order in $N$, sensory input vectors are thus orthonormal,

$$\frac{\boldsymbol{x}^s \cdot \boldsymbol{x}^{s'}}{N} = \langle x_i^s x_i^{s'} \rangle_i \simeq \delta_{ss'} \tag{41}$$

where $\delta_{ss'}$ is the Kronecker delta.

Each stimulus is associated with one among the two mutually exclusive categories A and B: the first half of stimuli is associated with A, the second half with B. The target value $\tilde{z}^s$ for the readout neuron is thus equal to $z^A$ for the first half of sensory inputs and $z^B$ for the second half. Since sensory input vectors are approximately orthogonal to each other, they are also linearly separable.

Our goal is to derive explicit expressions for the quantities analyzed in the main text: category selectivity (defined in *Equation 56*), and category correlation (defined in *Equation 71*). Both quantities depend on activity in the intermediate layer, $\boldsymbol{y}^s$, after learning, which is given in *Equation 29*. In the next section, we then write down an explicit expression for $\boldsymbol{y}^s$; after that, we compute category selectivity (Section Simple task: category selectivity) and category correlation (Section Simple task: category correlation). Further mathematical details are discussed in Sections Simple task: computing normalized dot products, Asymmetry in category response and Characterizing variability; a generalization of the current task is discussed and analyzed in Section Simple categorization task with structured inputs and heterogeneity.

### Simple task: computing activity

Examining *Equation 29*, we see that to compute the activity in the intermediate layer, $\boldsymbol{y}^s$, we need the asymptotic activity coordinates, $c^q$, and the spanning vectors, $\boldsymbol{v}^{qs}$. We start with the coordinates. To compute them, we solve the linear system of equations given in *Equation 26*. Using *Equation 41*, that system of equations becomes

$$\Phi^{-1}(\tilde{z}^s) - h_0^s = \langle \Psi'(k_{0,i}^s)^2 \rangle_i c^s + \frac{\eta_w}{\eta_u} \sum_{q=1}^{P} \langle \Psi(k_{0,i}^s) \Psi(k_{0,i}^q) \rangle_i c^q. \tag{42}$$

As a first step, we simplify the averages in the right-hand side. The law of large number guarantees that, when $N$ is large, the elements of the synaptic drive, $k_{0,i}^s$, are independently drawn from a Gaussian distribution. The statistics of this distribution are given by

$$\langle k_{0,i}^s \rangle_i = \sum_{j=1}^{N} \langle u_{0,ij} \rangle_i x_j^s = 0 \tag{43a}$$

$$\langle k_{0,i}^q k_{0,i}^s \rangle_i = \sum_{j=1}^{N} \sum_{j'=1}^{N} \langle u_{0,ij} u_{0,ij'} \rangle_i x_j^q x_{j'}^s = \langle x_j^q x_j^s \rangle_j \tag{43b}$$

where we have used the fact that, because of *Equation 8a*, $\langle u_{0,ij} u_{0,ij'} \rangle_i = \delta_{jj'}/N$. *Equation 43*, combined with *Equation 41*, implies that the $k_{0,i}^s$ have zero mean and unit variance, and are uncorrelated across stimuli. In addition, because the statistics of $k_{0,i}^s$ are independent of $s$, averages over $i$ of any function of $k_{0,i}^s$ are independent of $s$.

Using these observations, *Equation 42* can be written as

$$\Phi^{-1}(\tilde{z}^s) - h_0^s = \left( \langle \Psi'^2 \rangle + \frac{\eta_w}{\eta_u} \left( \langle \Psi^2 \rangle - \langle \Psi \rangle^2 \right) \right) c^s + \frac{\eta_w}{\eta_u} \langle \Psi \rangle^2 \sum_{q=1}^{P} c^q \tag{44}$$

where we used the short-hand notation $\langle F \rangle$ to indicate the average of a function $F$ whose argument is drawn from a zero-mean, unit-variance Gaussian distribution. That is,

$$\langle F \rangle \equiv \langle F(a) \rangle_a \tag{45}$$

where $a$ is a zero-mean, unit-variance Gaussian variable. This average can be computed via numerical integration, as detailed in Section Evaluation of averages (*Equation 184*).

The left-hand side of *Equation 44* consists of two terms: the target $\Phi^{-1}(\tilde{z}^s)$, which is fixed by the task, and $h_0^s$ (representing the synaptic drive of the readout neuron at initialization), which fluctuates across model realizations. The presence of the latter term indicates that connectivity and activity changes are not fully self-averaging; they are rather tuned to compensate for the initial state of the readout neuron. Here, we seek to analyze the *average* behaviour of the model, and so we drop the second, variable term. This approximation is discussed in detail in Section Characterizing variability.

With the variable terms neglected, the left-hand side of *Equation 44* can take only two values: $\Phi^{-1}(z^A)$ and $\Phi^{-1}(z^B)$. Combined with the symmetry of the right-hand side, this implies that the coordinates $c^s$ themselves can take only two values. Specifically, we have

$$c^s = \begin{cases} c^A & s \text{ in category A} \\ c^B & s \text{ in category B} . \end{cases} \tag{46}$$

The category-dependent coordinates, $c^A$ and $c^B$, are determined by the two-dimensional linear system of equations

$$\Phi^{-1}(z^A) = \alpha c^A + \beta(c^A + c^B) \tag{47a}$$

$$\Phi^{-1}(z^B) = \alpha c^B + \beta(c^A + c^B) \tag{47b}$$

where the scalars $\alpha$ and $\beta$ are defined as

$$\alpha = \langle \Psi'^2 \rangle + \frac{\eta_w}{\eta_u}(\langle \Psi^2 \rangle - \langle \Psi \rangle^2) \tag{48a}$$

$$\beta = \frac{\eta_w}{\eta_u} \frac{Q}{2} \langle \Psi \rangle^2 . \tag{48b}$$

This system is easily solved, yielding

$$c^A = \frac{1}{\alpha + 2\beta} \left( \Phi^{-1}(z^A) + \gamma \right) \tag{49a}$$

$$c^B = \frac{1}{\alpha + 2\beta} \left( \Phi^{-1}(z^B) - \gamma \right) \tag{49b}$$

where we have defined the shift

$$\gamma = \frac{\beta}{\alpha} \left( \Phi^{-1}(z^A) - \Phi^{-1}(z^B) \right) . \tag{50}$$

Note that $\gamma$ is positive, as $\alpha, \beta > 0$ and $\Phi^{-1}(z^A) > \Phi^{-1}(z^B)$, which in turn indicates that $c^A > c^B$.

To conclude the derivation of activity, we evaluate the spanning vectors, $v^{qs}$ (**Equation 28**). Because the sensory inputs $x^s$ are orthogonal (**Equation 41**), spanning vectors with $q \neq s$ vanish. Consequently, the activity, $y^s$ (**Equation 29**), reads

$$
y^s = \begin{cases} y_0^s + c^A v^{ss} & s \text{ in category A} \\ y_0^s + c^B v^{ss} & s \text{ in category B} . \end{cases} \tag{51}
$$

Using **Equation 31**, we can rewrite this as

$$
y^s = \begin{cases} y_0^s + c^A \rho w_0 + c^A \delta v^{ss} & s \text{ in category A} \\ y_0^s + c^B \rho w_0 + c^B \delta v^{ss} & s \text{ in category B} \end{cases} \tag{52}
$$

where we used **Equation 30** to define

$$
\rho \equiv \rho^{ss} = \langle \Psi'^2 \rangle . \tag{53}
$$

**Equation 52** indicates that activity consists of three components. The first one coincide with initial activity, $y_0^s$, which for this task is fully unstructured. The second one is a shared component along $w_0$ (whose strength is category dependent, as it is given by $c^A$ or $c^B$). The third one is a non-shared component along the residuals $\delta v^{ss}$, which represent the components of the spanning vectors that are perpendicular to the initial readout $w_0$. For the current task, the latter component is orthogonal across activity vectors, implying that activity vectors only overlap along $w_0$. To leading order in $N$, in fact

$$
\delta v^{ss} \cdot \delta v^{s's'} = \langle x_i^s x_i^s \rangle_i \langle x_i^{s'} x_i^{s'} \rangle_i \left[ \langle \Psi'(k_{0,i}^s) \Psi'(k_{0,i}^s) \Psi'(k_{0,i}^{s'}) \Psi'(k_{0,i}^{s'}) \rangle_i - \langle \Psi'(k_{0,i}^s)^2 \rangle_i^2 \right] \simeq 0 , \tag{54}
$$

which follows because $k_{0,i}^s$ and $k_{0,i}^{s'}$ are uncorrelated.

We observe that **Equation 52** is similar, but not identical to the expression that we used in the main text to describe activity evolution (**Equation 3**). By setting $d = \rho w_0$, that reads

$$
y^s = \begin{cases} y_0^s + c^A \rho w_0 & s \text{ in category A} \\ y_0^s + c^B \rho w_0 & s \text{ in category B} . \end{cases} \tag{55}
$$

Comparing **Equation 52** with **Equation 55**, we see that the residuals $\delta v^{ss}$ were neglected in the main text. This could be done because, for the current task (but not for the context-dependent one, see Section Context-dependent task: computing activity), residuals are all orthogonal to each other (**Equation 54**). As such, they do not add novel structure to activity, and do not significantly contribute to activity measures. This is showed and justified, in detail, in the next sections.

## Simple task: category selectivity

In this section, we evaluate the category selectivity of neurons in the intermediate layer (**Figure 2B, F, J**). For each neuron $i$, we evaluate the standard selectivity index (**Freedman and Assad, 2006**), defined in **Equation 2**. We repeat that definition here for convenience,

$$
S_i = \frac{\langle (y_i^s - y_i^{s'})^2 \rangle_{s,s' \text{ diff cat}} - \langle (y_i^s - y_i^{s'})^2 \rangle_{s \neq s' \text{ same cat}}}{\langle (y_i^s - y_i^{s'})^2 \rangle_{s,s' \text{ diff cat}} + \langle (y_i^s - y_i^{s'})^2 \rangle_{s \neq s' \text{ same cat}}} \tag{56}
$$

where the notation $\langle \cdot \rangle_{s,s'}$ denotes an average over sensory input pairs associated either with different, or the same, category. To evaluate this expression, we assume that the number of stimuli, $Q = P$, is moderately large ($1 \ll Q \ll N$). We show that, under this assumption, the category selectivity index for each neuron, which is approximately zero at $t = 0$, becomes positive over learning.

We start with

$$
y_i^s = y_{0,i}^s + c^s (\rho w_{0,i} + \delta v_i^{ss}) , \tag{57}
$$

which follows from *Equation 52*. The first term of the right-hand side is $O(1)$, while both terms in parentheses are $O(N^{-1/2})$. Thus, when evaluating the denominator in *Equation 56*, to lowest non-vanishing order in $N$ we can replace $y_i$ with $y_{0,i}$. Doing that, and expanding the square, we have

$$\langle (y_i^s - y_i^{s'})^2 \rangle_{s,s' \text{ diff cat}} + \langle (y_i^s - y_i^{s'})^2 \rangle_{s \neq s' \text{ same cat}} \simeq 4\langle (y_{0,i}^s)^2 \rangle_s - 2\langle y_{0,i}^s y_{0,i}^{s'} \rangle_{s,s' \text{ diff cat}} - 2\langle y_{0,i}^s y_{0,i}^{s'} \rangle_{s \neq s' \text{ same cat}}. \tag{58}$$

Noting that $y_{0,i}^s = \Psi(k_{0,i}^s)$, and using *Equation 43*, we see that the second two averages in the above equation are both equal to $\langle y_{0,i}^s \rangle_s^2$. Consequently,

$$\langle (y_i^s - y_i^{s'})^2 \rangle_{s,s' \text{ diff cat}} + \langle (y_i^s - y_i^{s'})^2 \rangle_{s \neq s' \text{ same cat}} \simeq 4\langle (y_{0,i}^s)^2 \rangle_s - 4\langle y_{0,i}^s \rangle_s^2. \tag{59}$$

Strictly speaking, this step is accurate only in the large-$Q$ limit, but is a good approximation even for moderate $Q$. Since $Q$ is moderately large, we can further approximate this as

$$\langle (y_i^s - y_i^{s'})^2 \rangle_{s,s' \text{ diff cat}} + \langle (y_i^s - y_i^{s'})^2 \rangle_{s \neq s' \text{ same cat}} \simeq 4\langle \Psi^2 \rangle - 4\langle \Psi \rangle^2, \tag{60}$$

where averages can be computed as described in Section Evaluation of averages.

For the numerator of *Equation 56*, the minus sign causes the $(y_{0,i}^s)^2$ terms to cancel, so we have

$$\langle (y_i^s - y_i^{s'})^2 \rangle_{s,s' \text{ diff cat}} - \langle (y_i^s - y_i^{s'})^2 \rangle_{s \neq s' \text{ same cat}} = 2\langle y_i^s y_i^{s'} \rangle_{s \neq s' \text{ same cat}} - 2\langle y_i^s y_i^{s'} \rangle_{s,s' \text{ diff cat}}. \tag{61}$$

Using *Equation 57*, we have (for $s \neq s'$)

$$y_i^s y_i^{s'} = y_{0,i}^s y_{0,i}^{s'} + c^{s'} y_{0,i}^s (\rho w_{0,i} + \delta v_i^{s's}) + c^s y_{0,i}^{s'} (\rho w_{0,i} + \delta v_i^{ss}) + c^s c^{s'} (\rho w_{0,i} + \delta v_i^{s's})(\rho w_{0,i} + \delta v_i^{ss}). \tag{62}$$

Apart from the first term, and the term proportional to $w_{0,i}^2$, all terms in the right-hand side have essentially random signs. Neglecting those for a moment, we obtain

$$y_i^s y_i^{s'} \simeq y_{0,i}^s y_{0,i}^{s'} + c^s c^{s'} \rho^2 w_{0,i}^2. \tag{63}$$

Inserting this into *Equation 61*, using the fact that $\langle y_{0,i}^s y_{0,i}^{s'} \rangle$ is independent of $s$ and $s'$, and performing a small amount of algebra, we arrive at

$$\langle (y_i^s - y_i^{s'})^2 \rangle_{s,s' \text{ diff cat}} - \langle (y_i^s - y_i^{s'})^2 \rangle_{s \neq s' \text{ same cat}} \simeq w_{0,i}^2 (c^A - c^B)^2 \rho^2. \tag{64}$$

Combining this with *Equation 60*, and using *Equation 53* for $\rho$, we arrive at

$$S_i \simeq \frac{w_{0,i}^2 (c^A - c^B)^2 \langle \Psi'^2 \rangle^2}{4(\langle \Psi^2 \rangle - \langle \Psi \rangle^2)}. \tag{65}$$

We conclude that single-neuron selectivity vanishes at $t = 0$ (when $c^A = c^B = 0$), and is positive at the end of learning. Furthermore, for each neuron, the magnitude of selectivity is determined by the magnitude of $w_{0,i}$, which measures initial connectivity with the readout neuron. As a result, neurons with large initial connectivity develop large selectivity values (*Figure 5B, C*).

Because of the factor $w_{0,i}^2$, the right-hand side of *Equation 65* is $O(N^{-1})$. To derive *Equation 65*, we neglected terms in the numerator that have random sign and thus contribute as noise. The dominant random terms are $O(1)$ in $N$, but $O(Q^{-1})$ in $Q$. This implies that, in simulated circuits with finite

$Q$, random deviations from *Equation 65* occur. For example, *Figure 2B* shows that selectivity values at $t = 0$ are small but non-zero; *Figure 5B, C*, instead, shows that the values of $S_i$ and $w_{0,i}^2$ are not perfectly correlated across the population.

We can, finally, average *Equation 65* over neurons, yielding

$$S = \langle S_i \rangle_i \simeq \frac{(c^A - c^B)^2 \langle \Psi'^2 \rangle^2}{4N(\langle \Psi^2 \rangle - \langle \Psi \rangle^2)} \tag{66}$$

where neglected random terms are now $O(N^{-1/2}Q^{-1})$. In **Figure 3—figure supplement 1** and **Figure 3—figure supplement 2**, we compare this approximate analytical expression for average category selectivity with values measured in finite-size circuits, and find good agreement between the two.

## Category clustering

Our derivation of the average category selectivity, **Equation 66**, was based on several assumptions: we assumed that the number of stimuli, $Q$, was large, and that terms with random signs could be neglected. A different, but related, activity measure is given by category clustering (**Bernardi et al., 2020**; **Engel et al., 2015**). That is defined as

$$\tilde{S} = \frac{\langle\langle(y_i^s - y_i^{s'})^2\rangle_i\rangle_{s,s'\text{ diff cat}} - \langle\langle(y_i^s - y_i^{s'})^2\rangle_i\rangle_{s\neq s'\text{ same cat}}}{\langle\langle(y_i^s - y_i^{s'})^2\rangle_i\rangle_{s,s'\text{ diff cat}} + \langle\langle(y_i^s - y_i^{s'})^2\rangle_i\rangle_{s\neq s'\text{ same cat}}}. \tag{67}$$

This measure is positive if activity vectors elicited by within-category stimuli are more similar, in norm, than activity vectors elicited by across-category stimuli – and negative otherwise. In contrast to average category selectivity, category clustering can be evaluated straightforwardly, and for any value of $Q$. We show this in the following.

By using the statistical homogeneity of activity vectors, we can rewrite

$$\tilde{S} = \frac{-\langle\langle y_i^s y_i^{s'}\rangle_i\rangle_{s,s'\text{ diff cat}} + \langle\langle y_i^s y_i^{s'}\rangle_i\rangle_{s\neq s'\text{ same cat}}}{2\langle\langle(y_i^s)^2\rangle_i\rangle_s - \langle\langle y_i^s y_i^{s'}\rangle_i\rangle_{s,s'\text{ diff cat}} - \langle\langle y_i^s y_i^{s'}\rangle_i\rangle_{s\neq s'\text{ same cat}}}. \tag{68}$$

Expressions in the form of $\langle y_i^s y_i^{s'}\rangle_i$ are evaluated in Section Simple task: computing normalized dot products; the derivation involves lengthy, but straightforward algebra. Using those results (**Equation 90** and **Equation 95**), we have:

$$\tilde{S} = \frac{(c^A - c^B)^2 \langle\Psi'^2\rangle^2}{4N(\langle\Psi^2\rangle - \langle\Psi\rangle^2) + 2\left[(c^A)^2 + (c^B)^2\right]\langle\Psi'^4\rangle - (c^A + c^B)^2\langle\Psi'^2\rangle^2}. \tag{69}$$

To the leading order in $N$, we obtain

$$\tilde{S} = \frac{(c^A - c^B)^2 \langle\Psi'^2\rangle^2}{4N(\langle\Psi^2\rangle - \langle\Psi\rangle^2)} \tag{70}$$

which is identical to the expression obtained for average category selectivity evaluated with $Q$ large (**Equation 66**).

To better understand the relationship between selectivity and clustering, we observe that clustering coincide with the average selectivity, $S = \langle S_i\rangle_i$, if the average over the numerator and the denominator of $S_i$ (**Equation 56**) is factorized. In general, the numerator and the denominator of $S_i$ are correlated, and the average cannot be factorized. We have however shown that, in the limit where both $Q$ and $N$ are large, $S_i$ can be approximated by an expression where the denominator is independent of $i$ (**Equation 65**). In that regime, the average can be factorized; average category selectivity $S$ and category clustering $\tilde{S}$ thus take very similar values, as quantified by **Equation 66** and **Equation 70**. We conclude that, for our activity expressions, average category selectivity and category clustering are expected to behave similarly when both $Q$ and $N$ are large. A detailed comparison between average selectivity and clustering within data from simulated circuits is provided in **Figure 3—figure supplement 1** and **Figure 3—figure supplement 2**.

## Simple task: category correlation

To quantify how the population as a whole responds to the two categories, we evaluate category correlation. This quantity, denoted $C$, is given by the average Pearson correlation coefficient of activity in response to stimuli associated with different categories. We have:

$$C = \langle C^{s_A s_B}\rangle_{s_A s_B} \tag{71}$$

where $s_A$ and $s_B$ are indices that denote sensory inputs associated, respectively, with categories A and B. The Pearson correlation $C^{s_A s_B}$ is given by

$$C^{s_A s_B} = \frac{\langle y_i^{s_A} y_i^{s_B} \rangle_i - \langle y_i^{s_A} \rangle_i \langle y_i^{s_B} \rangle_i}{\sqrt{\langle y_i^{s_A} y_i^{s_A} \rangle_i - \langle y_i^{s_A} \rangle_i^2} \sqrt{\langle y_i^{s_B} y_i^{s_B} \rangle_i - \langle y_i^{s_B} \rangle_i^2}}$$
$$= \frac{\langle y_i^{s_A} y_i^{s_B} \rangle_i - \langle \Psi \rangle^2}{\sqrt{\langle y_i^{s_A} y_i^{s_A} \rangle_i - \langle \Psi \rangle^2} \sqrt{\langle y_i^{s_B} y_i^{s_B} \rangle_i - \langle \Psi \rangle^2}}. \tag{72}$$

To go from the first to the second line, we used the fact that, for each sensory input,

$$\langle y_i^s \rangle_i = \langle y_{0,i}^s \rangle_i + c^s \langle v_i^{ss} \rangle_i = \langle \Psi \rangle \tag{73}$$

where the second equality follows from $\langle w_{0,i} \rangle_i = 0$ (**Equation 8a**), which in turns implies that $\langle v_i^{ss} \rangle_i = 0$ (**Equation 28**). Pearson correlation coefficients are displayed in the correlation matrices of **Figure 2C, G, K**.

As we show in Section Simple task: computing normalized dot products, in the large-$N$ limit, $\langle y_i^s y_i^{s'} \rangle_i$ only depends on the category $s$ and $s'$ are in. This makes the average over $s_A$ and $s_B$ in **Equation 71** trivial. Using **Equation 90** and **Equation 95**, we arrive at

$$C = \frac{c^A c^B \langle \Psi'^2 \rangle^2}{\sqrt{N(\langle \Psi^2 \rangle - \langle \Psi \rangle^2) + (c^A)^2 \langle \Psi'^4 \rangle} \sqrt{N(\langle \Psi^2 \rangle - \langle \Psi \rangle^2) + (c^B)^2 \langle \Psi'^4 \rangle}}. \tag{74}$$

In **Figure 3—figure supplement 1** and **Figure 3—figure supplement 2**, we compare this approximate analytical expression with values measured in finite-size circuits, and find good agreement between the two. We can further simplify **Equation 74** by Taylor expanding in $N$. To leading order, we obtain

$$C = \frac{1}{N} \frac{c^A c^B \langle \Psi'^2 \rangle^2}{\langle \Psi^2 \rangle - \langle \Psi \rangle^2}. \tag{75}$$

Before learning, $c^A = c^B = 0$, and so correlation vanishes. After learning, $C$ is non-zero, and its sign is given by the sign of the product $c^A c^B$. This has a simple geometric explanation: after mean subtraction, activity vectors associated with opposite categories only overlap along the direction spanned by the initial readout vector $w_0$. The coordinates of vectors associated with categories A and B along this direction are proportional, respectively, to $c^A$ and $c^B$ (**Equation 55**). When $c^A$ and $c^B$ have opposite sign, activity vectors acquire opposite components along $w_0$, which generates negative category correlation. When $c^A$ and $c^B$ have identical sign, instead, activity vectors acquire aligned components, which generates positive category correlation.

To determine how the product $c^A c^B$ depends on parameters, we use **Equation 49** for $c^A$ and $c^B$ to write

$$c^A c^B = \frac{1}{(\alpha + 2\beta)^2} \left( \Phi^{-1}(z^A) + \gamma \right) \left( \Phi^{-1}(z^B) - \gamma \right) \tag{76}$$

where the (positive) scalars $\alpha$ and $\beta$ are defined in **Equation 48**, and $\gamma$ in **Equation 50**. Consequently, the sign of $c^A c^B$, and thus, the sign of the category correlation $C$, depends on the value of the target synaptic drives $\Phi^{-1}(z^A)$ and $\Phi^{-1}(z^B)$ (**Figure 2—figure supplement 2F**), as well as on $\gamma$.

In particular, when $\Phi^{-1}(z^A)$ and $\Phi^{-1}(z^B)$ have opposite sign, **Equation 76** can only be negative, and thus category correlation can only be negative. When $\Phi^{-1}(z^A)$ and $\Phi^{-1}(z^B)$ have identical sign, **Equation 76** can be either negative or positive, depending on the value of the shift $\gamma$, and thus category correlation can be either negative or positive. For fixed target values $z^A$ and $z^B$, the relative sign of $\Phi^{-1}(z^A)$ and $\Phi^{-1}(z^B)$ depends on the shape of the activation function of the readout neuron $\Phi$. In the example given in **Figure 2—figure supplement 2F**, we show that the relative sign of $\Phi^{-1}(z^A)$ and $\Phi^{-1}(z^B)$ can be modified by changing the threshold of $\Phi$. More in general, changing both the gain and threshold of $\Phi$ can change the sign and magnitude of category correlation (**Figure 4A**).

What controls the value of the shift $\gamma$ (and, thus, the sign of correlation when $\Phi^{-1}(z^A)$ and $\Phi^{-1}(z^B)$ have identical sign)? Combining **Equation 50** for $\gamma$ with **Equation 48** for $\alpha$ and $\beta$, we have

$$\gamma = \frac{Q \langle \Psi \rangle^2}{2 \left[ (\eta_w/\eta_u)^{-1} \langle \Psi'^2 \rangle + (\langle \Psi^2 \rangle - \langle \Psi \rangle^2) \right]} \left( \Phi^{-1}(z^A) - \Phi^{-1}(z^B) \right). \tag{77}$$

Recall that $\Phi^{-1}(z^A) - \Phi^{-1}(z^B)$ is always positive. We observe that $\gamma$ depends on the learning rate ratio $\eta_w/\eta_u$: increasing this ratio increases the value of $\gamma$ and thus, via **Equation 76**, favours negative correlation (**Figure 4C**). It also depends on the number of stimuli, $Q$: increasing $Q$ increases the value

of $\gamma$, and thus also favours negative correlation (**Figure 4D**). Finally, $\gamma$ depends on the activation function of neurons in the intermediate layer, $\Psi$, through nonlinear population averages; by computing those averages, we find that decreasing the gain and threshold of $\Psi$ favours negative correlation (**Figure 4B**).

## Alternative definition

For completeness, we observe that an alternative way of quantifying category correlation consists of averaging activity over stimuli first (**Figure 2D, H and L**), and then computing the Person correlation coefficient between averaged responses. The correlation values obtained via this procedure are displayed in the legend of **Figure 2D, H, L**. This alternative definition yields qualitatively identical results to **Equation 71**; we show this below.

We start by defining the category-averaged activity

$$\boldsymbol{y}^{A} = \langle \boldsymbol{y}^{s_A} \rangle_{s_A} \tag{78a}$$

$$\boldsymbol{y}^{B} = \langle \boldsymbol{y}^{s_B} \rangle_{s_B}. \tag{78b}$$

We then define category correlation as

$$
\begin{aligned}
C &= \frac{\langle y_i^A y_i^B \rangle_i - \langle y_i^A \rangle_i \langle y_i^B \rangle_i}{\sqrt{\langle y_i^A y_i^A \rangle_i - \langle y_i^A \rangle_i^2} \sqrt{\langle y_i^B y_i^B \rangle_i - \langle y_i^B \rangle_i^2}} \\
&= \frac{\langle y_i^A y_i^B \rangle_i - \langle \Psi \rangle^2}{\sqrt{\langle y_i^A y_i^A \rangle_i - \langle \Psi \rangle^2} \sqrt{\langle y_i^B y_i^B \rangle_i - \langle \Psi \rangle^2}}.
\end{aligned}
\tag{79}
$$

Then, using **Equation 90** and **Equation 95** from Section Simple task: computing normalized dot products, we have

$$\langle y_i^A y_i^B \rangle_i = \left(\frac{2}{Q}\right)^2 \sum_{s_A=1}^{Q/2} \sum_{s_B=1}^{Q/2} \langle y_i^{s_A} y_i^{s_B} \rangle_i = \langle \Psi \rangle^2 + N^{-1} c^A c^B \langle \Psi'^2 \rangle^2, \tag{80}$$

while

$$
\begin{aligned}
\langle y_i^A y_i^A \rangle_i &= \left(\frac{2}{Q}\right)^2 \sum_{s_A=1}^{Q/2} \left[ \langle y_i^{s_A} y_i^{s_A} \rangle_i + \sum_{s_A' \neq s_A} \langle y_i^{s_A} y_i^{s_A'} \rangle_i \right] \\
&= \langle \Psi \rangle^2 + \frac{2}{Q}\left(\langle \Psi^2 \rangle - \langle \Psi \rangle^2\right) + N^{-1}(c^A)^2 \left[ \langle \Psi'^2 \rangle^2 + \frac{2}{Q}\left(\langle \Psi'^4 \rangle - \langle \Psi'^2 \rangle^2\right) \right]
\end{aligned}
\tag{81}
$$

and similarly for $\langle y_i^B y_i^B \rangle_i$, by replacing $c^A$ with $c^B$. Inserting this into **Equation 79**, we arrive at

$$C = \frac{c^A c^B \langle \Psi'^2 \rangle^2}{\sqrt{\frac{2}{Q}N\left(\langle \Psi^2 \rangle - \langle \Psi \rangle^2\right) + (c^A)^2\left[\langle \Psi'^2 \rangle^2 + \frac{2}{Q}\left(\langle \Psi'^4 \rangle - \langle \Psi'^2 \rangle^2\right)\right]} \sqrt{\frac{2}{Q}N\left(\langle \Psi^2 \rangle - \langle \Psi \rangle^2\right) + (c^B)^2\left[\langle \Psi'^2 \rangle^2 + \frac{2}{Q}\left(\langle \Psi'^4 \rangle - \langle \Psi'^2 \rangle^2\right)\right]}}. \tag{82}$$

Although the denominator of this expression is different from **Equation 74**, the numerator is identical. As the denominators in both expressions are positive, the qualitative behaviour of **Equation 82** is identical to **Equation 74**. Furthermore, to leading order in $N$, we obtain

$$C = \frac{Q}{2N} \frac{c^A c^B \langle \Psi'^2 \rangle^2}{\langle \Psi^2 \rangle - \langle \Psi \rangle^2} \tag{83}$$

which is proportional to **Equation 75**, with constant of proportionality equal to $Q/2$.

## Simple task: computing normalized dot products

We now compute the normalized dot products among pairs of activity vectors; namely

$$\langle y_i^s y_i^{s'} \rangle_i = \frac{\boldsymbol{y}^s \cdot \boldsymbol{y}^{s'}}{N}. \tag{84}$$

Those were used above to derive the behaviour of category clustering (Section Simple task: category selectivity) and correlations (Section Simple task: category correlation).

The dot product takes different values depending on whether or not sensory inputs $s$ and $s'$ coincide. We start with the former,

$$\langle (y_i^s)^2 \rangle_i = \langle (y_{0,i}^s)^2 \rangle_i + \langle (\Delta y_i^s)^2 \rangle_i. \tag{85}$$

We used the fact that the cross-term $\langle y_{0,i}^s \Delta y_i^s \rangle_i$ vanishes on average,

$$\langle y_{0,i}^s \Delta y_i^s \rangle_i = c^s \langle y_{0,i}^s v_i^{ss} \rangle_i = c^s \langle w_{0,i} \rangle_i \langle y_{0,i}^s \Psi'(k_{0,i}^s)^2 \rangle_i = 0 \tag{86}$$

where we used *Equation 51* for the first equality, *Equation 28* for the second, and *Equation 8a* for the third. By definition,

$$\langle (y_{0,i}^s)^2 \rangle_i = \langle \Psi^2 \rangle \tag{87}$$

while

$$\langle (\Delta y_i^s)^2 \rangle_i = (c^s)^2 \langle (v_i^{ss})^2 \rangle_i = N^{-1}(c^s)^2 \langle \Psi'^4 \rangle \tag{88}$$

where we have used the fact that, from *Equation 28*

$$\langle (v_i^{ss})^2 \rangle_i = \langle x_i^s x_i^s \rangle_i^2 \langle w_{0,i}^2 \Psi'(k_{0,i}^s)^4 \rangle_i = N^{-1} \langle \Psi'^4 \rangle. \tag{89}$$

Putting these results together, we have

$$\langle (y_i^s)^2 \rangle_i = \begin{cases} \langle \Psi^2 \rangle + N^{-1}(c^A)^2 \langle \Psi'^4 \rangle & s \text{ in category A} \\ \langle \Psi^2 \rangle + N^{-1}(c^B)^2 \langle \Psi'^4 \rangle & s \text{ in category B}. \end{cases} \tag{90}$$

Note that activity vectors associated with different categories are characterized by different norms (unless coordinates are fine-tuned to be symmetric: $c^A = -c^B$, which occurs when $\Phi^{-1}(z^A) = -\Phi^{-1}(z^B)$, as in *Figure 2E–H*). Asymmetry of activity in response to different categories is discussed in detail in Section Asymmetry in category response.

For dot products among different activity vectors, we have

$$\langle y_i^s y_i^{s'} \rangle_i = \langle y_{0,i}^s y_{0,i}^{s'} \rangle_i + \langle \Delta y_i^s \Delta y_i^{s'} \rangle_i. \tag{91}$$

with $s \neq s'$. In this case,

$$\langle y_{0,i}^s y_{0,i}^{s'} \rangle_i = \langle \Psi \rangle^2 \tag{92}$$

while

$$\langle \Delta y_i^s \Delta y_i^{s'} \rangle_i = c^s c^{s'} \langle v_i^{ss} v_i^{s's} \rangle_i = N^{-1} c^s c^{s'} \langle \Psi'^2 \rangle^2, \tag{93}$$

which comes from

$$\langle v_i^{ss} v_i^{s's'} \rangle_i = \langle x_i^s x_i^s \rangle_i \langle x_i^{s'} x_i^{s'} \rangle_i \langle w_{0,i}^2 \Psi'(k_{0,i}^s)^2 \Psi'(k_{0,i}^{s'})^2 \rangle_i = N^{-1} \langle \Psi'^2 \rangle^2. \tag{94}$$

Putting this together, we arrive at

$$\langle y_i^s y_i^{s'} \rangle_i = \begin{cases} \langle \Psi \rangle^2 + N^{-1}(c^A)^2 \langle \Psi'^2 \rangle^2 & s, s' \text{ in category A} \\ \langle \Psi \rangle^2 + N^{-1}(c^B)^2 \langle \Psi'^2 \rangle^2 & s, s' \text{ in category B} \\ \langle \Psi \rangle^2 + N^{-1} c^A c^B \langle \Psi'^2 \rangle^2 & s, s' \text{ in diff. categories}. \end{cases} \tag{95}$$

*Equation 95* has a simple geometric interpretation. The first term in the right-hand side, $\langle \Psi \rangle^2$, is generated by the overlap between the activity vectors along the direction spanned by the unit vector **1**. This component is due to the activation function $\Psi$ being positive, and is approximately constant over learning. The second term on the right-hand side emerges over learning. This arises because activity vectors become aligned, via the spanning vectors (*Equation 30*), along the direction spanned

by the initial readout vector $w_0$. Note that the components of activity that are aligned with the residual directions $\delta v^{ss}$ (*Equation 52*) do not contribute to the dot product. This can be verified by computing the dot product directly from *Equation 55*, where residuals are neglected, and observing that the same result is obtained. This was expected, as we have showed in *Equation 54* that, for the current task task, residuals are orthogonal to each other.

## Asymmetry in category response

In *Figure 2L* in the main text, activity in response to categories A and B is asymmetric: the number of neurons that respond more strongly to category A is significantly larger than the number that respond more strongly to category B. Furthermore, the mean and variance of activity across the population are larger in response to A than to B. Such asymmetry is not present at $t = 0$ (*Figure 2D*), and is thus a consequence of learning. Asymmetry has been reported in experimental data as well (*Fitzgerald et al., 2013*), where it was referred to as *biased category representations*. Here, we discuss in detail why and how response asymmetry arises in the model. We show that asymmetry is controlled by the value of the target readout activity, $z^A$ and $z^B$, and also by the shape of the activation functions of the intermediate and readout layer, $\Psi$ and $\Phi$.

*Figure 2L* displays activity in response to categories A and B averaged over stimuli; those are denoted, respectively, by $y^A$ and $y^B$ (*Equation 78*). We start deriving an explicit expression for $y^A$, from which the mean and variance across the population can be computed. Since initial activity is symmetric, we focus on the part of activity that is induced by learning. Combining *Equation 51* with *Equation 28*, we have

$$\Delta y_i^A \equiv \langle \Delta y_i^{s_A} \rangle_{s_A} \simeq c^A \langle \Psi'^2 \rangle w_{0,i} \tag{96}$$

where the last approximate equality follows if $Q$ is sufficiently large. The variance across the population is, therefore, given by

$$\langle (\Delta y_i^A)^2 \rangle_i = N^{-1} (c^A)^2 \langle \Psi'^2 \rangle^2. \tag{97}$$

For the variance across the population in response to category B, we simply replace $c^A$ with $c^B$.

Consequently, the variances in response to categories A and B are identical only if $(c^A)^2 = (c^B)^2$. From *Equation 49*, we see that this happens only if $\Phi^{-1}(z^A) = -\Phi^{-1}(z^B)$, which yields $c^A = -c^B$. *Figure 2H* shows a circuit where the activation function of the readout neuron, $\Phi$, was chosen to satisfy this relationship. In general, however, the two variances differ, and can have either $(c^A)^2 > (c^B)^2$ (the variance in response to A is lather than to B), or $(c^A)^2 < (c^B)^2$ (the opposite). *Figure 2L* corresponds to the first scenario, $(c^A)^2 > (c^B)^2$; this was achieved by setting $\Phi^{-1}(z^A) > \Phi^{-1}(z^B) > 0$, which yielded $c^A > c^B > 0$. *Figure 2—figure supplement 2A, B* correspond to the second scenario, $(c^A)^2 < (c^B)^2$; this was achieved by setting $\Phi^{-1}(z^B) < \Phi^{-1}(z^A) < 0$, which yielded $c^B < c^A < 0$. Note that in both cases, $c^A > c^B$, as it must be (*Equation 49*).

In *Figure 2L*, activity in response to category A is not only characterized by larger variance, but also larger mean. This observation does not emerge immediately from our analysis, since our equations predict that the mean of activity changes vanishes both in response to A and B: from *Equation 8a*, we see that in response to category A,

$$\langle \Delta y_i^A \rangle_i = c^A \langle \Psi'^2 \rangle \langle w_{0,i} \rangle_i = 0 \tag{98}$$

and similarly for category B. To understand how *Equation 98* can be reconciled with *Figure 2L*, recall that the equations we use for activity changes (*Equation 29*) provide a linearized estimate of activity changes, which is strictly valid only in infinitely wide networks. In finite width networks, a non-zero mean response can emerge from higher-order terms in the expansion of *Equation 27*. The leading higher-order terms of this expansion are quadratic, implying that the behaviour of the mean is controlled by the second-order derivative of the activation function of neurons in the intermediate layer, $\Psi''$. When the threshold of $\Psi$ is positive (so that activity is initialized close to the lower bound of $\Psi$), the second-order derivative $\Psi''$ is positive on average. Combined with $(c^A)^2 > (c^B)^2$, this implies that the mean of activity in response to category A is larger than to B; this case is illustrated in *Figure 2L*. When the threshold of $\Psi$ is negative (so that activity is initialized close to the upper bound of $\Psi$), the second-order derivative $\Psi''$ is negative on average. Combined with $(c^A)^2 > (c^B)^2$, this implies that the mean

of activity in response to category A is smaller than to B; this case is illustrated in *Figure 2—figure supplement 2C, D*.

Finally, *Equation 98* suggests that non-vanishing mean activity could also be obtained if the initial readout weights $w_{0,i}$ have a non-zero mean. This is likely to be verified in the brain, where intra-area connectivity is mainly excitatory. We leave the incorporation of non-zero mean connectivity, along with Dale's law, to future investigations.

## Characterizing variability

In Section Simple task: computing activity, when computing the value of activity coordinates $c^s$, we neglected the second terms within the left-hand side of *Equation 42*; because of this, the coordinates took on only two values, namely $c^A$ and $c^B$ (*Equation 46*). The neglected terms do not self-average, and thus fluctuate at random across model realizations. Had we included these variable terms, *Equation 46* would have read

$$c^s = \begin{cases} c^A + \delta^s & s \text{ in category A} \\ c^B + \delta^s & s \text{ in category B} \end{cases} \tag{99}$$

where the $\delta^s$ obey the linear system of equations

$$-h_0^s = \delta^s \left( \langle \Psi'^2 \rangle + \frac{\eta_w}{\eta_u} \left( \langle \Psi^2 \rangle - \langle \Psi \rangle^2 \right) \right) + \sum_{q=1}^{P} \delta^q \frac{\eta_w}{\eta_u} \langle \Psi \rangle^2. \tag{100}$$

Here, we further characterize the behaviour of the neglected terms $\delta^s$. For simplicity, we consider the case in which plasticity in the readout weights is much slower than plasticity in the input connectivity ($\eta_w \ll \eta_u$). In that regime, *Equation 100* greatly simplifies, and we obtain

$$\delta^s = -\frac{h_0^s}{\langle \Psi'^2 \rangle}. \tag{101}$$

There are two sources of random fluctuations in $h_0^s$: different realizations of the circuit (via different initializations of the intermediate and readout connectivity, $u$ and $w$), and different sensory inputs. In the following, we show that these two sources of variability can be decomposed, and one can write

$$h_0^s = \langle \Psi \rangle \kappa + \sqrt{\langle \Psi^2 \rangle - \langle \Psi \rangle^2} \varepsilon^s \tag{102}$$

where $\kappa$ and $\varepsilon^s$ are zero-mean, unit-variance Gaussian variables. For a given circuit realization, the value of $\kappa$ is fixed, while the value of $\varepsilon^s$ fluctuates across different sensory inputs. Combining *Equation 102* with *Equation 99*, we conclude that two different forms of variability (one that is frozen for a given circuit realization, represented by $\kappa$, and one that is not, represented by $\varepsilon^s$) impact activity coordinates $c^A$ and $c^B$; the absolute and relative amplitude of the two contributions is controlled by the shape of the activation function $\Psi$. Such factorization of variability is illustrated, for an example simulated circuit, in *Figure 2—figure supplement 2E*.

To derive *Equation 102*, we consider a given circuit realization, and assume that the number of stimuli $Q$ is sufficiently large, so that averages over stimuli approximately self-average. We start from *Equation 6b*, and compute the mean of of $h_0^s$ over sensory inputs, which yields

$$\langle h_0^s \rangle_s = \sum_{i=1}^{N} w_{0,i} \langle \Psi(k_{0,i}^s) \rangle_s = \langle \Psi \rangle \sum_{i=1}^{N} w_{0,i}. \tag{103}$$

By defining $\kappa \equiv \sum_{i=1}^{N} w_{0,i}$, the first term in the right-hand side of *Equation 102* follows. We then compute the variance of of $h_0^s$ over sensory inputs. By using:

$$\langle (h_0^s)^2 \rangle_s = \sum_{i=1}^{N} \sum_{j=1}^{N} w_{0,i} w_{0,j} \langle \Psi(k_{0,i}^s) \Psi(k_{0,j}^s) \rangle_s = \langle \Psi^2 \rangle + \langle \Psi \rangle^2 \sum_{i=1}^{N} \sum_{j \neq i} w_{0,i} w_{0,j} \tag{104}$$

and, from *Equation 103*

$$\langle h_0^s \rangle_s^2 = \langle \Psi \rangle^2 \sum_{i=1}^{N} \sum_{j=1}^{N} w_{0,i} w_{0,j} = \langle \Psi \rangle^2 + \langle \Psi \rangle^2 \sum_{i=1}^{N} \sum_{j \neq i} w_{0,i} w_{0,j} \tag{105}$$

we conclude that:

$$\langle h_0^{s\,2} \rangle_s - \langle h_0^s \rangle_s^2 = \langle \Psi^2 \rangle - \langle \Psi \rangle^2, \tag{106}$$

from which the second term in the right-hand side of *Equation 102* follows.

*Equation 66* and *Equation 74* indicate that activity measures such as category selectivity and correlation depend on the value of activity coordinates $c^A$ and $c^B$. As coordinates are variable (*Equation 99*), activity measures are variable as well. Importantly, activity measures involve averages over sensory inputs (see *Equation 56* and *Equation 71*). This implies that the two forms of variability described by *Equation 102* are expected to contribute in different ways: variability originating from the second term (which fluctuates across stimuli, and thus can be averaged out) is expected to be small, while variability originating from the first term (which is fixed for each circuit realization) is expected to be large.

Variability in simulated circuits is quantified in *Figure 3—figure supplements 1 and 2*, where it is represented as error bars. *Figure 3—figure supplement 1A* and *Figure 3—figure supplement 2A* show that variability in $c^A$ and $c^B$ is modulated by properties of the activation function $\Psi$ (third column); this is in agreement with *Equation 102*, which indicates that the magnitude of variability is $\Psi$-dependent. *Figure 3—figure supplement 1B, C* and *Figure 3—figure supplement 2B, C* show, furthermore, that variability in correlation is typically much larger than in average selectivity. This can be explained by observing that average selectivity (*Equation 66*) only depends on the difference between $c^A$ and $c^B$, so variability originating from the first, frozen term of *Equation 102* is expected to cancel; this is not the case for correlation (*Equation 74*), for which the cancellation does not occur.

## Simple categorization task with structured inputs and heterogeneity

The circuit and task we considered so far are characterized by several simplifying modelling assumptions, which allowed us to analyze activity evolution in great detail and develop useful analytical intuition. One important assumption is that sensory input vectors corresponding to different stimuli are orthogonal to each other. This choice was motivated by two observations: first, in many tasks from the experimental literature, sensory stimuli are taken to be very different from each other, and thus sensory inputs are expected to be uncorrelated (*Messinger et al., 2001*; *Fitzgerald et al., 2011*; *Wallis et al., 2001*); second, in tasks where sensory stimuli obey a continuous statistical structure (*Freedman and Assad, 2006*), pre-processing from sensory brain regions (*Albright, 1984*) is expected to decorrelate, at least partially, inputs to higher-level associative areas. A second important assumption is that neurons in the intermediate layer are statistically homogeneous, as they receive statistically identical inputs and are characterized by the same nonlinearity $\Psi$.

For some tasks and brain regions, those two assumptions might be inaccurate. For example, data collected during passive conditions (*Fanini and Assad, 2009*) indicate that some LIP neurons (*Freedman and Assad, 2006*; *Fitzgerald et al., 2011*; *Fitzgerald et al., 2013*) display weak, but significant direction tuning, which might be due to structured sensory inputs. Furthermore, activity profiles are heterogeneous, with different neurons characterized by different baseline activity levels. To investigate whether our findings extrapolate beyond our two simplifying hypotheses, here we construct a more biologically grounded model, and use simulations to systematically investigate activity evolution in the resulting circuit.

To begin with, we use sensory input vectors characterized by a continuous statistical structure, which implies continuous tuning in the intermediate layer activity prior to learning. We set

$$\boldsymbol{x}^s = \sqrt{1 - \Sigma^2} \boldsymbol{\mu}^s + \Sigma \left[ \boldsymbol{\xi}^1 \cos(\theta^s) + \boldsymbol{\xi}^2 \sin(\theta^s) \right] \tag{107}$$

where $\Sigma$ is a scalar that measures the fraction of inputs variance that is continuous. We fixed $\Sigma = 1/3$. Like $\boldsymbol{\mu}^s$, entries of the vectors $\boldsymbol{\xi}^1$ and $\boldsymbol{\xi}^2$ are generated at random from a zero-mean, unit-variance Gaussian distribution. We furthermore set

$$\theta^s = s \frac{2\pi}{Q}. \tag{108}$$

With this choice, when $s \neq s'$, we have $\langle x_i^s x_i^{s'} \rangle = \Sigma^2 \cos(2\pi(s - s')/Q)$, so stimuli with similar values of $s$ are more strongly correlated than stimuli with very different values of $s$. As in *Freedman and Assad, 2006*, we take $Q = 12$. Similar to the standard task we analyzed so far, sensory inputs with $s = 1, \ldots, Q/2$ are associated with category A, while $s = Q/2, \ldots, Q$ are associated with category B. Note that, as in the simple categorization task we analyzed so far, sensory input vectors are linearly separable for every value of $\Sigma$.

To introduce heterogeneity in the intermediate layer, we add an offset, so *Equation 5* becomes

$$\boldsymbol{y} = \Psi(\boldsymbol{k}) \tag{109a}$$

$$\boldsymbol{k} \equiv \boldsymbol{u} \cdot \boldsymbol{x} + \boldsymbol{b}. \tag{109b}$$

The entries of $\boldsymbol{b}$ are fixed bias terms that control the value of baseline activity for each neuron. We generate those entries from a zero-mean Gaussian distribution with standard deviation 0.2.

In contrast to the model we analyzed so far, initial activity is characterized by non-trivial activity measures. Specifically, initial population tuning is characterized by non-vanishing category correlation; the latter is modulated both by heterogeneity (which tends to increase signal correlations) and the continuous inputs structure (which tends to decrease them). For our choice of parameters, these two effects roughly balance each other, so that initial activity is characterized by initial correlation that is small in magnitude (*Figure 2—figure supplement 4A*).

We investigated numerically the evolution of activity with learning for this model. Two sample circuits are shown in *Figure 2—figure supplement 4B, C*; extensive analysis is presented in *Figure 2—figure supplement 4D, E*. We find that the behaviour of both category selectivity and correlation is qualitatively consistent with the behaviour of the simpler model analyzed so far. Specifically, we find that average category selectivity increases over learning *Figure 2—figure supplement 4D*; this behaviour is robust, and does not depend on circuit details. For completeness, we tested two definitions of category selectivity. The first one is identical to *Equation 56*; as initial activity is structured, this gives slightly positive initial values; the second one (which is used in related experimental work, *Freedman et al., 2001*; *Freedman and Assad, 2006*) is again identical to *Equation 56* – but pairs of stimuli $ss'$ are subsampled in a way that is tailored to inputs structure to yield vanishing initial selectivity. We show in *Figure 2—figure supplement 4D* that both selectivity definitions give qualitatively similar results. Whether category correlation increases or decreases over learning depends, on the other hand, on parameters (*Figure 2—figure supplement 4B, C, E*). Correlation depends on parameters in a way that is consistent with the simple task: it is strongly modulated by properties of the readout activation function $\Phi$ (*Figure 2—figure supplement 4E*, different shades of gray). It also depends on the activation function of neurons in the intermediate layer $\Psi$ (*Figure 2—figure supplement 4E*, left). Finally, it decreases with the learning ratio $\eta_w/\eta_u$ (*Figure 2—figure supplement 4E*, center) and with the number of stimuli $Q$ (*Figure 2—figure supplement 4E*, right).

## Context-dependent categorization task

### Context-dependent task: task definition

The second task we consider is a context-dependent categorization task. On each trial, both a stimulus, and a context cue, are presented to the network. For simplicity, we assume that the number of stimuli and context cues is identical, and is equal to $Q$. As in the simple task, each stimulus is represented by an input vector $\boldsymbol{\mu}^S$, with $S = 1, \ldots, Q$; each context cue is also represented by an input vector, denoted $\boldsymbol{\nu}^C$, with $C = 1, \ldots, Q$. The entries of both vectors, $\boldsymbol{\mu}^S$ and $\boldsymbol{\nu}^C$, are generated independently from a zero-mean, unit-variance Gaussian distribution. The total sensory input on each trial, $\boldsymbol{x}^s$, is given by the linear combination of the stimulus and context cue inputs,

$$\boldsymbol{x}^s = \tfrac{1}{\sqrt{2}} \left( \boldsymbol{\mu}^{S_s} + \boldsymbol{\nu}^{C_s} \right). \tag{110}$$

All combinations of stimuli and context cues are permitted; the total number of trials and sensory inputs is thus $P = Q^2$. Each trial $s$ is thus specified by a stimulus and context index: $s = (S_s C_s)$. In contrast to the simple task, sensory input vectors are not orthogonal among each other; using *Equation 110*, we see that to the leading order in $N$,

$$\langle x_i^s x_i^{s'} \rangle_i \simeq \begin{cases} 1 & S_s = S_s' \text{ and } C_s = C_s' \ (s = s') \\ 1/2 & S_s = S_s' \text{ or } C_s = C_s' \\ 0 & \text{otherwise.} \end{cases} \tag{111}$$

The task is defined as follows. When the context cue $C$ ranges between 1 and $Q/2$, context takes value 1. In context 1, the first half of the $Q$ stimuli is associated with category A ($\tilde{z} = z^A$), and the second half with B ($\tilde{z} = z^B$). When the context cue $C$ ranges between $Q/2$ and $Q$, context takes value 2. In context 2, stimuli-category associations are reversed: the first half of the $Q$ stimuli is associated with category B ($\tilde{z} = z^B$), and the second half with A ($\tilde{z} = z^A$).

Correlations in the sensory inputs (*Equation 111*) are such that, for every value of $Q$, inputs are not linearly separable (*Barak et al., 2013*). For $Q = 2$, the task is equivalent to a classical XOR computation. We focus however on $Q > 2$, for which each context is signaled by more than one context cue. As in experimental work (*Wallis et al., 2001*; *Stoet and Snyder, 2004*; *Brincat et al., 2018*), this allows to dissociate the activity dependence on the abstract variable context from the sensory variable context cue (see *Equation 122* and *Equation 123* in Section Context-dependent task: category and context selectivity).

We start by writing down explicit expressions for the activity (*Equation 29*) in the current task (Section Context-dependent task: computing activity). We then derive the expressions that quantify how activity measures, such as selectivity and correlations, evolve over learning (Sections Context-dependent task: category and context selectivity, Context-dependent task: category and context correlation and Context-dependent task: computing normalized dot products). These expressions are rather complex, and require numerical evaluation. To gain further mathematical insight, in Sections Detailed analysis of context selectivity, Detailed analysis of category selectivity and Analysis of patterns of context and category selectivity we consider specific cases and quantities, and derive their behaviour analytically.

## Context-dependent task: computing activity

We start by computing the value of coordinates $c^s$, which are solution to the linear system in *Equation 26*. As in Section Simple task: computing activity (see also Section Characterizing variability), we neglect the variable term $h_0^s$ in the left-hand side of that equation and, after a small amount of algebra, we find that it can be rewritten as

$$\Phi^{-1}(\tilde{z}^s) = c^s \left( \langle \Psi'^2 \rangle + \frac{\eta_w}{\eta_u} \left( \langle \Psi^2 \rangle - \langle \Psi \rangle^2 \right) \right) + \sum_{q \in N(s)} c^q \left( \frac{1}{2} \langle \Psi' \Psi' \rangle + \frac{\eta_w}{\eta_u} \left( \langle \Psi \Psi \rangle - \langle \Psi \rangle^2 \right) \right)$$
$$+ \sum_{q=1}^{P} c^q \frac{\eta_w}{\eta_u} \langle \Psi \rangle^2 \tag{112}$$

where we used the short-hand notation $N(s)$ to indicate the set of trials that are *neighbours* to $s$ (i.e., trials that have either the same stimulus or the same context cue of $s$). We have used the notation $\langle FF \rangle$ to indicate the average over the product of two nonlinear functions, $F$, whose arguments are given by two zero-mean and unit-variance Gaussian variables with covariance 1/2. That is,

$$\langle FF \rangle \equiv \langle F(a)F(b) \rangle_{a,b} \tag{113}$$

where both $a$ and $b$ are zero-mean, unit-variance Gaussian random variables with covariance 1/2. Detail on how these averages are computed numerically is given in Section Evaluation of averages (*Equation 186*).

As in the simple task (*Equation 46*), because the left-hand side can take on only two values, the coordinates $c^s$ can take on only two values,

$$c^s = \begin{cases} c^A & s \text{ in category A} \\ c^B & s \text{ in category B} . \end{cases} \tag{114}$$

The values of $c^A$ and $c^B$ are determined by the same linear system as in *Equation 47*, except now $\alpha$ and $\beta$ are given by

$$\alpha = \langle \Psi'^2 \rangle + \frac{\eta_w}{\eta_u}(\langle \Psi^2 \rangle - \langle \Psi \rangle^2) - 2\left[\frac{1}{2}\langle \Psi' \Psi' \rangle + \frac{\eta_w}{\eta_u}(\langle \Psi \Psi \rangle - \langle \Psi \rangle^2)\right] \tag{115a}$$

$$\beta = Q\left[\frac{1}{2}\langle \Psi' \Psi' \rangle + \frac{\eta_w}{\eta_u}(\langle \Psi \Psi \rangle - \langle \Psi \rangle^2)\right] + \frac{\eta_w}{\eta_u}\frac{Q^2}{2}\langle \Psi \rangle^2. \tag{115b}$$

To derive the expression above, we used the fact that every sensory input has $2Q - 2$ neighbours, of which $Q - 2$ are associated with the same category, and $Q$ with the opposite one. The final expression for $c^A$ and $c^B$ is thus given by *Equation 49a*; that expression depends on $\gamma$, which is given in *Equation 50*.

By comparing *Equation 115* with *Equation 48* we see that, with respect to the simple task, the expressions for $\alpha$ and $\beta$ include extra terms (shown in square brackets in the right-hand side of *Equation 115*). These arise because, unlike in the simple task, different inputs can be correlated (*Equation 111*). The extra term in the expression for $\beta$ (*Equation 115b*) scales with $Q$, while the extra term for $\alpha$ (*Equation 115a*) does not; this indicates the typical value of $\gamma$ (*Equation 50*), which is proportional to $\beta/\alpha$, is larger in this task than in the simple one. This in turn implies that the parameter region where one has approximately $c^A \simeq -c^B$ is larger in the current task than in the simple one; this approximation will later be used in Section Analysis of patterns of context and category selectivity. In the simple task, the parameter region where $c^A \simeq -c^B$ coincided with the region where category correlation were negative (*Equation 74*, Section Simple task: category correlation). This suggests that the parameter region where correlations are negative, also, is larger in this task than in the simple one. As it will be shown in Section Context-dependent task: category and context correlation, however, the expressions for correlations are much more complex in the current task than *Equation 74*; this hypothesis thus needs to be carefully verified – which is done, using numerical integration, in *Figure 6—figure supplement 2C*.

Since this task is an extension of the XOR task, sensory inputs are not linearly separable. This shows up as a singularity when the intermediate layer is linear (e.g., $\Psi(x) = x$). Indeed, in that case, the value of $\gamma$ (*Equation 50*) diverges, which in turn means both $c^A$ and $c^B$ diverge (*Equation 49*). That's because $\gamma$ is proportional to the ratio $\beta/\alpha$, and $\alpha$ vanishes, while $\beta$ does not. To see that $\alpha$ vanishes, we use *Equation 115a* to write

$$\begin{aligned}\alpha &= 1 + \frac{\eta_w}{\eta_u}(\langle \Psi^2 \rangle - \langle \Psi \rangle^2) - 2\left[\frac{1}{2} + \frac{\eta_w}{\eta_u}(\langle \Psi \Psi \rangle - \langle \Psi \rangle^2)\right] \\ &= \frac{\eta_w}{\eta_u}\left[\langle \Psi^2 \rangle - 2\langle \Psi \Psi \rangle + \langle \Psi \rangle^2\right] \\ &= \frac{\eta_w}{\eta_u}\left[1 - 2 \cdot \frac{1}{2} + 0\right] = 0.\end{aligned} \tag{116}$$

When the activation function $\Psi$ is nonlinear, instead, the values of $c^A$ and $c^B$ are finite; their magnitude depends on how close to linear $\Psi$ is in its effective activation range.

To conclude our characterization of activity, we evaluate spanning vectors, $\boldsymbol{v}^{qs}$, by combining *Equation 28* and *Equation 111*. Unlike in the simple task, for each activity vector, $\boldsymbol{y}^s$, there exists more than one spanning vector; those are given by $\boldsymbol{v}^{ss}$, and all vectors $\boldsymbol{v}^{qs}$ for which $q \in N(s)$. *Equation 29* thus reads

$$\begin{aligned}\boldsymbol{y}^s &= \boldsymbol{y}_0^s + c^s \boldsymbol{v}^{ss} + \sum_{q \in N(s)} c^q \boldsymbol{v}^{qs} \\ &= \boldsymbol{y}_0^s + c^s \boldsymbol{w}_0 \odot \Psi'(\boldsymbol{k}_0^s) \odot \Psi'(\boldsymbol{k}_0^s) + \frac{1}{2}\sum_{q \in N(s)} c^q \boldsymbol{w}_0 \odot \Psi'(\boldsymbol{k}_0^q) \odot \Psi'(\boldsymbol{k}_0^s)\end{aligned} \tag{117}$$

where the second line follows from *Equation 28* and the coordinates $c^q$ take values $c^A$ or $c^B$ depending on the category $\boldsymbol{x}^q$ is associated with (*Equation 114*). Using the notation $s = (S_s C_s)$, *Equation 117* can also be written in the compact form

$$\boldsymbol{y}^{S_s C_s} = \boldsymbol{y}_0^{S_s C_s} + \frac{1}{2}\sum_{S=1}^{Q} c^{S C_s} \boldsymbol{w}_0 \odot \Psi'^{S C_s} \odot \Psi'^{S_s C_s} + \frac{1}{2}\sum_{C=1}^{Q} c^{S_s C} \boldsymbol{w}_0 \odot \Psi'^{S_s C} \odot \Psi'^{S_s C_s} \tag{118}$$

where we used the short-hand notation $\Psi'^{SC} \equiv \Psi'(\boldsymbol{k}_0^{SC})$.

To isolate the effect of the nonlinearity $\Psi$, it will be instructive (see Sections Detailed analysis of context selectivity and Detailed analysis of category selectivity) to also compute the synaptic drive, $\boldsymbol{k}^s$, after learning. Using **Equation 5b** and **Equation 18a**, it is easy to see that

$$\boldsymbol{k}^s \equiv \boldsymbol{k}_0^s + \Delta \boldsymbol{k}^s = \boldsymbol{k}_0^s + c^s \boldsymbol{w}_0 \odot \Psi'(\boldsymbol{k}_0^s) + \frac{1}{2} \sum_{q \in N(s)} c^q \boldsymbol{w}_0 \odot \Psi'(\boldsymbol{k}_0^q), \tag{119}$$

or, equivalently,

$$\boldsymbol{k}^{S_s C_s} = \boldsymbol{k}_0^{S_s C_s} + \frac{1}{2} \sum_{S=1}^{Q} c^{SC_s} \boldsymbol{w}_0 \odot \Psi'^{SC_s} + \frac{1}{2} \sum_{C=1}^{Q} c^{S_s C} \boldsymbol{w}_0 \odot \Psi'^{S_s C}. \tag{120}$$

We conclude with a remark on the geometry of the spanning vectors, $\boldsymbol{v}^{qs}$. As in the simple task, those include a component that is aligned with the initial readout vector, $\boldsymbol{w}_0$, and a residual component that is perpendicular to it, $\delta \boldsymbol{v}^{qs}$ (**Equation 31**). In the simple task, residual components could be neglected (**Equation 55**) because they were orthogonal to each other, and did not contribute to novel activity structure. In this task, residual components are not, in general, orthogonal to each other, and thus cannot be neglected. In fact, we have

$$\delta \boldsymbol{v}^{qs} \cdot \delta \boldsymbol{v}^{q's'} = \langle x_i^q x_i^s \rangle_i \langle x_i^{q'} x_i^{s'} \rangle_i \left[ \langle \Psi'(k_{0,i}^q) \Psi'(k_{0,i}^s) \Psi'(k_{0,i}^{q'}) \Psi'(k_{0,i}^{s'}) \rangle_i - \langle \Psi'(k_{0,i}^q) \Psi'(k_{0,i}^s) \rangle_i \langle \Psi'(k_{0,i}^{q'}) \Psi'(k_{0,i}^{s'}) \rangle_i \right]. \tag{121}$$

The term in the right-hand side can be non-zero even when $sq$ are different from $s'q'$; this is due to **Equation 43b** and **Equation 111**, which imply that $k_{0,i}$ variables can be correlated among each other. The fact that residuals $\delta \boldsymbol{v}^{qs}$ cannot be neglected implies that activity evolution is not effectively one-dimensional, as it was the simple task, but higher-dimensional (this is evident in the PC plots in **Figure 6—figure supplement 1C, D**). All the directions along which activity evolve are, however, correlated with the initial readout vector $\boldsymbol{w}_0$ (**Equation 30**).

## Context-dependent task: category and context selectivity

In the present task, we can compute category, as well as context selectivity. In analogy with category selectivity, **Equation 56**, context selectivity is defined as

$$S_i^{\text{ctx}} = \frac{\langle (y_i^s - y_i^{s'})^2 \rangle_{s,s' \text{ diff ctx}} - \langle (y_i^s - y_i^{s'})^2 \rangle_{s \neq s' \text{ same ctx, diff ctx cue}}}{\langle (y_i^s - y_i^{s'})^2 \rangle_{s,s' \text{ diff ctx}} + \langle (y_i^s - y_i^{s'})^2 \rangle_{s \neq s' \text{ same ctx, diff ctx cue}}}. \tag{122}$$

Note that, in the average over pairs of trials from the same context, we excluded pairs of trials with the same context cue. This was done to exclude the possibility that context selectivity increases simply because activity in response to the same context cue become more similar over learning. For completeness, we also compute

$$S_i^{\text{ctx},2} = \frac{\langle (y_i^s - y_i^{s'})^2 \rangle_{s,s' \text{ diff ctx}} - \langle (y_i^s - y_i^{s'})^2 \rangle_{s \neq s' \text{ same ctx}}}{\langle (y_i^s - y_i^{s'})^2 \rangle_{s,s' \text{ diff ctx}} + \langle (y_i^s - y_i^{s'})^2 \rangle_{s \neq s' \text{ same ctx}}}, \tag{123}$$

which we plot in **Figure 6—figure supplement 1A, B**. Those plots show that the behaviour under this definition is similar to that of **Equation 122**.

We are interested in deriving theoretical expressions for average category and context selectivity, obtained by averaging **Equation 56** and **Equation 122** (or **Equation 123**) over $i$. For the present task, that is hard. Consequently, we use results from the simple task (Section Simple task: category selectivity) which indicated that, in the limit $N \gg Q \gg 1$, average category selectivity can be approximated with the category clustering measure, **Equation 67**; the latter is equivalent to separately averaging the numerator and denominator of selectivity over neurons.

For category, clustering is the same as in the simple task, **Equation 67** and **Equation 68**, which we repeat here for convenience,

$$
\begin{aligned}
\tilde{S}^{\text{cat}} &= \frac{\langle \langle (y_i^s - y_i^{s'})^2 \rangle_i \rangle_{s,s' \text{ diff cat}} - \langle \langle (y_i^s - y_i^{s'})^2 \rangle_i \rangle_{s \neq s' \text{ same cat}}}{\langle \langle (y_i^s - y_i^{s'})^2 \rangle_i \rangle_{s,s' \text{ diff cat}} + \langle \langle (y_i^s - y_i^{s'})^2 \rangle_i \rangle_{s \neq s' \text{ same cat}}} \\
&= \frac{-\langle \langle y_i^s y_i^{s'} \rangle_i \rangle_{s,s' \text{ diff cat}} + \langle \langle y_i^s y_i^{s'} \rangle_i \rangle_{s \neq s' \text{ same cat}}}{2 \langle \langle (y_i^s)^2 \rangle_i \rangle_s - \langle \langle y_i^s y_i^{s'} \rangle_i \rangle_{s,s' \text{ diff cat}} - \langle \langle y_i^s y_i^{s'} \rangle_i \rangle_{s \neq s' \text{ same cat}}}
\end{aligned}
\tag{124}
$$

where we used the statistical homogeneity of activity vectors. Similarly, for context selectivity, we may write

$$\tilde{S}^{\text{ctx}} = \frac{\langle\langle(y_i^s - y_i^{s'})^2\rangle_i\rangle_{s,s' \text{ diff ctx}} - \langle\langle(y_i^s - y_i^{s'})^2\rangle_i\rangle_{s\neq s' \text{ same ctx, diff ctx cue}}}{\langle\langle(y_i^s - y_i^{s'})^2\rangle_i\rangle_{s,s' \text{ diff ctx}} + \langle\langle(y_i^s - y_i^{s'})^2\rangle_i\rangle_{s,s' \text{ same ctx, diff ctx cue}}}$$
$$= \frac{-\langle\langle y_i^s y_i^{s'}\rangle_i\rangle_{s,s' \text{ diff ctx}} + \langle\langle y_i^s y_i^{s'}\rangle_i\rangle_{s\neq s' \text{ same ctx, diff ctx cue}}}{2\langle\langle(y_i^s)^2\rangle_i\rangle_s - \langle\langle y_i^s y_i^{s'}\rangle_i\rangle_{s,s' \text{ diff ctx}} - \langle\langle y_i^s y_i^{s'}\rangle_i\rangle_{s,s' \text{ same ctx, diff ctx cue}}}$$

(125)

and

$$\tilde{S}^{\text{ctx},2} = \frac{\langle\langle(y_i^s - y_i^{s'})^2\rangle_i\rangle_{s,s' \text{ diff ctx}} - \langle\langle(y_i^s - y_i^{s'})^2\rangle_i\rangle_{s\neq s' \text{ same ctx}}}{\langle\langle(y_i^s - y_i^{s'})^2\rangle_i\rangle_{s,s' \text{ diff ctx}} + \langle\langle(y_i^s - y_i^{s'})^2\rangle_i\rangle_{s\neq s' \text{ same ctx}}}$$
$$= \frac{-\langle\langle y_i^s y_i^{s'}\rangle_i\rangle_{s,s' \text{ diff ctx}} + \langle\langle y_i^s y_i^{s'}\rangle_i\rangle_{s\neq s' \text{ same ctx}}}{2\langle\langle(y_i^s)^2\rangle_i\rangle_s - \langle\langle y_i^s y_i^{s'}\rangle_i\rangle_{s,s' \text{ diff ctx}} - \langle\langle y_i^s y_i^{s'}\rangle_i\rangle_{s\neq s' \text{ same ctx}}}.$$

(126)

To evaluate those expressions, we need the normalized dot products over activity, $\langle y_i^s y_i^{s'}\rangle_i$. These are computed in Section Context-dependent task: computing normalized dot products. Finally, averages over trials are performed numerically. The resulting theoretical estimates for $\tilde{S}^{\text{cat}}$ and $\tilde{S}^{\text{ctx}}$ are shown in *Figure 6—figure supplement 1B* and *Figure 7—figure supplement 1*.

In *Figure 7—figure supplement 1A–C*, we compare theoretical estimates with simulations. Agreement is relatively good, although it is worse than for the simple task; as argued in Section Evolution of activity in finite-size networks, that is expected. Note that the values of average selectivity and clustering are not close (this is only verified in the $N \gg Q \gg 1$ limit, and would require values of $N$ larger than those used in simulations); the qualitative behaviour of the two quantities is, however, identical. In *Figure 6—figure supplement 1B*, we plot the theoretical estimates across a broad range of task and circuit parameters. These theoretical estimates indicate that, in all cases, category (*Equation 124*) and context (*Equation 125*, *Equation 126*) selectivity increase. This is in agreement with simulations, which are reported in *Figure 6—figure supplement 1A*.

## Context-dependent task: category and context correlation

To quantify how the population as a whole encodes category and context, we evaluate category and context correlations. Those quantities, denoted $C^{\text{cat}}$ and $C^{\text{ctx}}$, are given by the average Pearson correlation coefficient for trials in different categories and contexts. $C^{\text{cat}}$ is defined as in *Equation 71*. Similarly, $C^{\text{ctx}}$ is defined as

$$C^{\text{ctx}} = \langle C^{s_1 s_2}\rangle_{s_1 s_2}$$

(127)

where $s_1$ and $s_2$ are indices that denote, respectively, trials from contexts 1 and 2. Similar to *Equation 72*, the Pearson correlation coefficient $C^{s_1 s_2}$ is given by

$$C^{s_1 s_2} = \frac{\langle y_i^{s_1} y_i^{s_2}\rangle_i - \langle\Psi\rangle^2}{\sqrt{\langle y_i^{s_1} y_i^{s_1}\rangle_i - \langle\Psi\rangle^2}\sqrt{\langle y_i^{s_2} y_i^{s_2}\rangle_i - \langle\Psi\rangle^2}}.$$

(128)

To evaluate these expressions, we use the normalized dot products $\langle y_i^s y_i^{s'}\rangle_i$ that are computed in Section Context-dependent task: computing normalized dot products. Averaging over trials is, finally, done numerically.

For completeness, we also consider the alternative definition of correlations, where activity is averaged over trials first, and then the Pearson correlation is computed. The alternative definition for category correlation is identical to *Equation 79*. The alternative definition for context correlation is given by

$$C^{\text{ctx}} = \frac{\langle y_i^1 y_i^2\rangle_i - \langle y_i^1\rangle_i\langle y_i^2\rangle_i}{\sqrt{\langle y_i^1 y_i^1\rangle_i - \langle y_i^1\rangle_i^2}\sqrt{\langle y_i^2 y_i^2\rangle_i - \langle y_i^2\rangle_i^2}}$$
$$= \frac{\langle y_i^1 y_i^2\rangle_i - \langle\Psi\rangle^2}{\sqrt{\langle y_i^1 y_i^1\rangle_i - \langle\Psi\rangle^2}\sqrt{\langle y_i^2 y_i^2\rangle_i - \langle\Psi\rangle^2}}$$

(129)

where we have defined

$$\boldsymbol{y}^1 = \langle \boldsymbol{y}^{s_1} \rangle_{s_1}$$
$$\boldsymbol{y}^2 = \langle \boldsymbol{y}^{s_2} \rangle_{s_2}. \tag{130}$$

For the current task, there exists no simple mathematical relationship between correlations obtained from the standard, and the alternative definition. We thus checked numerically the behaviour of both quantities; results are reported in **Figure 6—figure supplement 1B**. As in the simple task, we found that the qualitative behaviour of both quantities is not fixed, but depends on task and circuit parameters. This is in agreement with simulations, which are illustrated in **Figure 6—figure supplement 1A**.

## Context-dependent task: computing normalized dot products

To conclude, we illustrate how normalized dot products, **Equation 84**, are computed for the current task. We start from **Equation 91**, which we repeat here for completeness,

$$\langle y_i^s y_i^{s'} \rangle_i = \langle y_{0,i}^s y_{0,i}^{s'} \rangle_i + \langle \Delta y_i^s \Delta y_i^{s'} \rangle_i. \tag{131}$$

The first term of the right-hand side reads

$$\langle y_{0,i}^s y_{0,i}^{s'} \rangle_i = \begin{cases} \langle \Psi^2 \rangle & S_s = S_s' \text{ and } C_s = C_s' \ (s = s') \\ \langle \Psi \Psi \rangle & S_s = S_s' \text{ or } C_s = C_s' \\ \langle \Psi \rangle^2 & \text{otherwise,} \end{cases} \tag{132}$$

where we used **Equation 5** together with **Equation 43b** and **Equation 111**. Using **Equation 29** together with **Equation 28**, the second term of the right-hand side of **Equation 131** reads

$$\langle \Delta y_i^s \Delta y_i^{s'} \rangle_i = \frac{1}{N} \sum_{q=1}^{P} \sum_{q'=1}^{P} c^q c^{q'} \langle x_i^q x_i^s \rangle_i \langle x_i^{q'} x_i^{s'} \rangle_i \langle \Psi'(k_{0,i}^q) \Psi'(k_{0,i}^s) \Psi'(k_{0,i}^{q'}) \Psi'(k_{0,i}^{s'}) \rangle_i \tag{133}$$

where sensory input correlations, $\langle x_i^q x_i^s \rangle_i$, are given in **Equation 111**.

Because $\langle x_i^q x_i^s \rangle_i$ can be non-zero even when $s \neq q$ (**Equation 111**), the number of non-zero terms in the sum in **Equation 133** is, in general, large. Each term contains an average, $\langle \Psi'(k_{0,i}^q) \Psi'(k_{0,i}^s) \Psi'(k_{0,i}^{q'}) \Psi'(k_{0,i}^{s'}) \rangle_i$, that includes four nonlinear functions. The value of those averages is specified by the correlations among the arguments, $k_{0,i}$, which in turn depend on the values of $s$, $q$, $s'$ and $q'$ (**Equation 111**, via **Equation 43b**). Averages are evaluated numerically; detail on how this is done is given in Section Evaluation of averages.

This procedure yields a set of normalized dot products that can be used to evaluate, numerically, the expressions for activity selectivity and correlation derived in Sections Context-dependent task: category and context selectivity and Context-dependent task: category and context correlation. As we rely on numerics, the results we obtain in this way are hard to interpret. For this reason, in the next sections we focus on specific cases were results can be obtained analytically; this allows us to extract a more intuitive understanding of how activity measures evolve over learning.

## Detailed analysis of context selectivity

We start clarifying how context selectivity increases over learning. Results from simulations, and numerical integration of **Equation 125**, indicate that context selectivity increases for the synaptic drive, $\boldsymbol{k}^s$; this increase is then reflected in the activity, $\boldsymbol{y}^s$ (**Figure 6—figure supplement 1A, B** and **Figure 7—figure supplement 1B**). In this section, we analyze the behaviour of context selectivity for the synaptic drive. Focussing on the synaptic drive, instead of activity, allows us to derive results analytically. In the following, we start from **Equation 125** and show that, for the synaptic drive $\boldsymbol{k}^s$, the value of $\tilde{S}^{\text{ctx}}$ increases over learning. At the end of this section, we comment on the insights provided by such derivation.

We start by simplifying the sums over trials contained in **Equation 125**, which involve pairs of trials $ss'$ from the same, or different context. To this end we observe that, because of task symmetries, these sums involve a large number of identical terms; for example, the term with $s = (11)$ and $s' = (12)$ is identical to $s = (21)$ and $s' = (22)$ (both pairs of trials are neighbours, and are associated with the same category). We thus perform averages over a reduced, and less redundant subset of pairs of trials.

First, we consider only two values of $s$: for concreteness, we take $s = (11)$ and $s = ((\hat{Q}+1)1)$, where we defined

$$\hat{Q} \equiv \frac{Q}{2}. \tag{134}$$

These $s$ trials are associated, respectively, with categories A and B. Second, for each value of $s$, we consider $s'$ trials with context cue equal to $C = 2$ and $C = \hat{Q} + 1$; these are associated, respectively, with context 1 and 2 (note that $C = 1$ must be avoided, as trials with the same context cue must be excluded, see *Equation 125*). This allows us to rewrite the averages contained in *Equation 125* as

$$\langle\langle(k_i^s - k_i^{s'})^2\rangle_i\rangle_{s,s' \text{ same ctx, diff ctx cue}} = \frac{1}{2Q}\left[\sum_{\bar{S}=1}^{Q}\langle(k_i^{11} - k_i^{\bar{S}2})^2\rangle_i + \sum_{\bar{S}=1}^{Q}\langle(k_i^{(\hat{Q}+1)1} - k_i^{\bar{S}2})^2\rangle_i\right] \tag{135a}$$

$$\langle\langle(k_i^s - k_i^{s'})^2\rangle_i\rangle_{s,s' \text{ diff ctx}} = \frac{1}{2Q}\left[\sum_{\bar{S}=1}^{Q}\langle(k_i^{11} - k_i^{\bar{S}(\hat{Q}+1)})^2\rangle_i + \sum_{\bar{S}=1}^{Q}\langle(k_i^{(\hat{Q}+1)1} - k_i^{\bar{S}(\hat{Q}+1)})^2\rangle_i\right]. \tag{135b}$$

The sums over $\bar{S}$ can further be simplified. By using again symmetries, we have:

$$\sum_{\bar{S}=1}^{Q}\langle(k_i^{11} - k_i^{\bar{S}2})^2\rangle_i = \langle(k_i^{11} - k_i^{12})^2\rangle_i + (\hat{Q}-1)\langle(k_i^{11} - k_i^{22})^2\rangle_i + \hat{Q}\langle(k_i^{11} - k_i^{(\hat{Q}+1)2})^2\rangle_i. \tag{136}$$

We can do the same for the other sums, yielding:

$$\sum_{\bar{S}=1}^{Q}\langle(k_i^{(\hat{Q}+1)1} - k_i^{\bar{S}2})^2\rangle_i = \langle(k_i^{(\hat{Q}+1)1} - k_i^{(\hat{Q}+1)2})^2\rangle_i + (\hat{Q}-1)\langle(k_i^{(\hat{Q}+1)1} - k_i^{(\hat{Q}+2)2})^2\rangle_i + \hat{Q}\langle(k_i^{(\hat{Q}+1)1} - k_i^{12})^2\rangle_i \tag{137a}$$

$$\sum_{\bar{S}=1}^{Q}\langle(k_i^{11} - k_i^{\bar{S}(\hat{Q}+1)})^2\rangle_i = \langle(k_i^{11} - k_i^{1(\hat{Q}+1)})^2\rangle_i + (\hat{Q}-1)\langle(k_i^{11} - k_i^{2(\hat{Q}+1)})^2\rangle_i + \hat{Q}\langle(k_i^{11} - k_i^{(\hat{Q}+1)(\hat{Q}+1)})^2\rangle_i \tag{137b}$$

$$\sum_{\bar{S}=1}^{Q}\langle(k_i^{(\hat{Q}+1)1} - k_i^{\bar{S}(\hat{Q}+1)})^2\rangle_i = \langle(k_i^{(\hat{Q}+1)1} - k_i^{(\hat{Q}+1)(\hat{Q}+1)})^2\rangle_i + (\hat{Q}-1)\langle(k_i^{(\hat{Q}+1)1} - k_i^{(\hat{Q}+2)(\hat{Q}+1)})^2\rangle_i + \hat{Q}\langle(k_i^{(\hat{Q}+1)1} - k_i^{1(\hat{Q}+1)})^2\rangle_i. \tag{137c}$$

It is easy to verify that, before learning starts, the right-hand sides of *Equation 135a* and *Equation 135b* are identical. This implies that the initial value of context selectivity, *Equation 125*, vanishes (*Figure 7—figure supplement 1B*). To show that context selectivity increases over learning, we thus need to show that the numerator of *Equation 125* becomes positive over learning. This is equivalent to show that *Equation 135a* is smaller than *Equation 135b*. Using *Equation 136* and *Equation 137*, this condition can be rewritten as

$$\hat{Q}\left[\langle(k_i^{11} - k_i^{22})^2\rangle_i + \langle(k_i^{11} - k_i^{(\hat{Q}+1)2})^2\rangle_i + \langle(k_i^{(\hat{Q}+1)1} - k_i^{(\hat{Q}+2)2})^2\rangle_i + \langle(k_i^{(\hat{Q}+1)1} - k_i^{12})^2\rangle_i\right] +$$
$$\langle(k_i^{11} - k_i^{12})^2\rangle_i + \langle(k_i^{(\hat{Q}+1)1} - k_i^{(\hat{Q}+1)2})^2\rangle_i - \langle(k_i^{11} - k_i^{22})^2\rangle_i - \langle(k_i^{(\hat{Q}+1)1} - k_i^{(\hat{Q}+2)2})^2\rangle_i <$$
$$\hat{Q}\left[\langle(k_i^{11} - k_i^{2(\hat{Q}+1)})^2\rangle_i + \langle(k_i^{11} - k_i^{(\hat{Q}+1)(\hat{Q}+1)})^2\rangle_i + \langle(k_i^{(\hat{Q}+1)1} - k_i^{(\hat{Q}+2)(\hat{Q}+1)})^2\rangle_i + \langle(k_i^{(\hat{Q}+1)1} - k_i^{1(\hat{Q}+1)})^2\rangle_i\right] \cdot$$
$$\langle(k_i^{11} - k_i^{1(\hat{Q}+1)})^2\rangle_i + \langle(k_i^{(\hat{Q}+1)1} - k_i^{(\hat{Q}+1)(\hat{Q}+1)})^2\rangle_i - \langle(k_i^{11} - k_i^{2(\hat{Q}+1)})^2\rangle_i - \langle(k_i^{(\hat{Q}+1)1} - k_i^{(\hat{Q}+2)(\hat{Q}+1)})^2\rangle_i. \tag{138}$$

We now use *Equation 119* to write

$$\langle(k_i^s - k_i^{s'})^2\rangle_i = \langle(k_{0,i}^s)^2\rangle_i + \langle(k_{0,i}^{s'})^2\rangle_i - 2\langle k_{0,i}^s k_{0,i}^{s'}\rangle_i + \langle(\Delta k_i^s)^2\rangle_i + \langle(\Delta k_i^{s'})^2\rangle_i - 2\langle\Delta k_i^s \Delta k_i^{s'}\rangle_i \tag{139}$$

where the terms containing the cross-products between $k_i^s$ and $\Delta k_i^s$ vanish on average because of *Equation 8a*. By using the statistical homogeneity of activity across contexts, we can rewrite *Equation 138* as

$$
\begin{aligned}
&\hat{Q}\left[\langle\Delta k_i^{11}\Delta k_i^{22}\rangle_i + \langle\Delta k_i^{11}\Delta k_i^{(\hat{Q}+1)2}\rangle_i + \langle\Delta k_i^{(\hat{Q}+1)1}\Delta k_i^{(\hat{Q}+2)2}\rangle_i + \langle\Delta k_i^{(\hat{Q}+1)1}\Delta k_i^{12}\rangle_i\right] + \\
&\langle\Delta k_i^{11}\Delta k_i^{12}\rangle_i + \langle\Delta k_i^{(\hat{Q}+1)1}\Delta k_i^{(\hat{Q}+1)2}\rangle_i - \langle\Delta k_i^{11}\Delta k_i^{22}\rangle_i - \langle\Delta k_i^{(\hat{Q}+1)1}\Delta K_i^{(\hat{Q}+2)2}\rangle_i > \\
&\hat{Q}\left[\langle\Delta k_i^{11}\Delta k_i^{2(\hat{Q}+1)}\rangle_i + \langle\Delta k_i^{11}\Delta k_i^{(\hat{Q}+1)(\hat{Q}+1)}\rangle_i + \langle\Delta k_i^{(\hat{Q}+1)1}\Delta k_i^{(\hat{Q}+2)(\hat{Q}+1)}\rangle_i + \langle\Delta k_i^{(\hat{Q}+1)1}\Delta k_i^{1(\hat{Q}+1)}\rangle_i\right] + \\
&\langle\Delta k_i^{11}\Delta k_i^{1(\hat{Q}+1)}\rangle_i + \langle\Delta k_i^{(\hat{Q}+1)1}\Delta k_i^{(\hat{Q}+1)(\hat{Q}+1)}\rangle_i - \langle\Delta k_i^{11}\Delta k_i^{2(\hat{Q}+1)}\rangle_i - \langle\Delta k_i^{(\hat{Q}+1)1}\Delta k_i^{(\hat{Q}+2)(\hat{Q}+1)}\rangle_i
\end{aligned}
\tag{140}
$$

or, re-arranging terms,

$$
\begin{aligned}
&\hat{Q}\left\{\left[\langle\Delta k_i^{11}\Delta k_i^{22}\rangle_i + \langle\Delta k_i^{11}\Delta k_i^{(\hat{Q}+1)2}\rangle_i + \langle\Delta k_i^{(\hat{Q}+1)1}\Delta k_i^{(\hat{Q}+2)2}\rangle_i + \langle\Delta k_i^{(\hat{Q}+1)1}\Delta k_i^{12}\rangle_i\right] - \right.\\
&\left.\left[\langle\Delta k_i^{11}\Delta k_i^{2(\hat{Q}+1)}\rangle_i + \langle\Delta k_i^{11}\Delta k_i^{(\hat{Q}+1)(\hat{Q}+1)}\rangle_i + \langle\Delta k_i^{(\hat{Q}+1)1}\Delta k_i^{(\hat{Q}+2)(\hat{Q}+1)}\rangle_i + \langle\Delta k_i^{(\hat{Q}+1)1}\Delta k_i^{1(\hat{Q}+1)}\rangle_i\right]\right\} + \\
&\left\{\left[\langle\Delta k_i^{11}\Delta k_i^{12}\rangle_i + \langle\Delta k_i^{(\hat{Q}+1)1}\Delta k_i^{(\hat{Q}+1)2}\rangle_i - \langle\Delta k_i^{11}\Delta k_i^{22}\rangle_i - \langle\Delta k_i^{(\hat{Q}+1)1}\Delta k_i^{(\hat{Q}+2)2}\rangle_i\right] - \right.\\
&\left.\left[\langle\Delta k_i^{11}\Delta k_i^{1(\hat{Q}+1)}\rangle_i + \langle\Delta k_i^{(\hat{Q}+1)1}\Delta k_i^{(\hat{Q}+1)(\hat{Q}+1)}\rangle_i - \langle\Delta k_i^{11}\Delta k_i^{2(\hat{Q}+1)}\rangle_i - \langle\Delta k_i^{(\hat{Q}+1)1}\Delta k_i^{(\hat{Q}+2)(\hat{Q}+1)}\rangle_i\right]\right\} > 0.
\end{aligned}
\tag{141}
$$

To show that context selectivity increases over learning, we need to verify that the equation above holds. To this end, we evaluate analytically the normalized dot products $\langle\Delta k_i^{SC}\Delta k_i^{S'C'}\rangle_i$ for each pair of trials involved. This is done in the next paragraph; here we simply use those results (*Equation 156*, *Equation 157*, *Equation 158*, and *Equation 159*).

We start evaluating the difference within the first set of curly parenthesis of *Equation 141*, which correspond to the dominant contribution in $Q$. By using *Equation 159*, we see that this can be rewritten as

$$
\left[\langle\Delta k_i^{11}\Delta k_i^{22}\rangle_i + \langle\Delta k_i^{(\hat{Q}+1)1}\Delta k_i^{(\hat{Q}+2)2}\rangle_i\right] - \left[\langle\Delta k_i^{11}\Delta k_i^{(\hat{Q}+1)(\hat{Q}+1)}\rangle_i + \langle\Delta k_i^{(\hat{Q}+1)1}\Delta k_i^{1(\hat{Q}+1)}\rangle_i\right].
\tag{142}
$$

Using *Equation 158*, this becomes

$$
\frac{(c^A-c^B)^2}{N}[\mathcal{A}_1 - \mathcal{A}_2].
\tag{143}
$$

We then evaluate the difference within the second set of curly parenthesis. Using *Equation 156*, *Equation 157*, *Equation 158* and *Equation 159*, it is straightforward to see that that difference vanishes. Putting results together, our condition to verify (*Equation 141*) becomes simply:

$$
(c^A - c^B)^2[\mathcal{A}_1 - \mathcal{A}_2] > 0,
\tag{144}
$$

which is satisfied whenever $\mathcal{A}_1 - \mathcal{A}_2 > 0$. This is always verified, as from *Equation 150* and *Equation 151* we have

$$
\mathcal{A}_1 = \sum_{S+,S'_+}\langle S+1, S'_+2\rangle = \frac{Q}{2}\langle\Psi'\Psi'\rangle + \frac{Q}{2}\left(\frac{Q}{2} - 1\right)\langle\Psi'\rangle^2
\tag{145}
$$

while

$$
\mathcal{A}_2 = \sum_{S+,S'_-}\langle S+1, S'_-2\rangle = \left(\frac{Q}{2}\right)^2\langle\Psi'\rangle^2
\tag{146}
$$

so that

$$
\mathcal{A}_1 - \mathcal{A}_2 = \frac{Q}{2}(\langle\Psi'\Psi'\rangle - \langle\Psi'\rangle^2) > 0
\tag{147}
$$

which concludes our derivation. We remark that *Equation 147* vanishes when $\Psi$ is linear. This indicates that, even if context selectivity also increases for synaptic drives (which are a linear transformation of the sensory inputs), this phenomenon is due to the nonlinearity of activation functions.

## Computing normalized dot products

We now compute the normalized dot product expressions, $\langle \Delta k_i^{SC} \Delta k_i^{S'C'} \rangle_i$, for each pair of trials involved in *Equation 141*. We illustrate in detail how one example dot product, $\langle \Delta k_i^{11} \Delta k_i^{12} \rangle_i$, is computed. Other expressions are computed in a similar way; results are given below (*Equation 157*, *Equation 158* and *Equation 159*).

We start from:

$$\langle \Delta k_i^{11} \Delta k_i^{12} \rangle_i = \frac{1}{4N} \left[ \sum_{S,S'} c^{S1} c^{S'2} \langle \Psi_i'^{S1} \Psi_i'^{S'2} \rangle_i + \sum_{C,C'} c^{1C} c^{1C'} \langle \Psi_i'^{1C} \Psi_i'^{1C'} \rangle_i \right.$$
$$\left. + \sum_{S,C'} c^{S1} c^{1C'} \langle \Psi_i'^{S1} \Psi_i'^{1C'} \rangle_i + \sum_{C,S'} c^{1C} c^{S'2} \langle \Psi_i'^{1C} \Psi_i'^{S'2} \rangle_i \right] \tag{148}$$

which was derived from *Equation 120* together with *Equation 8a*. We then rewrite the sums in the right-hand side by expanding each index in two set of indices: one running from 1 to $Q/2$ (denoted by the subscript +), and one running from $Q/2 + 1$ to $Q$ (denoted by the subscript −). The first sum in *Equation 148* becomes:

$$\sum_{S,S'} c^{S1} c^{S'2} \langle \Psi_i'^{S1} \Psi_i'^{S'2} \rangle_i = c^{A^2} \sum_{S_+, S'_+} \langle S_+ 1, S'_+ 2 \rangle + c^A c^B \sum_{S_+, S'_-} \langle S_+ 1, S'_- 2 \rangle + c^A c^B \sum_{S_-, S'_+} \langle S_- 1, S'_+ 2 \rangle + c^{B^2} \sum_{S_-, S'_-} \langle S_- 1, S'_- 2 \rangle \tag{149}$$

where we have used the short-hand notation $\langle SC, S'C' \rangle \equiv \langle \Psi_i'^{SC} \Psi_i'^{S'C'} \rangle_i$. We now observe that

$$\sum_{S_+, S'_+} \langle S_+ 1, S'_+ 2 \rangle = \sum_{S_-, S'_-} \langle S_- 1, S'_- 2 \rangle \equiv \mathcal{A}_1 \tag{150}$$

while

$$\sum_{S_+, S'_-} \langle S_+ 1, S'_- 2 \rangle = \sum_{S_-, S'_+} \langle S_- 1, S'_+ 2 \rangle \equiv \mathcal{A}_2 \tag{151}$$

so that

$$\sum_{S,S'} c^{S1} c^{S'2} \langle \Psi_i'^{S1} \Psi_i'^{S'2} \rangle_i = (c^{A^2} + c^{B^2}) \mathcal{A}_1 + 2 c^A c^B \mathcal{A}_2. \tag{152}$$

The second sum in *Equation 148* gives:

$$\sum_{C,C'} c^{1C} c^{1C'} \langle \Psi_i'^{1C} \Psi_i'^{1C'} \rangle_i = c^{A^2} \sum_{C_+, C'_+} \langle 1C_+, 1C'_+ \rangle + c^A c^B \sum_{C_+, C'_-} \langle 1C_+, 1C'_- \rangle +$$
$$c^A c^B \sum_{C_-, C'_+} \langle 1C_-, 1C'_+ \rangle + c^{B^2} \sum_{C_-, C'_-} \langle 1C_-, 1C'_- \rangle \tag{153}$$
$$\equiv (c^{A^2} + c^{B^2}) \mathcal{A}_3 + 2 c^A c^B \mathcal{A}_4$$

by appropriately defining $\mathcal{A}_3$ and $\mathcal{A}_4$. The third sum gives:

$$\sum_{S,C'} c^{S1} c^{1C'} \langle \Psi_i'^{S1} \Psi_i'^{1C'} \rangle_i = c^{A^2} \sum_{S_+, C'_+} \langle S_+ 1, 1C_+ \rangle + c^A c^B \sum_{S_+, C'_-} \langle S_+ 1, 1C_- \rangle +$$
$$c^A c^B \sum_{S_-, C'_+} \langle S_- 1, 1C_+ \rangle + c^{B^2} \sum_{S_-, C'_-} \langle S_- 1, 1C_- \rangle \tag{154}$$
$$\equiv c^{A^2} \mathcal{A}_5 + c^{B^2} \mathcal{A}_6 + 2 c^A c^B \mathcal{A}_7$$

and, similarly, the fourth one:

$$\sum_{C,S'} c^{1C} c^{S'2} \langle \Psi_i'^{1C} \Psi_i'^{S2} \rangle_i = c^{A^2} \mathcal{A}_5 + c^{B^2} \mathcal{A}_6 + 2 c^A c^B \mathcal{A}_7. \tag{155}$$

By putting those results together, we conclude that

$$\langle \Delta k_i^{11} \Delta k_i^{12} \rangle_i = \frac{1}{4N} \left[ (c^{A^2} + c^{B^2}) \mathcal{A}_1 + 2 c^A c^B \mathcal{A}_2 + (c^{A^2} + c^{B^2}) \mathcal{A}_3 + 2 c^A c^B \mathcal{A}_4 + 2 c^{A^2} \mathcal{A}_5 + 2 c^{B^2} \mathcal{A}_6 + 4 c^A c^B \mathcal{A}_7 \right]. \tag{156}$$

We can use the same procedure to evaluate dot products for all the remaining pairs of trials. This gives:

$$\langle \Delta k_i^{(\hat{Q}+1)1} \Delta k_i^{(\hat{Q}+1)2} \rangle_i = \frac{1}{4N} \left[ (c^{A2} + c^{B2})\mathcal{A}_1 + 2c^A c^B \mathcal{A}_2 + (c^{A2} + c^{B2})\mathcal{A}_3 + 2c^A c^B \mathcal{A}_4 + 2c^{B2}\mathcal{A}_5 + 2c^{A2}\mathcal{A}_6 + 4c^A c^B \mathcal{A}_7 \right]$$

$$\langle \Delta k_i^{11} \Delta k_i^{1(\hat{Q}+1)} \rangle_i = \frac{1}{4N} \left[ 2c^A c^B \mathcal{A}_1 + (c^{A2} + c^{B2})\mathcal{A}_2 + (c^{A2} + c^{B2})\mathcal{A}_3 + 2c^A c^B \mathcal{A}_4 + (c^{A2} + c^{B2})\mathcal{A}_5 + (c^{A2} + c^{B2})\mathcal{A}_6 + 4c^A c^B \mathcal{A}_7 \right]$$  (157)

$$\langle \Delta k_i^{(\hat{Q}+1)1} \Delta k_i^{(\hat{Q}+1)(\hat{Q}+1)} \rangle_i = \frac{1}{4N} \left[ 2c^A c^B \mathcal{A}_1 + (c^{A2} + c^{B2})\mathcal{A}_2 + (c^{A2} + c^{B2})\mathcal{A}_3 + 2c^A c^B \mathcal{A}_4 + (c^{A2} + c^{B2})\mathcal{A}_5 + (c^{A2} + c^{B2})\mathcal{A}_6 + 4c^A c^B \mathcal{A}_7 \right]$$

while

$$\langle \Delta k_i^{11} \Delta k_i^{22} \rangle_i = \frac{1}{4N} \left[ (2c^{A2} + 2c^{B2})\mathcal{A}_1 + 4c^A c^B \mathcal{A}_2 + 2c^{A2}\mathcal{A}_5 + 2c^{B2}\mathcal{A}_6 + 4c^A c^B \mathcal{A}_7 \right]$$  (158a)

$$\langle \Delta k_i^{(\hat{Q}+1)1} \Delta k_i^{(\hat{Q}+2)2} \rangle_i = \frac{1}{4N} \left[ (2c^{A2} + 2c^{B2})\mathcal{A}_1 + 4c^A c^B \mathcal{A}_2 + 2c^{B2}\mathcal{A}_5 + 2c^{A2}\mathcal{A}_6 + 4c^A c^B \mathcal{A}_7 \right]$$  (158b)

$$\langle \Delta k_i^{11} \Delta k_i^{(\hat{Q}+1)(\hat{Q}+1)} \rangle_i = \frac{1}{4N} \left[ 4c^A c^B \mathcal{A}_1 + (2c^{A2} + 2c^{B2})\mathcal{A}_2 + 2c^{B2}\mathcal{A}_5 + 2c^{A2}\mathcal{A}_6 + 4c^A c^B \mathcal{A}_7 \right]$$  (158c)

$$\langle \Delta k_i^{(\hat{Q}+1)1} \Delta k_i^{1(\hat{Q}+1)} \rangle_i = \frac{1}{4N} \left[ 4c^A c^B \mathcal{A}_1 + (2c^{A2} + 2c^{B2})\mathcal{A}_2 + 2c^{A2}\mathcal{A}_5 + 2c^{B2}\mathcal{A}_6 + 4c^A c^B \mathcal{A}_7 \right]$$  (158d)

and

$$\langle \Delta k_i^{11} \Delta k_i^{(\hat{Q}+1)2} \rangle_i = \langle \Delta k_i^{11} \Delta k_i^{2(\hat{Q}+1)} \rangle_i = \langle \Delta k_i^{(\hat{Q}+1)1} \Delta k_i^{12} \rangle_i = \langle (\Delta k_i^{(\hat{Q}+1)1} \Delta k_i^{(\hat{Q}+2)(\hat{Q}+1)}) \rangle_i$$

$$= \frac{1}{4N} \left[ (c^{A2} + c^{B2} + 2c^A c^B)\mathcal{A}_1 + (c^{A2} + c^{B2} + 2c^A c^B)\mathcal{A}_2 + (c^{A2} + c^{B2})\mathcal{A}_5 + (c^{A2} + c^{B2})\mathcal{A}_6 + 4c^A c^B \mathcal{A}_7 \right].$$  (159)

All the $\mathcal{A}$ coefficients can easily be evaluated analytically. However, we have shown in the previous paragraph that the only coefficients that do not cancel in *Equation 141* are $\mathcal{A}_1$ and $\mathcal{A}_2$; these two are evaluated analytically in *Equation 145* and *Equation 146*.

## Extracting intuition

Can we derive a more intuitive picture of why and how context selectivity increases over learning? We have seen in the previous paragraphs that context selectivity increases because the difference within the first set of curly parenthesis of *Equation 141* is positive (while the difference within the second set of curly parenthesis vanishes). To simplify the math, we assume that $c^A = -c^B$; this condition thus reads:

$$\langle \Delta k_i^{11} \Delta k_i^{22} \rangle_i - \langle \Delta k_i^{11} \Delta k_i^{(\hat{Q}+1)(\hat{Q}+1)} \rangle_i > 0.$$  (160)

(With respect to *Equation 142*, we could get rid of pairs of trials with $s = ((\hat{Q}+1)1)$ because, when $c^A = -c^B$, they give identical results to $s = (11)$.)

*Equation 160* indicates that, over learning, activity from trial $s = (11)$ becomes closer (i.e., more correlated) to activity from trials with the same category and context, such as $s' = (22)$, than trials with the same category but different context, such as $s' = ((\hat{Q}+1)(\hat{Q}+1))$. On the contrary, activity from trial $s = (11)$ becomes equally close to activity from trials with different category and same context, such as $s' = ((\hat{Q}+1)2)$, and trials with different category and different context, such as $s' = (2(\hat{Q}+1))$. This can be seen from *Equation 159*, from which

$$\langle \Delta k_i^{11} \Delta k_i^{(\hat{Q}+1)2} \rangle_i - \langle \Delta k_i^{11} \Delta k_i^{2(\hat{Q}+1)} \rangle_i = 0$$  (161)

follows.

The geometrical relationships implied by both *Equation 160* and *Equation 161* can be easily verified in *Figure 6—figure supplement 1C*, which shows the synaptic drive from simulated circuits; the middle panel shows a circuit for which we have exactly $c^A = -c^B$. Taken together, *Equation 160* and *Equation 161* indicate that the increase in context selectivity comes from activity clustering by context over learning; such clustering is, however, category dependent. This leads to the emergence of four statistically distinguishable clouds, one for each combination of category and context. This is visible in simulated activity from *Figure 6—figure supplement 1C*, and is illustrated in *Figure 7A–C*.

## Detailed analysis of category selectivity

We now provide extra detail on the behaviour of category selectivity. We start explaining why, as observed in *Figure 6A* and *Figure 7—figure supplement 1A*, initial selectivity does not vanish, but is weakly negative. This phenomenon is observed both for the synaptic drive $k^s$ and the activity $y^s$; for the sake of simplicity, we focus on the former.

Consider for a moment the case $Q = 2$ (XOR computation). The geometry of the initial synaptic drive is in that case particularly simple, and is illustrated in *Figure 6—figure supplement 2D*. As can be easily verified by using *Equation 5b* and *Equation 110*, each synaptic drive is given by the linear superposition of two vectors: a vector among $\tilde{\mu}^1$ and $\tilde{\mu}^2$, and a vector among $\tilde{\nu}^1$ and $\tilde{\nu}^2$. Vectors $\tilde{\mu}$ and $\tilde{\nu}$ are obtained by applying the initial connectivity $u_0$ to vectors $\mu$ and $\nu$ (*Equation 110*); for example, $\tilde{\mu}^1 = u_0 \cdot \mu^1$. In the plane spanned by vectors $\tilde{\mu}$ and $\tilde{\nu}$, the geometry of synaptic drives is square like (*Figure 6—figure supplement 2D*). To verify that, observe that the squared distance between consecutive vertices is identical – for example,

$$\langle (k_{0,i}^{11} - k_{0,i}^{12})^2 \rangle_i = 2\langle (k_{0,i}^{11})^2 \rangle_i - 2\langle k_{0,i}^{11} k_{0,i}^{12} \rangle_i = 2 - 1 = 1 \tag{162}$$

where we used *Equation 43* together with *Equation 111*. Opposite vertices have instead double squared distance – for example,

$$\langle (k_{0,i}^{11} - k_{0,i}^{22})^2 \rangle_i = 2\langle (k_{0,i}^{11})^2 \rangle_i - 2\langle k_{0,i}^{11} k_{0,i}^{22} \rangle_i = 2 - 0 = 2 \tag{163}$$

as expected for a square. Importantly, consecutive vertices are associated with different categories, while opposite vertices are associated with the same category; this implies that initial category selectivity is negative. In fact, using *Equation 162* and *Equation 163* into *Equation 124* yields:

$$\tilde{S}^{\text{cat}} = \frac{1-2}{1+2} = -\frac{1}{3} < 0. \tag{164}$$

It is easy to see that initial category selectivity is negative also when $Q > 2$. However, its magnitude converges to zero as the number of stimuli and context cues, $Q$, increases (*Figure 7—figure supplement 1A*). This is due to the fact that, as $Q$ becomes large, both the within-category and the across-category averages in *Equation 124* become dominated by pairs of trials with different stimulus and context cue; activity from those pairs of trials are characterized by identical initial distances (=2, as in *Equation 163*), and thus the two averages become similar.

We now shed light on a second phenomenon: the fact that category selectivity increases over learning for the activity $y^s$, but remains identical for the synaptic drive $k^s$. This is observed both in simulations (*Figure 6—figure supplement 1B* and *Figure 7—figure supplement 1A*), and in numerical integration of theoretical expressions (*Figure 6—figure supplement 1A* and *Figure 7—figure supplement 1A*). To see why this happens, we assume that the number of stimuli and context cues, $Q$, is fairly large ($1 \ll Q \ll N$). As discussed above, in this limit, initial category selectivity is approximately close to zero. To compute selectivity after learning, we use *Equation 124*, and evaluate the within-category and the across-category averages. We compute averages to the dominant terms in $Q$, which correspond to pairs of trials with different stimulus and context cue. Using the same $s$ and $s'$ trials as in Section Detailed analysis of context selectivity, we obtain

$$\langle \langle \langle (k_i^s - k_i^{s'})^2 \rangle_i \rangle_{s,s' \text{ same cat}} \simeq \langle (k_i^{11} - k_i^{22})^2 \rangle_i + \langle (k_i^{11} - k_i^{(\hat{Q}+1)(\hat{Q}+1)})^2 \rangle_i + \langle (k_i^{(\hat{Q}+1)1} - k_i^{(\hat{Q}+2)2})^2 \rangle_i + \langle (k_i^{(\hat{Q}+1)1} - k_i^{1(\hat{Q}+1)})^2 \rangle_i$$
$$\langle \langle \langle (k_i^s - k_i^{s'})^2 \rangle_i \rangle_{s,s' \text{ diff cat}} \simeq \langle (k_i^{11} - k_i^{(\hat{Q}+1)2})^2 \rangle_i + \langle (k_i^{11} - k_i^{2(\hat{Q}+1)})^2 \rangle_i + \langle (k_i^{(\hat{Q}+1)1} - k_i^{12})^2 \rangle_i + \langle (k_i^{(\hat{Q}+1)1} - k_i^{(\hat{Q}+2)(\hat{Q}+1)})^2 \rangle_i. \tag{165}$$

To show that category selectivity does not change over learning, we need to show that the two lines above are identical. Using *Equation 139*, this condition can be written as:

$$\langle \Delta k_i^{11} \Delta k_i^{22} \rangle_i + \langle \Delta k_i^{11} \Delta k_i^{(\hat{Q}+1)(\hat{Q}+1)} \rangle_i + \langle \Delta k_i^{(\hat{Q}+1)1} \Delta k_i^{(\hat{Q}+2)2} \rangle_i + \langle \Delta k_i^{(\hat{Q}+1)1} \Delta k_i^{1(\hat{Q}+1)} \rangle_i =$$
$$\langle \Delta k_i^{11} \Delta k_i^{(\hat{Q}+1)2} \rangle_i + \langle \Delta k_i^{11} \Delta k_i^{2(\hat{Q}+1)} \rangle_i + \langle \Delta k_i^{(\hat{Q}+1)1} \Delta k_i^{12} \rangle_i + \langle \Delta k_i^{(\hat{Q}+1)1} \Delta k_i^{(\hat{Q}+2)(\hat{Q}+1)} \rangle_i, \tag{166}$$

which can now be easily verified by using *Equation 158* and *Equation 159*.

*Equation 166* indicates that, on average across contexts, synaptic drives from trials with the same category are as close as trials with different category. This geometrical relationship can be easily verified in *Figure 6—figure supplement 1C*, which shows the synaptic drive from simulated circuits.

We focus on the middle panel, where we have $c^A = -c^B$. The four activity clouds corresponding to different combinations of category and context values are approximately arranged on the vertices of a square; consecutive vertices are associated with different categories, while opposite vertices are associated with the same category. To see why *Equation 166* holds, note that squared distances among synaptic drives associated with different category are approximately identical, while squared distances among synaptic drives associated within the same category are either 0 (approximately, half of the times), or twice the across-category distance (the other half). It is interesting to observe that this square-like configuration, which emerges over learning from an almost unstructured one (*Figure 6—figure supplement 1C*, left), strongly resembles the initial configuration of the XOR task (*Figure 6—figure supplement 2D*).

A fundamental feature of this configuration is that synaptic drives are not linearly separable by category. The activity vectors $y^s$, on the other hand, are linearly separable. Before learning, linear separability is guaranteed by the nonlinearity $\Psi$, which makes activity vectors linearly separable along random directions (*Barak et al., 2013*). After learning, activity vectors become linearly separable also along task-relevant directions. In the simplified scenario where $\eta_w \ll \eta_u$, the activity vectors become linearly separable along $w_0$; in the general case, they become linearly separable along a direction that is correlated with $w_0$. This is shown in *Figure 6—figure supplement 1D* the configuration of activity is very similar to synaptic drives, but activity vectors associated with different categories clusters, and thus become linearly separable, along an emerging, orthogonal direction. This drives the increase in category selectivity that was observed both in equations and simulations (*Figure 6—figure supplement 1A, B* and *Figure 7—figure supplement 1A*). A further insight on the relationship between selectivity and activity geometry is given in the next section.

We conclude with a remark. Although for activity variables category selectivity robustly increases, the fact that selectivity is weakly negative before learning implies that asymptotic values can be small, or even negative. This is compatible with findings in *Bernardi et al., 2020*, where very small values of category clustering (*Equation 124*) were observed. This observation stresses the importance of measuring, in experimental preparations, neural activity across multiple stages of learning.

## Analysis of patterns of context and category selectivity

In this section, we investigate how changes in context and category selectivity are distributed across neurons.

In the simple task, we found that the magnitude of selectivity changes for a given neuron, $i$, was correlated with the magnitude of the $i$th entry of the initial readout vector $w_0$ (*Equation 65*, *Figure 5B, C*). This vector defines the direction along which clustering by category takes place. In fact, if one draws the vector joining the centers of the activity clouds associated with different categories, $y^A$ and $y^B$ (*Equation 78*), the resulting direction is correlated with $w_0$ (*Equation 96*). This direction is indicated with $d$ in the main text; cloud centers $y^A$ and $y^B$ are plotted, in *Figure 3B, C* and *Figure 2—figure supplement 1B*, as magenta triangles.

In analogy with the simple task, we now hypothesize that the magnitude of changes in context and category selectivity for a given neuron, $i$, is related to the magnitude of the $i$th entry of the context and category directions, $d^{ctx}$ and $d^{cat}$. Those coincide with the directions along which clustering to context and category emerges (*Figure 7B, C*), and are given by the vectors joining the centers of the activity clouds associated with different contexts (*Equation 130*) and categories (*Equation 78*). The cloud centers for category and context are plotted, in *Figure 7B, C* and *Figure 6—figure supplement 1C, D*, as magenta and pink triangles. This assumption is verified in *Figure 6—figure supplement 3A, B*, which shows that selectivity changes and context and category directions are highly correlated. Our reasoning implies that, in order to understand how selectivity changes are distributed across neurons, we need to evaluate the entries of the context and category directions; this is done, analytically, in the rest of this section.

As we are interested in selectivity changes, we focus on activity changes, and approximate

$$d^{ctx} \simeq \Delta y^1 - \Delta y^2 \tag{167a}$$

$$d^{cat} \simeq \Delta y^A - \Delta y^B \tag{167b}$$

where, similar to *Equation 78* and *Equation 130*, we have taken

$$\Delta \boldsymbol{y}^1 = \langle \Delta \boldsymbol{y}^{s_1} \rangle_{s_1} \tag{168a}$$

$$\Delta \boldsymbol{y}^2 = \langle \Delta \boldsymbol{y}^{s_2} \rangle_{s_2} \tag{168b}$$

and

$$\Delta \boldsymbol{y}^A = \langle \Delta \boldsymbol{y}^{s_A} \rangle_{s_A} \tag{169a}$$

$$\Delta \boldsymbol{y}^B = \langle \Delta \boldsymbol{y}^{s_B} \rangle_{s_B}. \tag{169b}$$

We start with context. We have seen in Section Detailed analysis of context selectivity that context selectivity can also be studied at the level of the synaptic drive $\boldsymbol{k}^s$, which greatly simplifies the analysis. Starting from *Equation 120*, we thus compute

$$
\begin{aligned}
\Delta k_i^1 &= \frac{w_{0,i}}{2} \left[ \sum_S \langle c^{SC_{s_1}} \Psi_i'^{SC_{s_1}} \rangle_{s_1} + \sum_C \langle c^{S_{s_1}C} \Psi_i'^{S_{s_1}C} \rangle_{s_1} \right] \\
&= \frac{w_{0,i}}{2} \left[ \sum_S \langle c^{SC} \Psi_i'^{SC_+} \rangle_{C_+} + \sum_C \langle c^{SC} \Psi_i'^{SC} \rangle_S \right] \\
&= \frac{w_{0,i}Q}{4} \left[ c^A \langle \Psi_i'^{S_+C_+} \rangle_{S_+,C_+} + c^B \langle \Psi_i'^{S_-C_-} \rangle_{S_-,C_+} + 2 \langle c^{SC} \Psi_i'^{SC} \rangle_{S,C} \right].
\end{aligned} \tag{170}
$$

As in Section Detailed analysis of context selectivity, indices $S_+$ and $S_-$ (and, similarly, $C_+$ and $C_-$) run, respectively, from 1 to $Q/2$ and from $Q/2 + 1$ to $Q$. Similarly,

$$\Delta k_i^2 = \frac{w_{0,i}Q}{4} \left[ c^B \langle \Psi_i'^{S_+C_-} \rangle_{S_+,C_-} + c^A \langle \Psi_i'^{S_-C_-} \rangle_{S_-,C_-} + 2 \langle c^{SC} \Psi_i'^{SC} \rangle_{S,C} \right]. \tag{171}$$

Note that, because of the first two terms in the right-hand sides, *Equation 170* and *Equation 171* are not identical.

To further simplify the analysis, we assume that $c^B \simeq -c^A$. As discussed in Section Context-dependent task: computing activity, in the current task, this represents a good approximation for a large space of parameters; we verified with simulations that our main results also hold, qualitatively, in circuits where this approximation fails (notably, in the circuit illustrated in the third column of *Figure 6*, see *Figure 6—figure supplement 3C, D*). Combining *Equation 167a* with *Equation 170* and *Equation 171*, we then obtain

$$
\begin{aligned}
d_i^{\text{ctx}} = \Delta k_i^1 - \Delta k_i^2 &= \frac{w_{0,i}c^A Q}{4} \langle \langle \Psi_i'^{S_+C} \rangle_{S_+} - \langle \Psi_i'^{S_-C} \rangle_{S_-} \rangle_C \\
&\equiv \frac{w_{0,i}c^A Q}{4} \langle D_i^C \rangle_C
\end{aligned} \tag{172}
$$

where we have defined

$$D_i^C \equiv \langle \Psi_i'^{S_+C} \rangle_{S_+} - \langle \Psi_i'^{S_-C} \rangle_{S_-}. \tag{173}$$

*Equation 172* indicates that neurons exhibiting a strong increase in context selectivity are characterized by: (1) strong readout connectivity, before learning, as quantified by $w_{0,i}$, and (2) a large value of $D_i^C$, averaged over context cues. $D_i^C$ is a function of the response gain function, $\Psi'$, evaluated before learning; specifically, $D_i^C$ measures the difference in the initial gain in response to the two classes of stimuli (the first half, $S_+ = 1, \ldots, Q/2$, and the second half, $S_- = Q/2, \ldots, Q$). These predictions, which were derived for the synaptic drive $\boldsymbol{k}^s$, also hold, qualitatively, for the activity $\boldsymbol{y}^s$ (*Figure 7*).

We next compute the category direction $\boldsymbol{d}^{cat}$; we focus again on the synaptic drive $\boldsymbol{k}^s$ rather than activity $\boldsymbol{y}^s$. We observe that, before learning, the centers of synaptic drive vectors associated with categories A and B are perfectly identical. In fact,

$$
\begin{aligned}
k_{0,i}^A &= \frac{\langle \tilde{\mu}_i^{S_{s_A}} \rangle_{s_A} + \langle \tilde{\nu}_i^{C_{s_A}} \rangle_{s_A}}{\sqrt{2}} \\
&= \frac{\langle \tilde{\mu}_i^S \rangle_S + \langle \tilde{\nu}_i^C \rangle_C}{\sqrt{2}},
\end{aligned} \tag{174}
$$

and an identical expression is obtained for $k_{0,i}^B$. The fact that the centers are identical is due to the fact that sensory inputs for the two categories are collinear, and perfectly intermingled (*Figure 6—figure*

*supplement 2D*). We now consider the synaptic drive changes over learning. Starting from *Equation 120*, we have

$$
\begin{aligned}
\Delta k_i^{\text{A}} &= \frac{w_{0,i}}{2}\left[\sum_S \langle c^{SC_{s_{\text{A}}}}\Psi_i'^{SC_{s_{\text{A}}}}\rangle_{s_{\text{A}}} + \sum_C \langle c^{S_{s_{\text{A}}}C}\Psi_i'^{S_{s_{\text{A}}}C}\rangle_{s_{\text{A}}}\right]\\
&= \frac{w_{0,i}}{2}\left[\sum_S \langle c^{SC}\Psi_i'^{SC}\rangle_C + \sum_C \langle c^{SC}\Psi_i'^{SC}\rangle_S\right]\\
&= w_{0,i}Q\langle c^{SC}\Psi_i'^{SC}\rangle_{S,C}.
\end{aligned}
\tag{175}
$$

It is easy to show that $\Delta k_i^{\text{B}}$ yields the same result, implying that the centers for synaptic drive vectors associated with categories A and B remain identical over learning (*Figure 6—figure supplement 1C*, magenta triangles). This happens because the synaptic drive vectors associated with categories A and B remain intermingled, and nonlinearly separable, over learning. We conclude that the category axis $\boldsymbol{d}^{\text{cat}}$ (*Equation 167b*) vanishes, which is in agreement with the observation that category selectivity does not change for synaptic drives (Section Detailed analysis of category selectivity).

To compute $\boldsymbol{d}^{\text{cat}}$, we thus turn to activity $\boldsymbol{y}$. We start from *Equation 118*, and write

$$
\Delta y_i^{\text{A}} = \frac{w_{0,i}}{2}\left[\sum_S \langle c^{SC_{s_{\text{A}}}}\Psi_i'^{S_{s_{\text{A}}}C_{s_{\text{A}}}}\Psi_i'^{SC_{s_{\text{A}}}}\rangle_{s_{\text{A}}} + \sum_C \langle c^{S_{s_{\text{A}}}C}\Psi_i'^{S_{s_{\text{A}}}C_{s_{\text{A}}}}\Psi_i'^{S_{s_{\text{A}}}C}\rangle_{s_{\text{A}}}\right]
\tag{176a}
$$

$$
\Delta y_i^{\text{B}} = \frac{w_{0,i}}{2}\left[\sum_S \langle c^{SC_{s_{\text{B}}}}\Psi_i'^{S_{s_{\text{B}}}C_{s_{\text{B}}}}\Psi_i'^{SC_{s_{\text{B}}}}\rangle_{s_{\text{B}}} + \sum_C \langle c^{S_{s_{\text{B}}}C}\Psi_i'^{S_{s_{\text{B}}}C_{s_{\text{B}}}}\Psi_i'^{S_{s_{\text{B}}}C}\rangle_{s_{\text{B}}}\right].
\tag{176b}
$$

We then expand indices over stimuli and context cues, which yields

$$
\begin{aligned}
\Delta y_i^{\text{A}} = \frac{w_{0,i}Q}{2}\Big[\ &c^{\text{A}}\langle\Psi_i'^{S_+C_+}\Psi_i'^{\bar{S}_+C_+}\rangle + c^{\text{B}}\langle\Psi_i'^{S_-C_-}\Psi_i'^{\bar{S}_+C_-}\rangle + c^{\text{B}}\langle\Psi_i'^{S_+C_+}\Psi_i'^{\bar{S}_-C_+}\rangle + c^{\text{A}}\langle\Psi_i'^{S_-C_-}\Psi_i'^{\bar{S}_-C_-}\rangle +\\
&c^{\text{A}}\langle\Psi_i'^{S_+C_+}\Psi_i'^{S_+\bar{C}_+}\rangle + c^{\text{B}}\langle\Psi_i'^{S_-C_-}\Psi_i'^{S_-\bar{C}_+}\rangle + c^{\text{B}}\langle\Psi_i'^{S_+C_+}\Psi_i'^{S_+\bar{C}_-}\rangle + c^{\text{A}}\langle\Psi_i'^{S_-C_-}\Psi_i'^{S_-\bar{C}_-}\rangle\Big]
\end{aligned}
\tag{177}
$$

and

$$
\begin{aligned}
\Delta y_i^{\text{B}} = \frac{w_{0,i}Q}{2}\Big[\ &c^{\text{A}}\langle\Psi_i'^{S_-C_+}\Psi_i'^{\bar{S}_+C_+}\rangle + c^{\text{B}}\langle\Psi_i'^{S_+C_-}\Psi_i'^{\bar{S}_+C_-}\rangle + c^{\text{B}}\langle\Psi_i'^{S_-C_+}\Psi_i'^{\bar{S}_-C_+}\rangle + c^{\text{A}}\langle\Psi_i'^{S_+C_-}\Psi_i'^{\bar{S}_-C_-}\rangle +\\
&c^{\text{A}}\langle\Psi_i'^{S_+C_-}\Psi_i'^{S_+\bar{C}_+}\rangle + c^{\text{B}}\langle\Psi_i'^{S_-C_+}\Psi_i'^{S_-\bar{C}_+}\rangle + c^{\text{B}}\langle\Psi_i'^{S_+C_-}\Psi_i'^{S_+\bar{C}_-}\rangle + c^{\text{A}}\langle\Psi_i'^{S_-C_+}\Psi_i'^{S_-\bar{C}_-}\rangle\Big].
\end{aligned}
\tag{178}
$$

To reduce the clutter, we have removed subscripts after brackets $\langle.\rangle$; those indicate an average taken over all the $S$ and $C$ indices contained within.

As will become clear shortly, the two centers now differ (*Figure 6—figure supplement 1D*, magenta triangles). To simplify those expressions, we again assume that $c^{\text{B}} \simeq -c^{\text{A}}$; this allows us to write

$$
\begin{aligned}
d_i^{\text{cat}} &= \Delta y_i^{\text{A}} - \Delta y_i^{\text{B}}\\
&= \frac{w_{0,i}c^{\text{A}}}{2}\left(\frac{Q}{2}\right)^2 \times\\
&\Bigg[\sum_{C_+}\sum_{S_+}\Psi_i'^{S_+C_+}\left(\sum_{\bar{S}_+}\Psi_i'^{\bar{S}_+C_+} - \sum_{\bar{S}_-}\Psi_i'^{\bar{S}_-C_+}\right) - \sum_{C_+}\sum_{S_-}\Psi_i'^{S_-C_+}\left(\sum_{\bar{S}_+}\Psi_i'^{\bar{S}_+C_+} - \sum_{\bar{S}_-}\Psi_i'^{\bar{S}_-C_+}\right)\\
&- \sum_{C_-}\sum_{S_-}\Psi_i'^{S_-C_-}\left(\sum_{\bar{S}_+}\Psi_i'^{\bar{S}_+C_-} - \sum_{\bar{S}_-}\Psi_i'^{\bar{S}_-C_-}\right) + \sum_{C_-}\sum_{S_+}\Psi_i'^{S_+C_-}\left(\sum_{\bar{S}_+}\Psi_i'^{\bar{S}_+C_-} - \sum_{\bar{S}_-}\Psi_i'^{\bar{S}_-C_-}\right)\\
&+ \sum_{S_+}\sum_{C_+}\Psi_i'^{S_+C_+}\left(\sum_{\bar{C}_+}\Psi_i'^{S_+\bar{C}_+} - \sum_{\bar{C}_-}\Psi_i'^{S_+\bar{C}_-}\right) - \sum_{S_+}\sum_{C_-}\Psi_i'^{S_+C_-}\left(\sum_{\bar{C}_+}\Psi_i'^{S_+\bar{C}_+} - \sum_{\bar{C}_-}\Psi_i'^{S_+\bar{C}_-}\right)\\
&- \sum_{S_-}\sum_{C_-}\Psi_i'^{S_-C_-}\left(\sum_{\bar{C}_+}\Psi_i'^{S_-\bar{C}_+} - \sum_{\bar{C}_-}\Psi_i'^{S_-\bar{C}_-}\right) + \sum_{S_-}\sum_{C_+}\Psi_i'^{S_-C_+}\left(\sum_{\bar{C}_+}\Psi_i'^{S_-\bar{C}_+} - \sum_{\bar{C}_-}\Psi_i'^{S_-\bar{C}_-}\right)\Bigg].
\end{aligned}
\tag{179}
$$

With a little algebra, we can see that

$$
d_i^{\text{cat}} = \frac{w_{0,i}c^{\text{A}}}{2}\left(\frac{Q}{2}\right)^2\left[\sum_{C_+}\left(\sum_{\bar{S}_+}\Psi_i'^{\bar{S}_+C_+} - \sum_{\bar{S}_-}\Psi_i'^{\bar{S}_-C_+}\right)^2 + \sum_{C_-}\left(\sum_{\bar{S}_+}\Psi_i'^{\bar{S}_+C_-} - \sum_{\bar{S}_-}\Psi_i'^{\bar{S}_-C_-}\right)^2\right.
$$
$$
\left. + \sum_{S_+}\left(\sum_{\bar{C}_+}\Psi_i'^{S_+\bar{C}_+} - \sum_{\bar{C}_-}\Psi_i'^{S_+\bar{C}_-}\right)^2 + \sum_{S_-}\left(\sum_{\bar{C}_+}\Psi_i'^{S_-\bar{C}_+} - \sum_{\bar{C}_-}\Psi_i'^{S_-\bar{C}_-}\right)^2\right]
\tag{180}
$$

or, equivalently

$$
\begin{aligned}
d_i^{\text{cat}} &= \frac{w_{0,i} c^A Q}{2} \left[ \left\langle \left( \langle \Psi_i^{S_+ C} \rangle_{S_+} - \langle \Psi_i^{S_- C} \rangle_{S_-} \right)^2 \right\rangle_C + \left\langle \left( \langle \Psi_i^{SC_+} \rangle_{C_+} - \langle \Psi_i^{SC_-} \rangle_{C_-} \right)^2 \right\rangle_S \right] \\
&= \frac{w_{0,i} c^A Q}{2} \left[ \langle (D_i^C)^2 \rangle_C + \langle (D_i^S)^2 \rangle_S \right]
\end{aligned}
\tag{181}
$$

where we have defined

$$
D_i^S \equiv \langle \Psi_i'^{SC_+} \rangle_{C_+} - \langle \Psi_i'^{SC_-} \rangle_{C_-}.
\tag{182}
$$

*Equation 181* indicates that neurons characterized by a strong increase in category selectivity are characterized by: (1) strong readout connectivity, before learning, as quantified by $w_{0,i}$, and (2) large values of $D_i^C$ and/or $D_i^S$, averaged, respectively, over context cues and stimuli.

Note that neurons that are characterized by a strong increase in context selectivity (*Equation 172*), which have large $w_{0,i}$ and $D_i^C$ values, are also characterized by a strong increase in category selectivity (*Equation 181*). On the other hand, neurons with large $w_{0,i}$ and $D_i^S$ values are characterized by a strong increase in category selectivity (*Equation 181*), but not context (*Equation 172*). Overall, strongly selective neurons can thus be classified in two groups: one displaying mixed selectivity to category and context, and one displaying pure selectivity to category. By defining the quantity:

$$
G_i = \langle |D_i^C| \rangle_C - \langle |D_i^S| \rangle_S
\tag{183}
$$

we see that the former group is characterized by larger values of $G_i$ with respect to the latter. This is verified and illustrated in *Figure 8B, C*.

## Software
### Circuit simulations
Simulations were implemented with the Python programming language. Gradient-descent learning was implemented with the PyTorch package. We used the *SGD* optimization function, with loss *MSELoss*. On every learning epoch, the batch included all sensory input vectors. Training stopped when the loss function dropped below $10^{-5}$. Learning rates were taken to be $\eta = 0.1$ for input connectivity $\boldsymbol{u}$, and $\eta \cdot \eta_w/\eta_u$ (with values of $\eta_u$ and $\eta_w$ as indicated in Section Tables of parameters) for readout connectivity $\boldsymbol{w}$.

**Table 1.** Table of parameters for figures in the main text.

| Figure | $N$ | $Q$ | $\eta_w/\eta_u$ | $\Theta_1, \Psi$ | $\Theta_2, \Psi$ | $\Theta_1, \Phi$ | $\Theta_2, \Phi$ |
|---|---|---|---|---|---|---|---|
| *Figures 2, 3 and 5*, first and second columns | 200 | 20 | 0.0 | 1.0 | 2.0 | 1.0 | 0.0 |
| *Figures 2, 3 and 5*, third column | 200 | 20 | 0.0 | 1.0 | 2.0 | 1.0 | 2.0 |
| *Figure 4A* | 200 | 20 | 0.4 | 2.0 | 2.0 | varies | varies |
| *Figure 4B* | 200 | 20 | 0.4 | varies | varies | 1.0 | 2.0 |
| *Figure 4C* | 200 | 20 | varies | 2.0 | 2.0 | 1.0 | varies |
| *Figure 4D* | 200 | varies | 0.4 | 2.0 | 2.0 | 1.0 | varies |
| *Figure 6A–C*, first and second columns | 600 | 8 ($P = 64$) | 0.0 | 1.0 | 0.0 | 1.0 | 0.0 |
| *Figure 6A–C*, third column | 600 | 8 ($P=64$) | 0.0 | 1.0 | 0.0 | 1.0 | 4.0 |
| *Figure 7D* | 600 | 8 ($P = 64$) | 0.2 | 2.5 | 2.0 | varies | varies |
| *Figure 7E* | 600 | 8 ($P = 64$) | varies | 2.5 | 2.0 | 1.0 | varies |
| *Figure 7F* | 600 | varies | 0.2 | 2.5 | 2.0 | 1.0 | varies |

**Table 2.** Table of parameters for figure supplements.

| Figure supplement | $N$ | $Q$ | $\eta_w/\eta_u$ | $\Theta_1, \Psi$ | $\Theta_2, \Psi$ | $\Theta_1, \Phi$ | $\Theta_2, \Phi$ |
|---|---|---|---|---|---|---|---|
| *Figure 2—figure supplement 1A* | 200 | varies | varies | varies | varies | varies | varies |
| *Figure 2—figure supplement 2E* | 200 | 20 | 0.0 | 1.0 | 2.0 | 1.0 | 0.0 |
| *Figure 3—figure supplement 1*, first column | varies | 20 | 0.0 | 1.0 | 0.0 | 1.0 | 0.0 |
| *Figure 3—figure supplement 1*, second column | 200 | 20 | varies | 1.0 | 0.0 | 1.0 | 0.0 |
| *Figure 3—figure supplement 1*, third column | 200 | 20 | 0.0 | 1.0 | varies | 1.0 | 0.0 |
| *Figure 3—figure supplement 2*, first column | varies | 20 | 0.0 | 1.0 | 0.0 | 1.0 | 2.0 |
| *Figure 3—figure supplement 2*, second column | 200 | 20 | varies | 1.0 | 0.0 | 1.0 | 2.0 |
| *Figure 3—figure supplement 2*, third column | 200 | 20 | 0.0 | 1.0 | varies | 1.0 | 2.0 |
| *Figure 2—figure supplement 4A, B* | 200 | 12 | 0.0 | 1.0 | 2.0 | 1.0 | 0.0 |
| *Figure 2—figure supplement 4C* | 200 | 12 | 0.0 | 1.0 | 2.0 | 1.0 | 2.0 |
| *Figure 2—figure supplement 4D, E*, first column | 200 | 12 | 0.1 | 2.0 | varies | 1.0 | varies |
| *Figure 2—figure supplement 4D, E*, second column | 200 | 12 | varies | 2.0 | 2.0 | 1.0 | varies |
| *Figure 2—figure supplement 4D, E*, third column | 200 | varies | 0.1 | 2.0 | 2.0 | 1.0 | varies |
| *Figure 6—figure supplement 1A, B* | 600 | varies | varies | varies | varies | varies | varies |
| *Figure 6—figure supplement 2A* | 600 | 8 ($P = 64$) | 0.0 | 1.0 | 3.0 | 1.0 | 0.0 |
| *Figure 6—figure supplement 2B* | 600 | 8 ($P = 64$) | 0.0 | 1.0 | 3.0 | 1.0 | 4.0 |
| *Figure 6—figure supplement 2C* | 600 | varies | varies | varies | varies | varies | varies |
| *Figure 7—figure supplement 1A–C* | 600 | varies | 0.0 | 1.0 | 0.0 | 1.0 | 0.0 |
| *Figure 7—figure supplement 1D–F* | 600 | varies | 0.0 | 1.0 | 0.0 | 1.0 | 4.0 |

## Tables of parameters

We summarize below the parameters chosen for the simulations reported in figures and figure supplements. For figures not included in the tables below (*Table 1*, *Table 2*) parameters have been detailed in figures captions. We have taken everywhere $z^A = 0.75$, $z^B = 0.25$ (note that activity variables range between 0 and 1).

## Evaluation of averages

Evaluating the approximate theoretical expressions for activity measures given in Sections Simple categorization task and Context-dependent categorization task requires computing a number of Gaussian integrals over nonlinear functions. We compute those averages numerically; details are provided below.

The simplest average, which only involves one nonlinear function, was denoted by $\langle F \rangle$ (*Equation 45*). We rewrite *Equation 45* in an integral form, yielding

$$\langle F \rangle \equiv \int \mathcal{D}a F(a) \tag{184}$$

where we have used the short-hand notation

$$\int \mathcal{D}a \equiv \int_{-\infty}^{\infty} \mathrm{d}a \, \frac{\exp(-a^2/2\pi)}{\sqrt{2\pi}} \, . \tag{185}$$

This integral was computed numerically via Hermite–Gaussian quadrature.

Averages involving two nonlinear functions were denoted by $\langle FF \rangle$ (*Equation 113*). We rewrite *Equation 113* in an integral form, yielding

$$\langle FF \rangle \equiv \int \mathcal{D}a \int \mathcal{D}b_1 \int \mathcal{D}b_2 F\left(\frac{1}{\sqrt{2}}(a+b_1)\right) F\left(\frac{1}{\sqrt{2}}(a+b_2)\right) = \int \mathcal{D}a \left[\int \mathcal{D}b F\left(\frac{1}{\sqrt{2}}(a+b)\right)\right]^2 . \tag{186}$$

This integral was computed again via Hermite–Gaussian quadrature.

Averages involving four nonlinear functions, such as $\langle \Psi'(k_{0,i}^q)\Psi'(k_{0,i}^s)\Psi'(k_{0,i}^{q'})\Psi'(k_{0,i}^{s'})\rangle_i$ from **Equation 133** (Section Context-dependent task: computing normalized dot products) were computed instead via the function *nquad* from the Python *scipy.integrate* package. We start by rewriting the argument of the average as:

$$\Psi'(k_{0,i}^q)\Psi'(k_{0,i}^s)\Psi'(k_{0,i}^{q'})\Psi'(k_{0,i}^{s'}) = \Psi'\left(\frac{1}{\sqrt{2}}(a_{S_q}+b_{C_q})\right)\Psi'\left(\frac{1}{\sqrt{2}}(a_{S_s}+b_{C_s})\right)\Psi'\left(\frac{1}{\sqrt{2}}(a_{S_{q'}}+b_{C_{q'}})\right)\Psi'\left(\frac{1}{\sqrt{2}}(a_{S_{s'}}+b_{C_{s'}})\right). \quad (187)$$

For each value of the stimulus index $S$ and the context cue index $C$, $a_S$ and $b_C$ are two independent, zero-mean and unit-variance Gaussian variables. If the values of $S$ and $C$ are different across the four trials $q$, $s$, $q'$ and $s'$, then all $a$ and $b$ variables involved in **Equation 187** are different, and the average reads

$$\langle \Psi'(k_{0,i}^q)\Psi'(k_{0,i}^s)\Psi'(k_{0,i}^{q'})\Psi'(k_{0,i}^{s'})\rangle_i = \int \mathcal{D}a_{S_q} \int \mathcal{D}a_{S_s} \int \mathcal{D}a_{S_{q'}} \int \mathcal{D}a_{S_{s'}} \int \mathcal{D}b_{C_q} \int \mathcal{D}b_{C_s} \int \mathcal{D}b_{C_{q'}} \int \mathcal{D}b_{C_{s'}} \times$$
$$\Psi'\left(\frac{1}{\sqrt{2}}(a_{S_q}+b_{C_q})\right)\Psi'\left(\frac{1}{\sqrt{2}}(a_{S_s}+b_{C_s})\right)\Psi'\left(\frac{1}{\sqrt{2}}(a_{S_{q'}}+b_{C_{q'}})\right)\Psi'\left(\frac{1}{\sqrt{2}}(a_{S_{s'}}+b_{C_{s'}})\right) \quad (188)$$

which simplifies into

$$\langle \Psi'(k_{0,i}^q)\Psi'(k_{0,i}^s)\Psi'(k_{0,i}^{q'})\Psi'(k_{0,i}^{s'})\rangle_i = \left[\int \mathcal{D}a_{S_q} \int \mathcal{D}b_{C_q} \Psi'\left(\frac{1}{\sqrt{2}}(a_{S_q}+b_{C_q})\right)\right]^4 = \langle \Psi'\rangle^4. \quad (189)$$

If the stimulus $S$ or the context cue $C$ are, instead, identical across two o more trials ($q$, $s$, $q'$, and $s'$),

then some of the $a$ and $b$ variables in **Equation 187** are shared across nonlinear functions. This generates correlations, which determine the final value of the average. For example, assume $S_q = S_s$, while all other $S$ and $C$ values are different among each other. Then the average reads

$$\langle \Psi'(k_{0,i}^q)\Psi'(k_{0,i}^s)\Psi'(k_{0,i}^{q'})\Psi'(k_{0,i}^{s'})\rangle_i = \int \mathcal{D}a_{S_q} \int \mathcal{D}a_{S_{q'}} \int \mathcal{D}a_{S_{s'}} \int \mathcal{D}b_{C_q} \int \mathcal{D}b_{C_s} \int \mathcal{D}b_{C_{q'}} \int \mathcal{D}b_{C_{s'}} \times$$
$$\Psi'\left(\frac{1}{\sqrt{2}}(a_{S_q}+b_{C_q})\right)\Psi'\left(\frac{1}{\sqrt{2}}(a_{S_q}+b_{C_s})\right)\Psi'\left(\frac{1}{\sqrt{2}}(a_{S_{q'}}+b_{C_{q'}})\right)\Psi'\left(\frac{1}{\sqrt{2}}(a_{S_{s'}}+b_{C_{s'}})\right) \quad (190)$$

which simplifies into

$$\langle \Psi'(k_{0,i}^q)\Psi'(k_{0,i}^s)\Psi'(k_{0,i}^{q'})\Psi'(k_{0,i}^{s'})\rangle_i = \langle \Psi'\Psi'\rangle \langle \Psi'\rangle^2. \quad (191)$$

We considered all the possible configurations of $S$ and $C$ indices that can occur in the context-dependent task, and all the resulting correlation patterns. Then, we used analytics to simplify integrals when possible (as in the cases described above). We finally used numerics to evaluate the remaining integral expressions.

## Acknowledgements

FM would like to thank Friedrich Schuessler for useful discussions.

## Additional information

### Competing interests
Peter Latham: Reviewing editor, *eLife*. The other authors declare that no competing interests exist.

## Funding

| Funder | Grant reference number | Author |
|---|---|---|
| Gatsby Charitable Foundation | | Francesca Mastrogiuseppe<br>Naoki Hiratani<br>Peter Latham |
| Wellcome Trust | 110114/Z/15/Z | Francesca Mastrogiuseppe<br>Peter Latham |

The funders had no role in study design, data collection, and interpretation, or the decision to submit the work for publication. For the purpose of Open Access, the authors have applied a CC BY public copyright license to any Author Accepted Manuscript version arising from this submission.

## Author contributions

Francesca Mastrogiuseppe, Conceptualization, Software, Formal analysis, Investigation, Methodology, Writing – original draft, Writing – review and editing; Naoki Hiratani, Conceptualization, Investigation, Writing – original draft, Writing – review and editing; Peter Latham, Conceptualization, Formal analysis, Supervision, Investigation, Writing – original draft, Writing – review and editing

## Author ORCIDs

Francesca Mastrogiuseppe ⬤ http://orcid.org/0000-0002-7682-5178
Naoki Hiratani ⬤ http://orcid.org/0000-0002-8568-2033
Peter Latham ⬤ http://orcid.org/0000-0001-8713-9328

## Decision letter and Author response

Decision letter https://doi.org/10.7554/eLife.79908.sa1
Author response https://doi.org/10.7554/eLife.79908.sa2

# Additional files

## Supplementary files

- MDAR checklist

## Data availability

The current manuscript is a computational study, so no data have been generated for this manuscript. Code is available online at https://github.com/fmastrogiuseppe/EvolutionActivity (copy archived at swh:1:rev:a6b7e083ac6d306599b5c29005dc6aa499e2209a, *Mastrogiuseppe, 2022*).

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
