## [Editor Report]

The findings of the paper are very valuable for neuroscientists studying the learning of abstract representations. It provides compelling evidence that neural networks trained on two-way classification tasks will develop responses whose category and context selectivity profiles depend on key network details, such as neural activation functions and initial connectivity. These results can explain apparently contradictory results in the experimental literature, and make new experimental predictions for testing in the future.

---

## [Decision Letter]

**Decision letter after peer review:**

Thank you for submitting your article "Evolution of neural activity in circuits bridging sensory and abstract knowledge" for consideration by *eLife*. Your article has been reviewed by 2 peer reviewers, and the evaluation has been overseen by a Reviewing Editor and Joshua Gold as the Senior Editor. The following individual involved in review of your submission has agreed to reveal their identity: Guillaume Hennequin (Reviewer #1).

Essential revisions:

1) Justification of some of the model choices that the reviewers asked about and discussion of how much the results depend on them (especially the use of gradient descent).

2) More intuitive explanations for some of the mathematical statements to help make the paper more accessible for the majority of *eLife* readers.

3) A longer discussion about how this paper fits into the current literature, and how it can be generalised.

*Reviewer #1 (Recommendations for the authors):*

We do have a couple of concerns which we would recommend adding some discussion about:

1. We found it strange that authors operationalised what are essentially (binary) classification tasks as regression problems; although the network output is pushed through a sigmoidal nonlinearity with a [0--1] range, here this output is never interpreted as a class probability; indeed, instead of maximizing the cross-entropy loss (i.e. maximising the data likelihood) as is normally done in such a context, the authors chose to minimise a squared error loss; this forced them to introduce somewhat arbitrary "target" outputs (0.25 and 0.75) for the two categories. Several questions arise from this odd choice:

a) why did the authors do that? (e.g. is this standard in the NTK literature?)

b) do the results depend on this choice? robustness to this choice is not a priori obvious to us, because the network is not regularised here (see point 2 below), and learning operates in the vastly overparameterised regime where the cross-entropy loss may (does?) not even have a minimum, so weights could diverge. We suspect this may be the answer to (a), but it would be great to discuss this briefly.

2. The paper does not consider generalisation at all, and indeed the tasks studied here are set up in a way that does not even allow thinking about generalisation; the inputs are chosen completely randomly, as opposed to randomly on some category-specific manifold that could be re-sampled, so it is not even clear how a test set would be constructed. Is this an issue? Does that limit the relevance of this study to only a subset of previously studied relevant neuroscience tasks? Again it would be great to discuss.

3. It was not clear to us how much of the results presented here depend on doing gradient descent, as opposed to more generally minimising the loss? This question was apparently already asked as "Comment 2" in the first round of reviews but we found that the rebuttal did not really answer it, beyond saying that solutions are degenerate.

4. After l311, would it be possible to give the intuition for the emergence of context selectivity, in a couple of sentences? The text currently says that it is a "signature of gradient descent" and that the "mechanism is described in detail in Methods 5.6", but we wonder if the authors could be a bit more specific without getting too technical. It is important because the emergence of context selectivity is the most unexpected of the results presented here.

*Reviewer #2 (Recommendations for the authors):*

– The methods section seems very long to me, maybe too long for a general audience. I would strongly suggest focusing on the details and derivations that are necessary to reproduce the results and to understand the main findings. The rest (as beautiful as it is), I would put in the supplementary material (if *eLife* allows for it).

– Do you need both u and w to be plastic? Have you tried to keep one set fixed? Is the system underdetermined? You show how the cat or ctx selectivity depends on the ratio of learning rates but I am not completely sure which set of weights is the crucial one or let's say the dominant one that is strictly necessary to get the results.

– I am a bit confused how you implemented the context. The stimulus and the context are linearly combined to yield the total sensory input x. But from the perspective of the network that is not different from the simple task. The network sees an input vector x. Whether x is a vector representing the stimulus alone or the stimulus and a context is indistinguishable to the network. Is the difference here that the stimuli were random and orthogonal and if they are linearly combined with another set that is random and orthogonal, the resulting vectors do not necessarily share the same properties? Maybe I misunderstand something here. Would be great if you could clarify that and maybe highlight the differences in the text.

– Maybe worth discussing in the Discussion (these are merely suggestions, I don't expect the authors to run simulations on that):

– comment on limitations and – if possible – how you envision your results to change (or hold) in E/I networks (that abide by Dale's principle) and/or recurrent connections are included.

– How strongly depend the results (or which results) on the gradient descent you used?

– How would the results generalise to more than 2 categories? And what is with the option „unclassified" (no mapping possible).

– I assume each neuron in x is projecting to all neurons in y. Is this clearly stated somewhere? Maybe I missed it – sorry it is hot today!

– After learning, cat selectivity, and later ctx selectivity, cover a broad range with many neurons being close to zero and some showing positive selectivity. However, most of them are still unselective it seems. Is this range in line with experimental results? Could you comment on this heterogeneity? In line 95/96 you say that the cat selectivity increases for EACH neuron but to me it seems some neurons remain at zero selectivity, do I read the figure wrong?

– The distribution of the cat selectivity between the two tasks is different. The cat selectivity for the context-dependent task has many cells with a selectivity close to zero or even negative. Could you comment on that. I apologise if I overlooked it.

---

## [Author Response]

Essential revisions:Reviewer #1 (Recommendations for the authors):We do have a couple of concerns which we would recommend adding some discussion about:1. We found it strange that authors operationalised what are essentially (binary) classification tasks as regression problems; although the network output is pushed through a sigmoidal nonlinearity with a [0--1] range, here this output is never interpreted as a class probability; indeed, instead of maximizing the cross-entropy loss (i.e. maximising the data likelihood) as is normally done in such a context, the authors chose to minimise a squared error loss; this forced them to introduce somewhat arbitrary "target" outputs (0.25 and 0.75) for the two categories. Several questions arise from this odd choice:a) why did the authors do that? (e.g. is this standard in the NTK literature?)b) do the results depend on this choice? robustness to this choice is not a priori obvious to us, because the network is not regularised here (see point 2 below), and learning operates in the vastly overparameterised regime where the cross-entropy loss may (does?) not even have a minimum, so weights could diverge. We suspect this may be the answer to (a), but it would be great to discuss this briefly.

The reviewer is correct. We focused on the mean squared error loss because, in our setup, it simplifies the analysis. With such a loss, we can assume that the learning process converges to a zero-loss minimum; this allows us to derive closed-form expressions for activity at convergence. Using the cross-entropy loss would result instead in diverging weights. Classical NTK work (Lee et al., 2019) has showed that, during such divergence process, the accuracy of the NTK approximation gradually deteriorates (which is expected, because weights changes are no longer small). We now write in the Discussion:

(line 647) “To model categorization, we assumed a quadratic function for the error E (Methods Circuit) – an assumption that effectively casts our categorization tasks into a regression problem. This made9 the model amenable to mathematical analysis, and allowed us to derive transparent equations to characterize activity evolution. Recent machine learning work has showed that, at least in some categorization setups (Hui and Belkin, 2021), a cross-entropy function might result in better learning performance. The mathematical framework used here is, however, not well suited to studying networks with such an error function (Lee et al., 2019). Investigating whether and how our findings extend to networks trained with a cross-entropy error function represents an interesting direction for future work.”

2. The paper does not consider generalisation at all, and indeed the tasks studied here are set up in a way that does not even allow thinking about generalisation; the inputs are chosen completely randomly, as opposed to randomly on some category-specific manifold that could be re-sampled, so it is not even clear how a test set would be constructed. Is this an issue? Does that limit the relevance of this study to only a subset of previously studied relevant neuroscience tasks? Again it would be great to discuss.

As the reviewer says, the goal of this work was to model neuroscience tasks that do not focus on generalization. Nevertheless, there is a large body of machine learning work (e.g. Canatar et al., 2021) addressing generalization in the learning regime we investigate here (NTK). We thus expect to be able to extend our work to include generalization. We now write in the Discussion:

(line 632) “Throughout this work, we focused on two simplified categorization tasks, aimed at capturing the fundamental features of the categorization tasks commonly used in systems neuroscience (Freedman and Assad, 2006; Fitzgerald et al., 2011; Wallis et al., 2001). The mathematical framework we developed to analyze those tasks could, however, easily be extended in several directions, including tasks with more than two categories (Fitzgerald et al., 2011; Reinert et al., 2021; Mante et al., 2013) and tasks involving generalization to unseen stimuli (Barak et al., 2013; Canatar et al., 2021).”

3. It was not clear to us how much of the results presented here depend on doing gradient descent, as opposed to more generally minimising the loss? This question was apparently already asked as "Comment 2" in the first round of reviews but we found that the rebuttal did not really answer it, beyond saying that solutions are degenerate.

The requirement of bringing the loss to zero does not imply any of the results discussed in this paper. This is clearly exemplified by models with fixed intermediate connectivity *u* (see, e.g. Barak et al. 2013, Babadi and Sompolinsky, 2014), which can bring the loss to zero in both our tasks without displaying any change in activity at all. It is also easy to imagine other scenarios where the loss is minimized, but our results do not apply (for example: a circuit model implementing the simple task by aligning activity with changes in the readout vector ∆*w*, rather than its initial value *w*_0_; this would invalidate subsequent findings, Figures 4 and 5).

As neural representations are not heavily constrained by loss minimization, they provide a tool to investigate the underlying learning algorithm. We believe it is specifically important to study representations induced by gradient descent because there is now large evidence that this algorithm performs well, within multi-layer brain-like architectures, on both simple and complex tasks. The experimental predictions we obtained here for gradient descent should be contrasted with predictions from other algorithms. In the Discussion, we highlight differences with models with fixed intermediate connectivity (Barak et al. 2013, Babadi and Sompolinsky, 2014), and reward-modulated Hebbian plasticity (Engel et al., 2015). It will be also important to consider, in future research, other forms of gradient-descent, and learning algorithms that provide biologically-plausible approximations to it. A first step in this direction was taken very recently by Bordelan and Pehlevan, 2022.

We have added more comments on this theme to the Discussion. We now write:

(line 588) “A number of recent theoretical studies have proposed biologically-plausible architectures and plasticity rules that can approximate back-propagation on simple and complex tasks (Lillicrap et al., 2016; Sacramento et al., 2018; Akrout et al., 2019; Whittington and Bogacz, 2017; Payeur et al., 2021; Pogodin and Latham, 2020; Boopathy and Fiete, 2022). Understanding whether these different implementations lead to differences in activity represents a very important direction for future research. Interestingly, recent work has showed that it is possible to design circuit models where the learning dynamics is identical to the one studied in this work, but the architecture is biologically plausible (Boopathy and Fiete, 2022). We expect our results to directly translate to those models. Other biologically-plausible setups might be characterized, instead, by different activity evolution. Recent work (Song et al., 2021; Bordelon and Pehlevan, 2022) made use of a formalism similar to ours to describe learning dynamics induced by a number of different biologically-plausible algorithms and uncovered non-trivial, qualitatively different dynamics. Whether any of these dynamics leads to different neural representations in neuroscience inspired categorization tasks like the ones we studied here is an open, and compelling, question.”

4. After l311, would it be possible to give the intuition for the emergence of context selectivity, in a couple of sentences? The text currently says that it is a "signature of gradient descent" and that the "mechanism is described in detail in Methods 5.6", but we wonder if the authors could be a bit more specific without getting too technical. It is important because the emergence of context selectivity is the most unexpected of the results presented here.

A technically-sound explaination would require, at least, knowledge of Equation 120 from Methods, which we believe is too involved for the main text. We have added a quite simplified version of that explaination.

We now write:

(line 437) “Such novel structure is a signature of the gradient-descent learning rule used by the circuit (Canatar et al., 2021). The mechanism through which context clustering emerges is described in detail in Methods Detailed analysis of context selectivity. But, roughly speaking, context clustering emerges because, for a pair of sensory inputs, how similarly their intermediate-layer representations evolve during learning is determined both by their target category and their correlations (Equation 27, Methods Evolution of connectivity and activity in large circuits). In the simple task, initial correlations were virtually nonexistent (Figure 2C), and thus activity changes were specified only by category; in the context-depend task, initial correlations have structure (Figure 6C), and that structure critically affects neural representations. In particular, inputs with the same context tend to be relatively correlated, and those are also likely to be associated with the same category; their representations are thus clustered by the learning algorithm, resulting in context clustering.”

Reviewer #2 (Recommendations for the authors):– The methods section seems very long to me, maybe too long for a general audience. I would strongly suggest focusing on the details and derivations that are necessary to reproduce the results and to understand the main findings. The rest (as beautiful as it is), I would put in the supplementary material (if eLife allows for it).

Unfortunately, *eLife* does not allow for supplementary material. We will ask the Editors to maintain the table of contents in place. Hopefully, that will help – so readers can pick and choose what they want to look at. We agree that the current Methods section is very long (and we really wish we could have made it shorter!). But it’s not just the length that makes it unsuitable for a general audience, it’s also the math. And we believe that shortening the latter would make the paper even less readable.

– Do you need both u and w to be plastic? Have you tried to keep one set fixed? Is the system underdetermined? You show how the cat or ctx selectivity depends on the ratio of learning rates but I am not completely sure which set of weights is the crucial one or let's say the dominant one that is strictly necessary to get the results.

When either *u* or *w* are fixed, the network can learn our tasks. When *u* is fixed (*η_u_* = 0), the intermediate layer representations do not change. This model has been studied before (e.g. Barak et al. 2013 or Babadi and Sompolinsky 2014, and random features models in machine learning). Its main limitations are that (i) it cannot be used to model learning-induced activity changes in the brain; (ii) it performs poorly on complex machine-learning tasks. In this work, we consider circuit models where *u* does change (all values of *η_w_* and *η_u_* with *η_u_* 6 = 0); in all those cases, intermediate layer representations do change. Our results indicate that category and context selectivity emerge for all values of *η_w_* and *η_u_*, including *η_w_* = 0 (which corresponds to the case when *w* is fixed).

– I am a bit confused how you implemented the context. The stimulus and the context are linearly combined to yield the total sensory input x. But from the perspective of the network that is not different from the simple task. The network sees an input vector x. Whether x is a vector representing the stimulus alone or the stimulus and a context is indistinguishable to the network. Is the difference here that the stimuli were random and orthogonal and if they are linearly combined with another set that is random and orthogonal, the resulting vectors do not necessarily share the same properties? Maybe I misunderstand something here. Would be great if you could clarify that and maybe highlight the differences in the text.

The reviewer is correct. The crucial difference between the two tasks is that now input vectors across different trials are no longer random and orthogonal (and thus uncorrelated). Consider, for example, two trials in which the stimulus part of the input is the same but the context cues are different. The network has to learn that, even though the stimulus vectors are identical (and so the input vectors look very similar overall!), the output can be different (different output category for the same stimulus in two different contexts). This computation (which reduces to the classic XOR task in the case of two stimuli and context cues) is notoriously hard for neural networks. To implement it, a hidden layer with non-linear activation functions is strictly required (in contrast to the simple task, which could be implemented even without hidden layer).

We have expanded the description of the context-dependent task in the main text by adding:

(line 348) “This task is computationally much more involved than the previous one, primarily because context induces nontrivial correlational structure: in the simple task, all sensory input vectors were uncorrelated; in the context-dependent task, that is no longer true. For instance, two sensory inputs with the same stimulus and different context cues are highly correlated. In spite of this high correlation, though, they can belong to different categories (when context cues are associated with different contexts). In contrast, two sensory inputs with different stimuli and different context cues are uncorrelated, but they can belong to the same category. From a mathematical point of view, this correlational structure makes sensory input vectors non linearly-separable. This is in stark contrast to the simple task, for which sensory input vectors were linearly-separable (Barak et al., 2013).”

– Maybe worth discussing in the Discussion (these are merely suggestions, I don't expect the authors to run simulations on that):– comment on limitations and – if possible – how you envision your results to change (or hold) in E/I networks (that abide by Dale's principle) and/or recurrent connections are included.– How strongly depend the results (or which results) on the gradient descent you used?– How would the results generalise to more than 2 categories? And what is with the option „unclassified" (no mapping possible).

We have expanded the Discussion to comment on those aspects. We now write:

(line 632) “Throughout this work, we focused on two simplified categorization tasks, aimed at capturing the fundamental features of the categorization tasks commonly used in systems neuroscience (Freedman and Assad, 2006; Fitzgerald et al., 2011; Wallis et al., 2001). The mathematical framework we developed to analyze those tasks could, however, easily be extended in several directions, including tasks with more than two categories (Fitzgerald et al., 2011; Reinert et al., 2021; Mante et al., 2013) and tasks involving generalization to unseen stimuli (Barak et al., 2013; Canatar et al., 2021). An important feature missing in our tasks, though, is memory: neuroscience tasks usually involve a delay period during which the representation of the output category must be sustained in the absence of sensory inputs (Freedman and Assad, 2006; Fitzgerald et al., 2011; Wallis et al., 2001). Experiments indicate that category representations are different in the stimulus presentation and the delay periods (Freedman and Assad, 2006). Investigating these effects in our tasks would require the addition of recurrent connectivity to the model. Mathematical tools for analyzing learning dynamics in recurrent networks is starting to become available (Mastrogiuseppe and Ostojic, 2019; Schuessler et al., 2020; Dubreuil et al., 2022; Susman et al., 2021), which could allow our analysis to be extended in that direction.”

And

(line 656) “Finally, in this study we focused on a circuit model with a single intermediate layer. In the brain,in contrast, sensory inputs are processed across a number of stages within the cortical hierarchy. Our analysis could easily be extended to include multiple intermediate layers. That would allow our predictions to be extended to experiments involving multi-area recordings, which are increasingly common in the field (Goltstein et al., 2021). Current recording techniques, furthermore, allow monitoring neural activity throughout the learning process (Reinert et al., 2021; Goltstein et al., 2021); those data could be used in future studies to further test the applicability of our model.”

And

(line 588) “A number of recent theoretical studies have proposed biologically-plausible architectures and plasticity rules that can approximate back-propagation on simple and complex tasks (Lillicrap et al., 2016; Sacramento et al., 2018; Akrout et al., 2019; Whittington and Bogacz, 2017; Payeur et al., 2021; Pogodin and Latham, 2020; Boopathy and Fiete, 2022). Understanding whether these different implementations lead to differences in activity represents a very important direction for future research. Interestingly, recent work has showed that it is possible to design circuit models where the learning dynamics is identical to the one studied in this work, but the architecture is biologically plausible (Boopathy and Fiete, 2022). We expect our results to directly translate to those models. Other biologically-plausible setups might be characterized, instead, by different activity evolution. Recent work (Song et al., 2021; Bordelon and Pehlevan, 2022) made use of a formalism similar to ours to describe learning dynamics induced by a number of different biologically-plausible algorithms and uncovered non-trivial, qualitatively different dynamics. Whether any of these dynamics leads to different neural representations in neuroscienceinspired categorization tasks like the ones we studied here is an open, and compelling, question.”

– I assume each neuron in x is projecting to all neurons in y. Is this clearly stated somewhere? Maybe I missed it – sorry it is hot today!

The reviewer is correct. We now specify it (“all-to-all”) in the model description.

– After learning, cat selectivity, and later ctx selectivity, cover a broad range with many neurons being close to zero and some showing positive selectivity. However, most of them are still unselective it seems. Is this range in line with experimental results? Could you comment on this heterogeneity? In line 95/96 you say that the cat selectivity increases for EACH neuron but to me it seems some neurons remain at zero selectivity, do I read the figure wrong?

The theory predicts that, in the simple task, category selectivity increases for all neurons (as long as the number of neurons and stimuli is sufficiently large). This increase, however, can be very small – depending on the magnitude of the initial readout, *w*_0*,i*_ (Equation 65). Since, in our model, *w*_0*,i*_ is drawn from a centered Gaussian distribution, the selectivity increase is small for many neurons.

It is also true, however, that in finite-size simulations selectivity does not increase strictly for all the neurons (one can think of finite size as a small “noise” term on top of the theory prediction). We thus modified “each” in the main text into “most”.

Concerning heterogeneity: as mentioned in the text (line 218), experimental data display strong heterogeneity as well (see, e.g. Figure 3c of Freedman and Assad, 2006 and Figure 2f of Reinart et al., 2021). Linking the strength of selectivity increase to the strength of projections to the readout neuron, our model proposes a possible mechanism for the presence of such heterogeneity.

– The distribution of the cat selectivity between the two tasks is different. The cat selectivity for the context-dependent task has many cells with a selectivity close to zero or even negative. Could you comment on that. I apologise if I overlooked it.

We have added

“Note that the distribution of category selectivity values is different from the distribution observed in the simple task; the distribution is now heavy-tailed, with only a fraction of the neurons acquiring strong category selectivity (see also Figure 8B).”